# Accelerated Diffusion using Closed-form Discriminator Guidance

## Abstract

Diffusion models are a state-of-the-art generative modeling framework that transform noise to images via Langevin sampling, guided by the score, which is the gradient of the logarithm of the data distribution. Recent works have shown empirically that the generation quality can be improved when guided by classifier network, which is typically the discriminator trained in a generative adversarial network (GAN) setting. In this paper, we propose a theoretical framework to analyze the effect of the GAN discriminator on Langevin-based sampling, and show that in IPM GANs, the optimal generator matches *score-like* functions, involving the flow-field of the kernel associated with a chosen IPM constraint space. Further, we show that IPM-GAN optimization can be seen as one of smoothed score-matching, where the scores of the data and the generator distributions are convolved with the kernel associated with the constraint. The proposed approach serves to unify score-based training and optimization of IPM-GANs. Based on these insights, we demonstrate that closed-form discriminator guidance, using a kernel-based implementation, results in improvements (in terms of CLIP-FID and KID metrics) when applied atop baseline diffusion models. We demonstrate these results by applying closed-form discriminator guidance to denoising diffusion implicit model (DDIM) and latent diffusion model (LDM) settings on the FFHQ and CelebA-HQ datasets. We also demonstrate improvements to accelerated time-step-shifted diffusion, when coupled with a wavelet-based noise estimator for latent-space image generation.

## 1 Introduction

Generative modeling is the process of learning the underlying distribution of data, either with the aim of evaluating the density, or generating new unseen samples from the underlying distribution. Over the past few years, diffusion models (Song & Ermon, 2019; Ho et al., 2020) have become the *de facto* approach for generative modeling. Diffusion modeling treats image generation as a denoising process, and models the transformation by means of a stochastic differential equation (SDE) (Song & Ermon, 2020). The sampling process involves learning the denoising function, or equivalently, the gradient of the logarithm of the data distribution, known as the *score* (Hyvärinen, 2005), and subsequently discretizing the SDE. Diffusion models achieve state-of-the-art performance for image generation (Karras et al., 2022; Kim et al., 2023; Zheng & Yang, 2024). Prior to diffusion models, generative adversarial networks (GANs, Goodfellow et al. (2014)) were the most popular framework for image generation, owing to their superior single-step sampling performance (Karras et al., 2020; 2021; Sauer et al., 2022). As shown by Kim et al. (2023), GANs and diffusion models can be combined into a unified model, wherein the gradients of an auxiliary standard GAN (Goodfellow et al., 2014) discriminator can be used improve the score. We consider the aforementioned setting and develop strong theoretical and experimental foundations to IPM-GAN-based discriminator guidance for diffusion.

**Score-based Diffusion Models**: Score matching was originally proposed by Hyvärinen (2005) in the context of independent component analysis. Let the underlying distribution of the data to be modeled be denoted by $p_d(\boldsymbol{x})$. The *Stein score* (Liu et al., 2016) is the gradient of logarithm of the density function with respect to the data, i.e., $\nabla_{\boldsymbol{x}} \ln(p_d(\boldsymbol{x}))$. It generates a vector field that points in the direction where the data density grows most steeply. In score matching, the score can be approximated by a parametric function $S_\phi^{\mathcal{D}}(\boldsymbol{x})$ obtained by minimizing the Fisher divergence between the true score and the score estimated by the network. (Cover & Thomas, 2006) The output of the trained

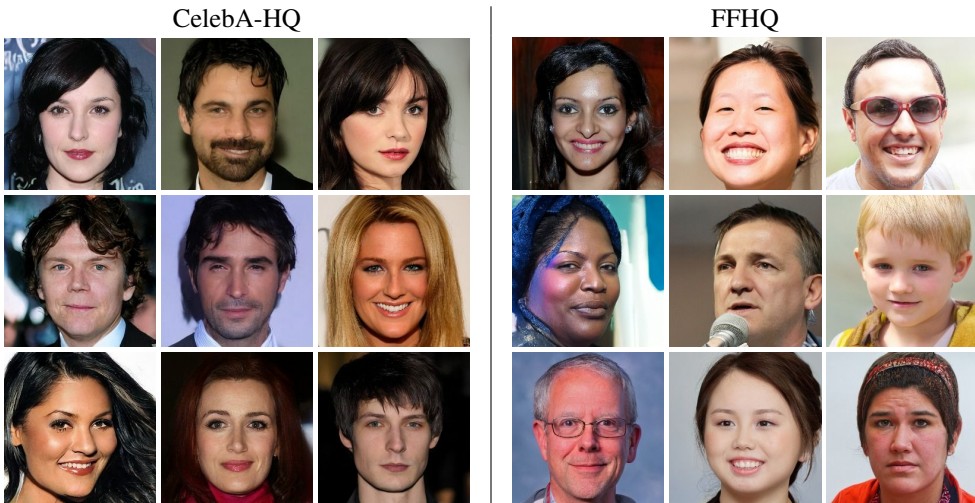

Figure 1: Images generated by the proposed closed-form discriminator guidance (DG* approach for the latent difusion model (LDM) on the 256-dimensional CelebA-HQ and FFHQ datasets.

network is used to generate samples through annealed Langevin dynamics in noise-conditioned score networks (NCSN) (Song & Ermon, 2019). Recent approaches aim at either improving the approximation quality of the score network (Song et al., 2020; Ho et al., 2020; Song & Ermon, 2020; Song et al., 2021b; Gong & Li, 2021), or better discretizing the underlying differential equations to accelerate sampling (Jolicoeur-Martineau et al., 2021; Karras et al., 2022). Upon discretization of the SDE, the evolution of the images is indexed by time $t$ is denoted as $\boldsymbol{x}_t \in \mathbb{R}^n$, with $\boldsymbol{x}_0 \sim p_d$;, and $\boldsymbol{x}_T \sim \mathcal{N}(\mathbf{0}, \mathbb{I})$, which is the standard Gaussian distribution. Image generation follows the reverse process, and is equivalent to sequentially denoising the sample $\boldsymbol{x}_T$, to ultimately generate a realistic image that ideally comes from the distribution $p_d$.

**Generative Adversarial Networks (GANs)**: GANs are a two-player game between a generator network $G\colon \mathbb{R}^d \to \mathbb{R}^n$ and a discriminator network $D\colon \mathbb{R}^n \to \mathbb{R}$, $n \gg d$. Similar to the reverse process in diffusion, the generator transforms a noise vector $\boldsymbol{z} \sim p_{\boldsymbol{z}}$; $\boldsymbol{z} \in \mathbb{R}^d$, typically standard Gaussian, and transforms it into a *fake* sample $G(\boldsymbol{z})$, with the push-forward distribution $p_g = G_\#(p_{\boldsymbol{z}})$. The discriminator accepts an input drawn either from the target distribution, $\boldsymbol{x} \sim p_d$; $\boldsymbol{x} \in \mathbb{R}^n$, or from the output of a generator, and learns a *real versus fake* classifier. The objective is to learn the *optimal generator* that can create realistic samples, which is equivalent to modeling the reverse process in a single step. GAN literature considers two main classes of loss functions: (a) $f$-divergence-based losses, and (b) integral probability metric (IPM) based losses. The standard GAN (SGAN, Goodfellow et al. (2014)), least-squares GAN (LSGAN, Mao et al. (2017)) and $f$-GANs (Nowozin et al., 2016) formulations, fall into the first category, wherein the discriminator models a chosen *divergence* metric between the target and generator distributions, while the generator network is trained to minimize this divergence. In IPM-GANs, the discriminator performs the role of a *critic*, and approximates the IPM, which in turn relates to a constraint class. For example, in Wasserstein GAN (WGAN), Arjovsky et al. (2017) consider Lipschitz-1 critics, while variants such as the Sobolev GAN Mroueh et al. (2018), BWGAN Adler & Lunz (2018), and PolyGAN Asokan & Seelamantula (2023a) consider discriminator functions drawn from Sobolev spaces, with a corresponding penalty on the energy in the gradient. Gretton et al. (2012) showed that the minimization of IPM losses can be equivalently solved through the minimization of kernel-based statistics in a reproducing-kernel Hilbert space (RHKS). Maximum-mean discrepancy GANs (MMD-GANs) (Li et al., 2017; Bińkowski et al., 2018) and Coulomb GAN (Unterthiner et al., 2018) are examples of kernel-based GANs.

*GAN Discriminator Guidance in Diffusion Models*: Dhariwal & Nichol (2021) and Ho & Salimans (2022) proposed the use of classifier gradients in conjunction with the score estimate of a diffusion model to improve the diversity of conditional image generation. Kim et al. (2023) were the first to leverage the GAN discriminators, and showed that the score learnt at the time instant $t$ in the NCSN (Song & Ermon, 2019) could be improved by a correction term involving the SGAN discriminator gradients. Subsequently, Naderiparizi et al. (2024); Um et al. (2024); Bansal et al. (2023) and Yang et al. (2024) have also explored discriminator guidance for superior coverage of the

image manifold in diffusion models, while Ekström Kelvinius & Lindsten (2024) and Kerby & Moon (2024) proposed discriminator guidance paired with discrete diffusion models for molecular graph generation. However, these approaches typically either consider only the SGAN discriminator, or are unable to provide an explanation for the the effectiveness of discriminator guidance when going beyond the SGAN setting.

***Unifying GANs and Diffusion Models***: There has been a significant research focus on the optimality of the GAN discriminator function. Mroueh et al. (2018); Zhu et al. (2020); Liang (2021); Franceschi et al. (2022); Yi et al. (2023); Asokan & Seelamantula (2023b) consider a functional approach, and derive the differential equations that govern the optimal discriminator, given the generator. Along another vertical, Pinetz et al. (2018), Stanczuk et al. (2021) and Korotin et al. (2022) showed that, in practical gradient-descent-based training, the optimal discriminator is not attained. In the recent past, there has been a strong push to develop a unifying theory to explain GAN optimization, potentially leveraging results from flow-based approaches. For example, Yi et al. (2023); Heng et al. (2023) propose a unifying theory for all $f$-GANs under the umbrella of Wasserstein flows, while (Asokan et al., 2023) link the generator optimization in SGANs to score-based sampling, and Franceschi et al. (2023); Zhang et al. (2023) formulate both GANs and score-based diffusion models as special cases of particle flows. While in most scenarios, the generator can be linked to minimizing the chosen divergence or IPM, the actual functional optimization has not been thoroughly explored. Motivated by the strong links between the guidance in diffusion and the GANs discriminator (Kim et al., 2023), and the equivalences between GAN training and Langevin sampling (Franceschi et al., 2023), in this paper, we seek to answer the question: **How does the closed-form optimization of the GAN generator link to discriminator guidance for diffusion?**

## 1.1 OUR CONTRIBUTIONS

In this paper, we analyze the links between GAN optimization and score-based diffusion, and provide a principled approach to applying IPM-GAN discriminator guidance for diffusion models. We consider the GAN optimization setting, and draw a parallel between the generator optimization in GANs and score-based diffusion. When analyzed through the lens of *Variational Calculus*, the generator optimality condition in divergence-minimizing and IPM-based GAN formulations closely resembles the score-matching condition seen in diffusion models. Considering the family of $f$-GANs, we extend the analysis of Asokan et al. (2023) to the optimization of the generator loss in IPM-GANs, given the optimal discriminator. We show that the optimal generator in these settings minimizes a *smoothed score-matching* difference term, where the scores are conditioned by means of the kernel associated with the reproducing kernel Hilbert space (RKHS) from which the IPM discriminator is drawn, akin to noise-conditioned score networks (NCSN) (Song & Ermon, 2019). Futher, we show that, in IPM GANs, the *smoothed score-matching* formulation is equivalent to one of minimizing a flow induced by the gradient field of a kernel function (cf. Section 3). These results can be viewed as a generalization of Sobolev descent (Mroueh et al., 2019), MMD-Flows (Arbel et al., 2019) and MonoFlows (Yi et al., 2023). The results establish a fundamental connection between GANs, score-based models, and flow-based generative models. Leveraging these insights, we employ the closed-form IPM-GAN discriminator as a guidance term in score-based diffusion. Leveraging a kernel-based discriminator enables the proposed closed-form discriminator guidance (abbreviated DG$^*$) approach to be compatible with any existing Langevin sampling framework. We show that the guidance model can also be deployed in Langevin sampling without explicit use of the score function (cf. Section 4). Proceeding further, we include closed-form discriminator guidance (DG$^*$) in the elucidating the design space of diffusion models (EDM) setting (cf. Section 4) and latent-space diffusion models (LDM) (cf. Section 5). Lastly, considering time-step-shifted diffusion, we show that the inclusion of DG$^*$ can also accelerate the denoising process, allowing for larger jumps in noise levels when transitioning from discriminator guidance to score-based sampling.

Our **key contributions** are two-fold: We develop a strong theoretical foundation for employing closed-form IPM-GAN discriminators for guidance, based on the established equivalence between GAN-generator optimality and a smoothed version of the score-matching constraint. We leverage these insights to develop a novel closed-form discriminator guidance framework that be applied in a *plug-and-play* fashion with an existing diffusion model. We demonstrate this capability through experimental results on NCSN (Song & Ermon, 2019), EDM (Karras et al., 2022), trainable discriminator-guidance (Kim et al., 2023), and LDMs (Rombach et al., 2022).

## 2 BACKGROUND ON DIFFUSION AND GANS

In this section, we briefly introduce the training and sampling procedure in diffusion probabilistic models (DPM), Latent Diffusion Models (LDM), and GANs.

### 2.1 DIFFUSION PROBABILISTIC MODELS

Diffusion probabilistic models (DPMs) primarily model the *forward process* wherein Gaussian noise is progressively added to an image $\boldsymbol{x} \sim p_d$. The noise is modelled as adhering to a fixed variance schedule $\beta(t)$. The generative task is one of modeling the reverse process, essentially iterated denoising. Given the data distribution $p_d$ and a fixed noise schedule $\beta(t) \in (0,1), \forall t = 1 \dots T$, the forward process, structured as a Markov process, is expressed as $p(\boldsymbol{x}_{1,2,\dots,T}|\boldsymbol{x}_0) = \prod_{t=1}^{T} p(\boldsymbol{x}_t|\boldsymbol{x}_{t-1})$. In the DPM setting, the forward transition kernel at time $t$, given by $p(\boldsymbol{x}_t|\boldsymbol{x}_{t-1})$ can be defined as a Gaussian $\mathcal{N}(\sqrt{\alpha_t}\boldsymbol{x}_{t-1}, \beta_t\mathbb{I})$, centered around the sample of the previous time instant $\sqrt{\alpha_t}\boldsymbol{x}_{t-1}$, where $\alpha_t = 1 - \beta_t$ (Ho et al., 2020). By means of the reparameterization trick, the conditional distribution can be expressed as:

$$p(\boldsymbol{x}_t|\boldsymbol{x}_0) = \sqrt{\bar{\alpha}_t}\boldsymbol{x}_0 + \sqrt{1-\bar{\alpha}_t}\epsilon_t \quad \Rightarrow \quad p(\boldsymbol{x}_{t-1}|\boldsymbol{x}_t, \boldsymbol{x}_0) = \mathcal{N}(\tilde{\mu}_t, \tilde{\beta}_t) \tag{1}$$

wherein, $\bar{\alpha}_t = \prod_{i=1}^{t} \alpha_i$ and $\epsilon_t \sim \mathcal{N}(\mathbf{0}, \mathbb{I})$, $\tilde{\mu}_t = \frac{1}{\sqrt{\alpha_t}}\left(\boldsymbol{x}_t - \frac{1-\alpha_t}{\sqrt{1-\bar{\alpha}_t}}\epsilon_t\right)$, $\tilde{\beta}_t = \frac{(1-\bar{\alpha}_{t-1})}{1-\bar{\alpha}_t}\beta_t$
and $p(\boldsymbol{x}_0) = p_d$. Training DPMs involves learning a neural network $\epsilon_\theta$ to approximate $\epsilon_t$, with the following mean-squared-error loss Song et al. (2021a):

$$\mathcal{L}_{\text{DPM}} = \mathbb{E}_{t,\boldsymbol{x}_t,\epsilon_t \sim \mathcal{N}(0,\mathbb{I})}[\|\epsilon_\theta(\boldsymbol{x}_t, t) - \epsilon_t\|_2^2] \tag{2}$$

In practice, the model is trained on a variational lower bound of the negative log-likelihood loss. Consequently, generation starts by sampling $\boldsymbol{x}_T$ from a standard Gaussian, *i.e.,* $\boldsymbol{x}_T \sim \mathcal{N}(\mathbf{0}, \mathbb{I})$, and progressively generating samples according to the backward recursion:

$$\boldsymbol{x}_{t-1} = \mu_\theta(\boldsymbol{x}_t, t) + \Sigma_\theta(\boldsymbol{x}_t, t).\boldsymbol{z}_t, \text{ where } \boldsymbol{z}_t \sim \mathcal{N}(\mathbf{0}, \mathbb{I}), \text{ and } t = T, T-1, \dots, 1, 0$$

where $\mu_\theta$ and $\Sigma_\theta$ are the estimates of the noise mean and covariance, as output by $\epsilon_\theta$. The SDE governing the above process was generalized by Song et al. (2021a), wherein the discretized update is given by:

$$\boldsymbol{x}_{t-1} = \underbrace{\sqrt{\frac{\alpha_{t-1}}{\alpha_t}}\boldsymbol{x}_t - \sqrt{\frac{\alpha_{t-1}}{\alpha_t}}\sqrt{(1-\alpha_t)}\epsilon_\theta(\boldsymbol{x}_t, t)}_{\hat{\boldsymbol{x}}_0} + \sqrt{(1-\alpha_{t-1}) - \sigma_t^2} \cdot \epsilon_\theta(\boldsymbol{x}_t, t) + \sigma_t\epsilon_t \tag{3}$$

where $\hat{\boldsymbol{x}}_0$ can be viewed as the *prediction* of $\boldsymbol{x}_0$; the term $\sqrt{(1-\alpha_{t-1}) - \sigma_t^2} \cdot \epsilon_\theta^t(\boldsymbol{x}_t)$ represents the direction pointing towards $\boldsymbol{x}_t$ with $\alpha_0 = 1$; and $\sigma_t\epsilon_t$ is the diffusion term with $\epsilon_t \sim \mathcal{N}(0, \mathbb{I})$ being standard Gaussian and independent of $\boldsymbol{x}_t$. Different values of $\sigma$ lead to different generative processes while keeping $\epsilon_\theta$ fixed, thus removing the necessity to retrain the models. When $\sigma_t$ is set to $\sqrt{(1-\alpha_{t-1})/(1-\alpha_t)}\sqrt{(1-\alpha_t/\alpha_{t-1})}$, for all $t$, the resulting generative process becomes DDPM Song et al. (2021a). On the other hand, when $\sigma_t = 0$ for all $t$, the samples generated obey a deterministic procedure and this specific generative trajectory is referred to as denoising diffusion implicit model (DDIM) sampling. DDIM sampling can generate high-quality samples with fewer time-steps $\tau < T$ with no changes in the training procedure of the DDPM denoiser $\epsilon_\theta$ which was trained over $T$ timesteps. In general, we can set $\sigma_{\tau(\eta)} = \eta\sqrt{(1-\alpha_{t-1})/(1-\alpha_t)}\sqrt{(1-\alpha_t/\alpha_{t-1})}$ to interpolate between the DDPM and DDIM settings (Song et al., 2021a). The choice of $\eta$ directly controls the stochasticity in sampling, with $\eta = 1$ and $\eta = 0$ corresponding to DDPM and DDIM, respectively. In this work, we explore the inclusion of closed-form discriminator guidance in the DDIM setting.

### 2.2 OPTIMALITY OF GANS

GAN optimization can be viewed as minimizing either the $f$-divergence between the target distribution $p_d$ and the distribution of the generated samples (denoted as $p_g$), or an integral probability metric (IPM) between $p_d$ and $p_g$. Nowozin et al. (2016) proposed $f$-GANs, considering $f$-divergences

of the form: $\mathfrak{D}_f(p_d \| p_{t-1}) = \int_{\mathcal{X}} f\left(r_{t-1}(\boldsymbol{x})\right) p_d(\boldsymbol{x}) \, d\boldsymbol{x}$, where $f \colon \mathbb{R}_+ \to \mathbb{R}$ is a convex, lower-semicontinuous function over the support $\mathcal{X}$ and satisfies $f(1) = 0$ and $r_{t-1}(\boldsymbol{x})$ is the density ratio $r_{t-1}(\boldsymbol{x}) = \frac{p_d(\boldsymbol{x})}{p_{t-1}(\boldsymbol{x})}$. The optimization is given by

$$\min_G \left\{ \max_D \left\{ \mathbb{E}_{\boldsymbol{x} \sim p_d}[T(\boldsymbol{x})] - \mathbb{E}_{\boldsymbol{x} \sim p_g}[f^c(T(G(\boldsymbol{z})))] \right\} \right\}, \tag{4}$$

where $T(\boldsymbol{x}) = g(D(\boldsymbol{x}))$, is the output of the discriminator $D$ subjected to the activation $g$, and $D^*(\boldsymbol{x})$ is the optimal discriminator, and $f^c$ denotes the Fenchel conjugate of $f$. In practice, the optimization is an alternating one, wherein the discriminator $D_t$ is derived given the generator of the previous iteration $G_{t-1}$, and the subsequent generator optimization involves computing $G_t$, given $D_t$ and $G_{t-1}$. Within this setting, (Asokan et al., 2023) presented the following result:

**Theorem 1.** *(Asokan et al., 2023) Consider the generator loss in $f$-GANs, given by Equation (4). The **optimal $f$-GAN generator** satisfies the following score-matching condition: $r_{t-1}(\boldsymbol{x})g'(t)\big|_{t=D_t^*} D_t^{*\prime}(y)\big|_{y=\ln(r_{t-1})} \nabla_{\boldsymbol{x}}\left(\ln r_{t-1}(\boldsymbol{x})\right) = \boldsymbol{0}$, where $g'(t)$ denotes the derivative of the activation function with respect to $D$ evaluated at $D_t^*$, $D_t^{*\prime}(y)$ denotes the derivative of the optimal discriminator function with respect to $y = \ln(r_{t-1}(\boldsymbol{x}))$, evaluated at $\ln(r_{t-1}(\boldsymbol{x}))$. For $\boldsymbol{z}$ such that $r_{t-1}(\boldsymbol{x})g'(t)D_t^{*\prime}(y) \neq 0$, the optimization yields the score-matching cost:*

$$\nabla_{\boldsymbol{x}} \ln\left(p_{t-1}(\boldsymbol{x})\right)\big|_{\boldsymbol{x}=G_t^*(\boldsymbol{z})} = \nabla_{\boldsymbol{x}} \ln\left(p_d(\boldsymbol{x})\right)\big|_{\boldsymbol{x}=G_t^*(\boldsymbol{z})}.$$

In the IPM-GAN setting, Arjovsky et al. (2017) proposed Wasserstein GANs (WGANs) as an alternative to divergence-minimizing GANs. Motivated by *optimal transport*, the discriminator (also called the *critic*) minimizes the Wasserstein-1 distance between $p_d$ and $p_g$. The IPM GAN optimization is defined through Kantorovich–Rubinstein duality as:

$$\min_{p_g} \left\{ \max_D \left\{ \mathbb{E}_{\boldsymbol{x} \sim p_d}[D(\boldsymbol{x})] - \mathbb{E}_{\boldsymbol{x} \sim p_g}[D(\boldsymbol{x})] + \Omega_D \right\} \right\}, \tag{5}$$

where $\Omega_D$ is an appropriately chosen regularizer. Arjovsky et al. (2017) enforced a Lipschitz-1 discriminator by clipping the network weights. Subsequent variants considered regularizers that bound the energy in the discriminator gradient (Petzka et al., 2018; Mroueh et al., 2018; Adler & Lunz, 2018; Asokan & Seelamantula, 2023a), resulting in Sobolev constraint spaces. The optimal discriminator in these variants has been shown to be the solution to partial differential equations (PDEs) (Mroueh et al., 2018; Asokan & Seelamantula, 2023a), which can be represented through convolutions with the Green's function of the PDEs. As in the case of $f$-GANs, consider the alternating minimization involving $G_{t-1}$, $D_t$ and $G_t$. The optimal discriminator in gradient-regularized WGANs is given by a kernel-based convolution (Unterthiner et al., 2018; Asokan & Seelamantula, 2023a):

$$D_t^*(\boldsymbol{x}) = \mathfrak{C}_\kappa\left((p_{t-1} - p_d) * \kappa\right)(\boldsymbol{x}), \tag{6}$$

where the kernel $\kappa$ is the Green's function to the differential operator governing the optimal discriminator and $\mathfrak{C}_\kappa$ is a positive constant. In Poly-WGAN (Asokan & Seelamantula, 2023a), the kernel corresponds to the family of polyharmonic splines, given by

$$\kappa(\boldsymbol{x}) = \begin{cases} \|\boldsymbol{x}\|^k & \text{if } k < 0 \text{ or } n \text{ is odd,} \\ \|\boldsymbol{x}\|^k \ln(\|\boldsymbol{x}\|) & \text{if } k \geq 0 \text{ and } n \text{ is even,} \end{cases}$$

where in turn, $k = 2m - n$, $m$ being a hyper-parameter that controls to smoothness of the discriminator and $n$ is the dimensionality of the data. In this paper, we extend the results derived for $f$-GANs to the IPM-GAN setting, and leverage the resulting solution for discriminator guidance in DDIMs.

We now derive the optimality condition on the IPM-GAN generator, and derive its relationship to score-based diffusion.

## 3 THE OPTIMAL GENERATOR IN IPM GANS

To motivate our results, consider the solution to Theorem 1. We observe that the optimal $f$-GAN generator is the one that matches the score of the generator push-forward distribution to the score of the data distribution. While this results in the dicriminator guidance framework (Kim et al., 2023),

$f$-divergence GANs are known to be unstable to train (Arjovsky & Bottou, 2017; Kim et al., 2023). Furthermore, as noted by (Yi et al., 2023), $f$-GANs can be viewed as a special case of IPM-GANs. Therefore, we derive the general solution to generator optimality that holds for all IPM-GAN variants. Consider the IPM-GAN optimization problem given in Equation (5). The following theorem presents the optimality condition for the generator in kernel-based GANs:

**Theorem 2.** *Consider the generator loss given by* $\mathcal{L}_G^\kappa(G; D_t^*, G_{t-1}) = -\mathbb{E}_{\boldsymbol{z} \sim p_{\boldsymbol{z}}}[D_t^*(G(\boldsymbol{z}))]$, *and the optimal discriminator given in Equation 6. The **optimal IPM-GAN generator** satisfies*

$$\mathfrak{C}_\kappa \left( \underset{\boldsymbol{y} \sim p_{t-1}}{\mathbb{E}} \left[ \nabla_{\boldsymbol{y}} \ln p_{t-1}(\boldsymbol{y}) \kappa(\boldsymbol{x} - \boldsymbol{y}) \right] - \underset{\boldsymbol{y} \sim p_d}{\mathbb{E}} \left[ \nabla_{\boldsymbol{y}} \ln p_d(\boldsymbol{y}) \kappa(\boldsymbol{x} - \boldsymbol{y}) \right] \right) \Big|_{\boldsymbol{x} = G_t^*(\boldsymbol{z})} = \boldsymbol{0}, \quad (7)$$

*for all* $\boldsymbol{x} = G_t^*(\boldsymbol{z})$, $\boldsymbol{z} \sim p_{\boldsymbol{z}}$, *where* $\mathfrak{C}_\kappa$ *is a non-zero constant dependent on the kernel* $\kappa$.

The above theorem shows that the optimal generator in IPM GANs is also one of score-matching, where the score is conditioned by the kernel function, centered around $\boldsymbol{x}$. We observe that the condition presented in Theorem 2 is equivalent to a condition on the kernel gradient, given by the following lemma.

**Lemma 3.** *Consider the optimality condition for the IPM generator, presented in Theorem 2. The condition can be written equivalently as:* $\mathfrak{C}_\kappa \left( (p_d - p_{t-1}) * \nabla_{\boldsymbol{x}} \kappa \right) (\boldsymbol{x}) \big|_{\boldsymbol{x} = G_t^*(\boldsymbol{z})} = \boldsymbol{0}$, *where* $\nabla_{\boldsymbol{x}} \kappa$ *denotes the gradient vector of the kernel, and the convolution must be interpreted element-wise, i.e.,* $p_d(\boldsymbol{x}) - p_{t-1}(\boldsymbol{x})$ *is convolved with each entry of* $\nabla_{\boldsymbol{x}} \kappa$.

The proof of Theorem 2 and Lemma 3 and are presented in detail in Appendix D.1. The optimal IPM-GAN generator can be seen as minimizing a proxy to the score — similar to the Stein score — where the gradient field induced by the kernel $\kappa$ is maximized at locations where data samples are present. As observed in Coulomb GANs, these are akin to charge-potential fields, with *attractive* data samples and *repulsive* generator samples. While we use the polyharmonic spline kernel for the choice of $\kappa$ due to its stability (Asokan & Seelamantula, 2023a), a discussion on other choices is presented in Appendix D.

### 3.1 LINKING THE OPTIMAL IPM-GAN GENERATOR TO SCORE-BASED DIFFUSION

Based on the theoretical insights, we see that, given the optimal discriminator $D_t^*$ that admits a kernel-based interpolation form at training iteration $t - 1$, the optimal generator at the subsequent iteration $G_t^*$ can be derived as a one that minimizes the value of the convolution between the density difference, and the gradient of the optimal discriminator kernel, *i.e.,*, minimize $((p_d - p_t) * \nabla \kappa)$. For most popular positive-definite kernels $\kappa$ (cf. Table 3 of the Appendix), this term would be minimized when the generator distribution $p_t$ moves towards the data distribution $p_d$. Furthermore, from Lemma 3, we see that the gradient field of the kernels convolved with the density difference, and the data score $\nabla_{\boldsymbol{x}} \ln (p_d(\boldsymbol{x}))$, serve similar purposes, which is to output an arbitrarily large value at data sample location, and low values elsewhere. Unlike the score, however, the kernel gradients produce a repulsive force at the location of generator samples, resulting in a *push-pull* framework – The target distribution creates a *pull*, while the generator distribution creates the *push*. This serves to validate why IPM GANs typically do not suffer from vanishing gradients (Arjovsky & Bottou, 2017), as opposed to the $f$-divergence counterparts. When $p_0(\boldsymbol{x})$ is initialized far from the target, although the *influence* of the score is weak, the repulsive force of the kernel-based loss is strong. The derived solution can also be used to explain denoising diffusion GANs (DDGAN, Xiao et al. (2022)), wherein a GAN is trained to model the reverse diffusion process, with the generator and discriminator networks conditioned on the time index. DDGAN can be seen as a special instance of our approach, with Langevin updates over the gradient field of the time-conditioned discriminator (cf. Appendix D). The kernel-convolved score-matching condition can also be viewed as generalized score matching (Lyu, 2009) where the IPM-GAN generators minimize a *generalized score*, *i.e.,* given an IPM GAN, an equivalent diffusion model exists, with the flow field induced by the kernel of the discriminator, and vice versa. We demonstrate this approach in Section 4.

This results allows us to explore Langevin sampling, wherein the score of the data is either replaced, or guided using the gradient of the kernel-based discriminator. While the score of the data possesses a *strong attractive force* in regions close to the target data, it does not significantly influence samples that are far away. On the other hand, the kernel gradients possess a repulsive term that *pushes* particles

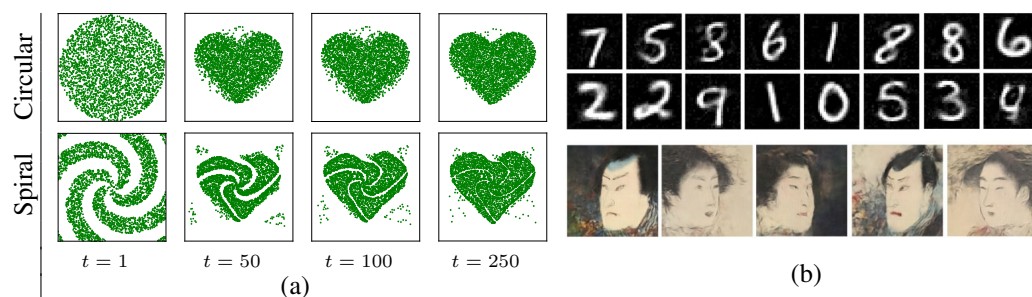

Figure 2: (🎨 Color online) (a) Shape morphing using the proposed discriminator-guided Langevin sampler. For relatively simpler input shapes, such as the circular pattern, the sampler converges in about 100 iterations, while in the spiral case, the sampler converges in about 500 steps. (b) Images generated using the discriminator-guided Langevin sampler on MNIST and Ukiyo-E faces datasets. The score in standard diffusion models is replaced with the gradient field of the discriminator, obviating the need for training a neural network.

away from where they previously were, thereby accelerating convergence. We consider the following update scheme:

$$\boldsymbol{x}_{t+1} = \boldsymbol{x}_t - \alpha_t \nabla_{\boldsymbol{x}} D_t^*(\boldsymbol{x}_t) + \gamma_t \boldsymbol{z}_t, \quad \text{where} \quad \boldsymbol{z}_t \sim \mathcal{N}(\boldsymbol{0}_n, \mathbb{I}_n)$$

and the discriminator gradient is an $N$-sample estimate with centers consisting of data samples $\boldsymbol{d}^i \sim p_d$, and the set of samples generated at the previous iteration $\{\boldsymbol{x}_{t-1} \mid \boldsymbol{x}_{t-1} \sim p_{t-1}\}$, given by:

$$\nabla_{\boldsymbol{x}} D_t^*(\boldsymbol{x}_t) = \mathfrak{C}_k' \sum_{\boldsymbol{g}^j \sim \{\boldsymbol{x}_{t-1}\}} \nabla_{\boldsymbol{x}} \kappa(\boldsymbol{x}_t - \boldsymbol{g}^j) - \mathfrak{C}_k' \sum_{\boldsymbol{d}^i \sim p_d} \nabla_{\boldsymbol{x}} \kappa(\boldsymbol{x}_t - \boldsymbol{d}^i). \tag{8}$$

Typically, $\gamma_t = \sqrt{2\alpha_t}$, while $\alpha_t$ is decayed geometrically (Song & Ermon, 2019). Within this framework, the training time is *traded in* for memory overhead. We do not require a trained score/discriminator network, but require random batches of samples drawn $\{\boldsymbol{d}^i \sim p_d\}$ at each sampling step.

## 4 EXPERIMENTATION – DISCRIMINATOR-GUIDED LANGEVIN DIFFUSION

To demonstrate the performance of the discriminator-guided Langevin flow, we consider shape morphing, proposed by Mroueh et al. (2019). The source and target samples are drawn uniformly from the interior regions of pre-defined shapes. Figure 7(a) depicts two such scenarios, where the target shape is a heart, and the input shapes are a disk, and a spiral, respectively. Additional combinations are presented in Appendix E. The discriminator-guided Langevin sampler converges in about 500 iterations in all the scenarios considered, compared to the 800 iterations reported in Sobolev descent (Mroueh et al., 2019; Mroueh & Rigotti, 2020), without the need for training a network approximation of the discriminator.

We extend the proposed approach to images, considering MNIST, SVHN and Ukiyo-E (Pinkney & Adler, 2020) datasets. Ablation experiments on the choice of $\alpha_t$ and $\gamma_t$ are provided in Appendix E. Figure 7(b) presents the samples generated by this discriminator-guided Langevin sampler on MNIST and 256-dimensional Ukiyo-E faces. The model converges to realistic images in as few as 300 steps of sampling, resulting in performance comparable to baseline NCSN (Song & Ermon, 2019). Subsequent iterations, akin to NCSN models, serve to *clean* the noisy images generated. Additional experiments are provided in Appendix E.

Since the proposed approach suggests the interoperability of the score and the discriminator-kernel gradient in Langevin flow, we also consider discriminator-guided Langevin sampling on the CIFAR-10 and ImageNet-64 datasets, considering EDMs as the baseline (Karras et al., 2022). In both the scenarios, we also replace the sampler in discriminator-guided Langevin diffusion with the one used for the baseline considered by Karras et al. (2022). Based on the experiments in Appendix F of the present submission, we replace the score with the gradient of the polyharmonic kernel discriminator, with a constant coefficient, and ignore the exploratory noise term in our approaches. Images generated by the proposed method are provided in Figure 3, while side-by-side comparisons with the baseline

**Ours** + Heun sampler (40 steps)          **Ours** + EDM sampler (80 steps)

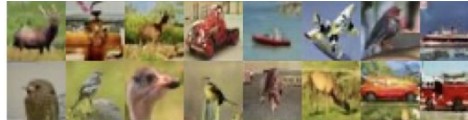

Figure 3: (🎨 Color online) Samples generated by the proposed discriminator-guided Langevin diffusion on the CIFAR-10 and ImageNet-64 datasets, using the second-order Heun and EDM samplers, respectively, and sampling parameters as described by Karras et al. (2022) for the baseline. While the images generated by the proposed approach lack diversity, the sampler converges in fewer steps and generation is performed without having to train a score network.

Table 1: A comparison of the proposed closed-form discriminator guidance for LDM (LDM+DG*) and the baseline LDM sampler on CelebA-HQ and FFHQ datasets, in terms of standard evaluation metrics. LDM+DG* outperforms the baseline on the Clean-FID, CLIP-FID and KID metrics. * While the FID reported by (Rombach et al., 2022) is 5.11, we were unable to reproduce these numbers (even with pre-trained models) using standard metric libraries (Clean-FID (Parmar et al., 2021) and Torch Fidelity (Obukhov et al., 2020)). A † denotes a metric computed via Torch Fidelity, and ‡ denotes a metric computed via Clean-FID.

|  | Method | *FID† ↓ | Clean-FID‡ ↓ | CLIP-FID‡ ↓ | KID‡ ↓ | Precision† ↑ | Recall† ↑ |
|---|---|---|---|---|---|---|---|
| CelebA-HQ | LDM | **18.21** | 21.53 | 7.17 | $2.208 \times 10^{-2}$ | **0.5434** | 0.4406 |
| | LDM+DG* (**Ours**) | 18.46 | **20.49** | **6.48** | $\mathbf{2.041 \times 10^{-2}}$ | 0.4932 | 0.4806 |
| | WANDA (**Ours**) | 19.84 | 22.76 | 7.98 | $2.270 \times 10^{-2}$ | 0.4570 | **0.4990** |
| FFHQ | LDM | **10.972** | 8.65 | 7.16 | $3.43 \times 10^{-3}$ | **0.545** | 0.563 |
| | LDM+DG* (**Ours**) | 11.056 | **7.92** | **6.51** | $\mathbf{3.02 \times 10^{-3}}$ | 0.537 | **0.571** |
| | WANDA (**Ours**) | 11.787 | 8.79 | 7.06 | $3.39 \times 10^{-3}$ | 0.540 | 0.568 |

EDM are provided in Appendix E (cf. Figures 15-23). For CIFAR-10, we consider the second-order Heun sampler with 128 sampler steps in the baseline, while the proposed approach converges in 40 steps. For ImageNet-64, the baseline EDM sampler took 255 steps, while discriminator-guided Langevin diffusion took 80 steps to converge.

However, we observe two limitations to this brute-force approach. First, diffusion models like EDM (Karras et al., 2022) and NCSN (Song & Ermon, 2019) work directly on the pixel space, making both the training and inference of the score network, and the evaluation of the closed-form discriminator computationally expensive. These approaches are therefore infeasible on high-resolution datasets such as CelebA-HQ (Karras et al., 2018) and FFHQ (Karras et al., 2019). Furthermore, we observe that the inclsion of the discriminator guidance over all iterations may not be optimal. As we observe from Figure 3 that the inclusion of discriminator guidance at all time stems might worsen image quality. We now present approaches to circumvent these two challenges in Section 5

## 5 EXTENSION TO LATENT DIFFUSION MODELS

Given the limitations of the pixel-space generation given above, we extend the closed-form discriminator-guidance approach to latent diffusion models (LDMs) (Vahdat et al., 2021; Rombach et al., 2022). The modified latent-space DDIM update with discriminator guidance is:

$$\boldsymbol{e}_{\boldsymbol{x}_{t-1}} = \sqrt{\frac{\alpha_{t-1}}{\alpha_t}}\boldsymbol{e}_{\boldsymbol{x}_t} - \sqrt{\frac{\alpha_{t-1}}{\alpha_t}}\sqrt{(1-\alpha_t)}\epsilon_\theta(\boldsymbol{e}_{\boldsymbol{x}_t}, t)$$

$$+ \sqrt{(1-\alpha_{t-1})-\sigma_t^2} \cdot \epsilon_\theta(\boldsymbol{e}_{\boldsymbol{x}_t}, t) + \sigma_t \epsilon_t + w_{dg,t}\nabla_{\boldsymbol{e}_{\boldsymbol{x}}} D_t^*(\boldsymbol{e}_{\boldsymbol{x}_t}),$$

where $w_{dg,t}$ is a temporal weighting factor to gradually decay the effect of the closed-form discriminator guidance (DG*) term and $\boldsymbol{e}_{\boldsymbol{x}} = \mathcal{E}_{\text{LDM}}(\boldsymbol{x})$ is the LDM-encoded representation of $\boldsymbol{x}$. The resulting LDM baseline is therefore a DDIM sampler working on encoder representations. Experimentally, we

| LDM | LDM+DG* | WANDA |
|---|---|---|

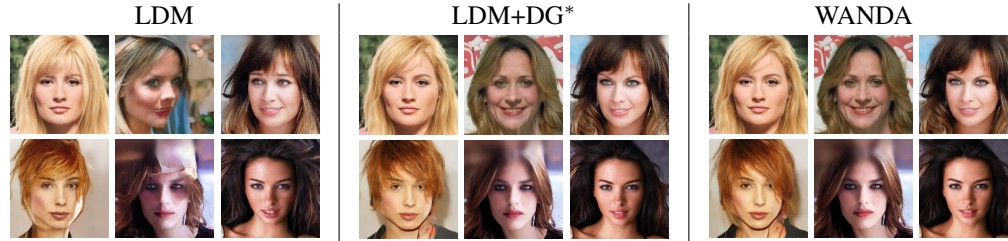

Figure 4: (🎨 Color online) A comparison of the 256-dimensional CelebA-HQ images generated (given the same input) by the baseline latent diffusion model (LDM), and the proposed closed-form discriminator guidance models with and without time-step-shifted sampling (WANDA and LDM-DG*, respectively). The discriminator guidance in LDM-DG* significantly improves the quality of the images generated, by removing artifacts. WANDA is capable of generating images with a quality comparable to that of LDM-DG*, with relatively fewer function evaluations.

found that setting $w_{dg,T} = 5$ with an exponential decay resulted in superior image generation quality. Ablations on this choice are discussed in Section 5.1

Figure 4 presents the samples generated using vanilla LDM update and LDM+DG* approach sampled using the equation above, on CelebA-HQ. Similar comparisons on the FFHQ dataset are provided in Appendix E. Both approaches are initialized with the deterministic sampler ($\eta = 0$) on the CelebA-HQ dataset while with the stochastic sampler ($\eta = 1$) on the FFHQ dataset. We observe that the LDM-DG* sampler converges to different samples and results in visually superior images in comparison to the vanilla DDIM. Table 1 presents the standard performance metrics — FID (Parmar et al., 2021), KID (Bińkowski et al., 2018), CLIP-FID (Kynkäänniemi et al., 2023), and precision-recall (Kynkäänniemi et al., 2019) scores. LDM+DG* outperforms the baseline in terms of the Clean-FID, CLIP-FID and KID metrics.

Given the acceleration that was shown by EDM+DG* setting, we also explore accelerating the LDM+DG* sampler, using time-step shifted samples, proposed by Li et al. (2024)

**Discriminator Guidance with Time-Shifted Sampling**: Li et al. (2024) proposed the time-shifted sampler to mitigate *exposure bias* in DPMs caused due to poor inference-time generalization, *i.e.,* $\epsilon_\theta$ is trained on ground-truth samples $x_t$, but inference is performed on $\hat{x}_{t-1}$. Due to this discrepancy between training and generated samples, the exposure bias accumulates across the reverse process, causing it to divert from the intended trajectory. To mitigate this issue, given the sample $\hat{x}_t$ an estimate of the noise variance in the image is used to evaluate a superior coupling time $t_s$ than the iteration's backward time $t$. Further, they also show that diffusion models basically contain *two stages* – The initial phase, wherein the input Gaussian distribution moves towards the image space, and the second phase, wherein patterns and structure emerge from latching onto a specific image to generate. Acceleration mechanisms such as time-step shifting (Li et al., 2024) and the proposed DG* operate in the first stage, which is why we focus the discriminator guidance to earlier iterations. Motivated by the fact that LDM+DG*, when applied for all time steps reduces images quality, (cf. Figure 3) we adopt the time-shifted discriminator-guided diffusion strategy to ensure that the effect of discriminator guidance is restricted to the earlier step. However, we observed that the noise-variance estimation technique proposed in the baseline was at a pixel-level sample estimate and could be improved. In particular, Mallat (2009) and Donoho (1995) showed that, in the context of image denoising, the noise variance can be estimated robustly using the Haar wavelet representation. The noise standard deviation is estimated as $\tilde{\sigma} = \frac{M_x}{0.6745}$, wherein $M_x$ is the median of the absolute of the wavelet coefficients of the image $x$, and one level of decomposition suffices. The details are presented in Appendix F. We refer to the wavelet-based noise estimation for DG* guidance as WANDA.

Table 1 presents various evaluation metrics, when sampling using WANDA, compared against the baseline LDM, and LDM+DG* approaches. Figure 4 presents the images generated by the proposed approach. WANDA achieves comparable performance, while running fewer sampling steps than the baseline approaches. The key takeaway from these results is that the closed-form discriminator guidance (DG*) approach can be applied over any existing diffusion model at no additional training cost, with a marginal increase in memory, to store the centres of the kernel-based discriminator expansion. These are akin to a *non-trainable set of discriminator guidance parameters*.

Table 2: Ablations of the proposed closed-form discriminator guidance for LDM (LDM+DG$^*$) on the CelebA-HQ dataset. LDM+DG$^*$ with an exponential decay of the discriminator guidance weight performs the best, in terms of the Clean-FID, CLIP-FID and KID metrics. We also observe that fewer DG$^*$ steps leads to superior performance. Essentially, the DG$^*$ steps provide good initialization to the subsequent LDM sampling steps. † denotes that the metric is computed via Torch Fidelity (Obukhov et al., 2020), and ‡ denotes that the metric is computed via Clean-FID (Parmar et al., 2021).

| Method | Clean-FID‡ | CLIP-FID‡ | KID‡ | Precision† | Recall† |
|---|---|---|---|---|---|
| LDM+DG$_\theta$ (Kim et al., 2023) | 21.44 | 7.08 | $2.191 \times 10^{-2}$ | 0.5465 | 0.4420 |
| LDM+DG$^*$ (linear $w_{dg,t}$) | 31.68 | 10.99 | $3.125 \times 10^{-2}$ | 0.3602 | 0.5787 |
| LDM+DG$^*$ ($T_D = 50$) | **20.49** | **6.48** | $\mathbf{2.041 \times 10^{-2}}$ | **0.4932** | 0.4806 |
| WANDA ($T_D = 50$) | 22.76 | 7.98 | $2.270 \times 10^{-2}$ | 0.4570 | 0.4990 |
| WANDA ($T_D = 100$) | 28.79 | 10.02 | $2.845 \times 10^{-2}$ | 0.3574 | **0.5413** |
| WANDA ($T_D = 200$) | 37.83 | 12.64 | $3.688 \times 10^{-2}$ | 0.2030 | 0.5330 |

## 5.1 ABLATIONS

To better understand the effect of the time-shifted diffusion, and the effect of the closed-form discriminator on generation performance, we perform ablations on the CelebA-HQ dataset. We ablate on the choice of the decay parameter, $w_{dg,t}$ considering linear, exponential, and step-wise decay profiles. For the linear vs. exponential decay setting, considering LDM+DG$^*$, we found that exponential decay with $w_{dg,T} = 1$. gave superior performance. Performance comparisons with a linear decay and $w_{dg,T} = 0.1$, which leads to a comparable values for the weight as sampling completes (*i.e.,* $w_{dg,t}$ approach similar values in both cases, as $t \to 0$. We compare the performance of the LDM+DG$^*$ against a model wherein the discriminator is trained akin to the procedure described by (Kim et al., 2023). We employ a noise-embedded U-Net encoder with sigmoid activation as the discriminator that learns to classify the real and fake samples across all noise levels. The model is trained using the binary cross-entropy (BCE) loss. From Table 2, we observe that the LDM model with the trained discriminator (LDM+D$_\theta$) either outperforms or is on par with the baselines. However, the trainable discriminator requires significantly more compute. On the contrary, the proposed LDM-DG$^*$ can be applied in a *plug-and-play* manner, with no additional training costs, and achieves a superior performance in terms of FID and KID metrics, compared to the LDM+D$_\theta$ sampler. Lastly, we ablate on the time-step shifting algorithm with DG$^*$. We consider a sampling strategy wherein the discriminator is applied for the first $T_D$ steps, and subsequently, transitioned to the base LDM sampler. We ablate over $T_D \in \{50, 100, 200\}$. From the metrics shown in Table 2, we observe that fewer discriminator steps lead to a superior performance. Emprically, this was found to be $T_D^* \approx 50$. We observe that in the WANDA setting, there is a stark jump initially, of about 10 or so steps via the noise-variance-based time-step shifting. These observations show that DG$^*$ can be viewed as providing a quick high-quality transition at the initial iterations.

## 6 CONCLUSION

In this paper, we considered the setting of discriminator guidance in diffusion models, and developed strong theoretical links to GAN generator optimization. We showed, using variational calculus, that the optimality of IPM-GAN generator corresponds to a smoothed score-matching condition. Based on this novel insight, we developed a kernel-based closed-form discriminator guidance framework that can be applied in a *plug-and-plan* fashion to any existing diffusion model. We demonstrated the feasibility of this approach by means of experimentation with a discriminator-only Langevin sampler. Subsequently, we showed that closed-form discriminator guidance, applied to EDMs and DDIMs, results in superior image quality at no additional training cost. We also demonstrated an extension to accelerated DDIM by means of a time-step-shifted diffusion model considering a novel wavelet-based noise variance estimate. While the presented experiments demonstrate the versatility of the closed-form discriminator guidance approach, exploring applications to other state-of-the-art diffusion models, or leveraging other techniques from GAN training for accelerating diffusion, are promising directions for future research.

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

# Appendix

## Table of Contents

## A  COMPUTATIONAL RESOURCES

All experiments were carried out using TensorFlow 2.0 (Abadi et al., 2016) and PyTorch (Paszke et al., 2019) backend. Experiments on NCSN, EDM, and LDM were built atop publicly available implementations (URL: `https://github.com/Xemnas0/NCSN-TF2.0`, `https://github.com/NVlabs/edm`, and `https://github.com/CompVis/latent-diffusion`, respectively). Experiments were performed on SuperMicro workstations with 256 GB of system RAM comprising two NVIDIA GTX 3090 GPUs, each having 24 GB VRAM, and NVIDIA RTX A6000 with 8 GPUs.

## B  CODE REPOSITORY AND ANIMATIONS

The TF 2.0 (Abadi et al., 2016) based source code for implementing discriminator-guided Langevin diffusion and LDM-based experiments have been included as part of the *Supplementary Material* and will be made accessible on GitHub upon paper acceptance. Additionally, we have also provided animations corresponding to the *Shape Morphing* experiments presented in Figure 7, and the images generated in Figures 8–10, Figure 14 and Figure 4. Full-resolution versions of images presented in the paper will also be made accessible in the GitHub Repository.

## C MATHEMATICAL PRELIMINARIES

Consider a vector $\boldsymbol{z} = [z_1, z_2, \ldots, z_n]^{\mathrm{T}} \in \mathbb{R}^n$ and the generator $G : \mathbb{R}^n \rightarrow \mathbb{R}^n$, *i.e.,*, $G(\boldsymbol{z}) = [G_1(\boldsymbol{z}), G_2(\boldsymbol{z}), \ldots; G_n(\boldsymbol{z})]^{\mathrm{T}}$, where $G_i(\boldsymbol{z})$ denotes the $i^{th}$ entry of $G$. The notation $\nabla_{\boldsymbol{z}} G(\boldsymbol{z})$ represents the gradient matrix of the generator, with entries consisting of the partial derivatives of the entries of $G$ with respect to the entries of $\boldsymbol{z}$ and is given by

$$\nabla_{\boldsymbol{z}} G(\boldsymbol{z}) = \begin{bmatrix} \frac{\partial G_1}{\partial z_1} & \frac{\partial G_2}{\partial z_1} & \cdots & \frac{\partial G_n}{\partial z_1} \\ \frac{\partial G_1}{\partial z_2} & \frac{\partial G_2}{\partial z_2} & \cdots & \frac{\partial G_n}{\partial z_2} \\ \vdots & \vdots & \ddots & \vdots \\ \frac{\partial G_1}{\partial z_n} & \frac{\partial G_2}{\partial z_n} & \cdots & \frac{\partial G_n}{\partial z_n} \end{bmatrix}.$$

The Jacobian J *measures* the transformation that the function imposes locally near the point of evaluation and is given as the transpose of the gradient matrix, *i.e.,* $\mathrm{J}_G(\boldsymbol{z}) = (\nabla_{\boldsymbol{z}} G(\boldsymbol{z}))^{\mathrm{T}}$.

***Calculus of Variations***: Our analysis centers around deriving the optimal generator in the functional sense, leveraging the *Fundamental Lemma of the Calculus of Variations* (Goldstine, 1980; Ferguson, 2004). Consider an integral cost $\mathcal{L}$, to be optimized over a function $h$:

$$\mathcal{L}(h, h') = \int_{\mathcal{X}} \mathcal{F}(\boldsymbol{x}, h(\boldsymbol{x}), h'(\boldsymbol{x})) \; \mathrm{d}\boldsymbol{x}, \tag{9}$$

where $h$ is assumed to be continuously differentiable or at least possess a piecewise-smooth derivative $h'(\boldsymbol{x})$ for all $\boldsymbol{x} \in \mathcal{X}$. If $h^*(\boldsymbol{x})$ denotes the optimum, The *first variation* of $\mathcal{L}$, evaluated at $h^*$, is defined as the derivative $\delta\mathcal{L}(h^*; \eta) = \frac{\partial \mathcal{L}_\epsilon(h^*)}{\partial \epsilon}$ evaluated at $\epsilon = 0$, where $\mathcal{L}_\epsilon(h^*)$ denotes an $\epsilon$-perturbation of the argument $h$ about the optimum $h^*$, given by

$$\mathcal{L}_{h,\epsilon}(\epsilon) = \mathcal{L}(h^*(\boldsymbol{x}) + \epsilon\,\eta(\boldsymbol{x}), h^{*'}(\boldsymbol{x}) + \epsilon\,\eta'(\boldsymbol{x}))$$

where, in turn, $\eta(\boldsymbol{x})$ is a family of *perturbations* that are compactly supported, infinitely differentiable functions, and vanishing on the boundary of $\mathcal{X}$. Then, the optimizer of the cost $\mathcal{L}$ satisfies the following first-order condition:

$$\left. \frac{\partial \mathcal{L}_{h,\epsilon}(\epsilon)}{\partial \epsilon} \right|_{\epsilon=0} = 0$$

Another core concept in deriving functional optima is the *Fundamental Lemma of Calculus of Variations*, which states that, if a function $g(\boldsymbol{x})$ satisfies the condition

$$\int_{\mathcal{X}} g(\boldsymbol{x})\,\eta(\boldsymbol{x})\,\mathrm{d}\boldsymbol{x} = 0$$

for all compactly supported, infinitely differentiable functions $\eta(\boldsymbol{x})$, then $g$ must be identically zero almost everywhere in $\mathcal{X}$. Together, these results are used to derive the condition that the optimal generator transformation satisfies, within various GAN formulations.

# D   OPTIMALITY OF IPM-BASED GANS

We now derive the proofs for theorems presented in the context of IPM GANs. The $f$-GAN counterparts are provided in Asokan et al. (2023).

## D.1   OPTIMALITY OF KERNEL-BASED IPM-GANS (PROOFS OF THEOREM 2 AND LEMMA 3)

Mroueh et al. (2018), in the context of SobolevGAN, showed that IPM-GANs with a gradient-based constraint defined with respect to a base density $\mu(\boldsymbol{x})$ results in the optimal discriminator solving the Fokker-Planck partial differential equation (PDE), given by:

$$\text{div.} \left( \mu \, \nabla D \right) \big|_{D=D_t^*(\boldsymbol{x})} = \text{c} \left( p_d(\boldsymbol{x}) - p_{t-1}(\boldsymbol{x}) \right),$$

where $\text{div}$ denotes the divergence operator and $\text{c}$ is a constant. Considering a uniform base measure, Asokan & Seelamantula (2023b) showed that the optimization results in a Poisson differential equation, while in the case of higher-order gradient penalties (Adler & Lunz, 2018; Asokan & Seelamantula, 2023a), the optimal discriminator is the solution to an iterated Laplacian equation, and generalizes the SobolevGAN formulation. The optimal discriminator that satisfies the iterated-Laplacian operator was shown to be (Asokan & Seelamantula, 2023a):

$$D_t^*(\boldsymbol{x}) = \mathfrak{C}_\kappa \left( (p_{t-1} - p_d) * \kappa \right)(\boldsymbol{x}),$$

where $\mathfrak{C}_\kappa = \frac{(-1)^{m+1}\varrho}{2\lambda}$ and $\varrho$ are positive constants, and the kernel $\kappa$ is the Green's function associated with the differential operator. In Poly-WGAN, the kernel corresponds to the family of polyharmonic splines, given by

$$\kappa(\boldsymbol{x}) = \begin{cases} \|\boldsymbol{x}\|^k & \text{if } k < 0 \text{ or } n \text{ is odd,} \\ \|\boldsymbol{x}\|^k \ln(\|\boldsymbol{x}\|) & \text{if } k \geq 0 \text{ and } n \text{ is even,} \end{cases}$$

where in turn, $k = 2m - n$. The above was also shown to be an $m^{th}$-order generalization to the Plummer kernel considered in Coulomb GANs (Unterthiner et al., 2018). Given the optimal discriminator, consider the generator optimization. Only the terms involving $G(\boldsymbol{z})$ influence the alternating optimization in practice, and the other terms can be neglected. Then, the cost is given by:

$$\mathcal{L}_G^\kappa(G; D_t^*, G_{t-1}) = - \underset{\boldsymbol{z} \sim p_{\boldsymbol{z}}}{\mathbb{E}} \left[ D_t^* \left( G(\boldsymbol{z}) \right) \right] = - \int_{\mathcal{Z}} D_t^*(G(\boldsymbol{z})) \, p_{\boldsymbol{z}}(\boldsymbol{z}) \, \mathrm{d}\boldsymbol{z}$$

Let $\mathcal{L}_{G,i,\epsilon}$ denote the loss considering an $\epsilon$ perturbation of the $i^{th}$ entry about the optimum, given by:

$$G_{t,i,\epsilon}^*(\boldsymbol{z}) = [G_{1,t}^*(\boldsymbol{z}), G_{2,t}^*(\boldsymbol{z}), \, \ldots, \, G_{i,t}^*(\boldsymbol{z}) + \epsilon\eta(\boldsymbol{z}), \, \ldots, \, G_{n,t}^*(\boldsymbol{z})]^{\mathrm{T}},$$

where $\eta(\boldsymbol{z})$ is drawn from a family of compactly supported, infinitely differentiable functions. The loss can then be written as a function of $\epsilon$. Consider the perturbed optimal generator $G_{t,i,\epsilon}^*(\boldsymbol{z})$, and the corresponding cost $\mathcal{L}_{G,i,\epsilon}(\epsilon)$. Substituting for $D_t^*$ and expanding the convolution integral yields:

$$\mathcal{L}_{G,i,\epsilon}^\kappa(\epsilon) = - \int_{\mathcal{Z}} \mathfrak{C}_\kappa \, p_{\boldsymbol{z}}(\boldsymbol{z}) \int_{\mathcal{Y}} \left( p_{t-1}(G_{t,i,\epsilon}^*(\boldsymbol{z}) - \boldsymbol{y}) - p_d(G_{t,i,\epsilon}^*(\boldsymbol{z}) - \boldsymbol{y}) \right) \kappa(\boldsymbol{y}) \, \mathrm{d}\boldsymbol{y} \, \mathrm{d}\boldsymbol{z}, \quad (10)$$

where $\mathcal{Y}$ is the union of the supports of $p_d$ and $p_{t-1}$ when they are overlapping, and the convex hull of their supports when non-overlapping. Differentiating the above with respect to $\epsilon$ and setting it to zero at $\epsilon = 0$ gives:

$$\frac{\partial \mathcal{L}_{G,i,\epsilon}^\kappa(\epsilon)}{\partial \epsilon} \bigg|_{\epsilon=0} = - \int_{\mathcal{Z}} \mathfrak{C}_\kappa \, p_{\boldsymbol{z}}(\boldsymbol{z}) \int_{\mathcal{Y}} \left( p_{t-1}(\boldsymbol{y}) - p_d(\boldsymbol{y}) \right) \frac{\partial \kappa(G_{t,i,\epsilon}^*(\boldsymbol{z}) - \boldsymbol{y})}{\partial \epsilon} \bigg|_{\epsilon=0} \mathrm{d}\boldsymbol{y} \, \mathrm{d}\boldsymbol{z}$$

$$= - \int_{\mathcal{Z}} \mathfrak{C}_\kappa \, p_{\boldsymbol{z}}(\boldsymbol{z}) \int_{\mathcal{Y}} \left( p_{t-1}(\boldsymbol{y}) - p_d(\boldsymbol{y}) \right) \frac{\partial \kappa(\boldsymbol{w})}{\partial x_i} \bigg|_{\boldsymbol{w}=G_t^*(\boldsymbol{z})-\boldsymbol{y}} \frac{\partial [G_{t,i,\epsilon}^*(\boldsymbol{z})]_i}{\partial \epsilon} \mathrm{d}\boldsymbol{y} \, \mathrm{d}\boldsymbol{z}$$

$$= - \int_{\mathcal{Z}} \mathfrak{C}_\kappa \, p_{\boldsymbol{z}}(\boldsymbol{z}) \int_{\mathcal{Y}} \left( p_{t-1}(\boldsymbol{y}) - p_d(\boldsymbol{y}) \right) \frac{\partial \kappa(\boldsymbol{w})}{\partial w_i} \bigg|_{\boldsymbol{w}=G_t^*(\boldsymbol{z})-\boldsymbol{y}} \eta(\boldsymbol{z}) \, \mathrm{d}\boldsymbol{y} \, \mathrm{d}\boldsymbol{z} = 0.$$

The inner integral represents a convolution, given by

$$\frac{\partial \mathcal{L}_{G,i,\epsilon}^{\kappa}(\epsilon)}{\partial \epsilon}\bigg|_{\epsilon=0} = -\mathfrak{C}_{\kappa} \int_{\mathcal{Z}} \left( (p_{t-1} - p_d) * \kappa_i' \right)(\boldsymbol{x}) \bigg|_{\boldsymbol{x}=G_t^*(\boldsymbol{z})} p_{\boldsymbol{z}}(\boldsymbol{z})\eta(\boldsymbol{z}) \, \mathrm{d}\boldsymbol{z} = 0,$$

where $\kappa_i'$ is the partial derivative of the kernel $\kappa$ with respect to its $i^{th}$ entry. From the *Fundamental Lemma of Calculus of Variations*, we have

$$\mathfrak{C}_{\kappa} \left( (p_{t-1} - p_d) * \kappa_i' \right)(\boldsymbol{x}) \bigg|_{\boldsymbol{x}=G_t^*(\boldsymbol{z})} = 0, \qquad \forall \; \boldsymbol{z} \in \mathcal{Z}. \tag{11}$$

Since the above holds for all $i$, the above can be written compactly as

$$\mathfrak{C}_{\kappa} \left( (p_{t-1} - p_d) * \nabla_{\boldsymbol{x}} \kappa \right)(\boldsymbol{x}) \bigg|_{\boldsymbol{x}=G_t^*(\boldsymbol{z})} = \boldsymbol{0}, \qquad \forall \; \boldsymbol{z} \in \mathcal{Z},$$

where the convolution between a scalar- and vector-valued function is carried out element-wise. This completes the proof of Lemma 3. Table 3 lists a few common kernels used across GAN variants and their corresponding gradient vectors.

***Proof of Theorem 2***: An alternative approach to solving the aforementioned optimization, is to leverage the properties of convolution in Equation (11). Consider the convolution integral:

$$\left( (p_{t-1} - p_d) * \kappa_i' \right)(\boldsymbol{w}) = \int_{\mathcal{Y}} (p_{t-1}(\boldsymbol{y}) - p_d(\boldsymbol{y})) \frac{\partial \kappa(\boldsymbol{w})}{\partial w_i} \, \mathrm{d}\boldsymbol{y} \bigg|_{\boldsymbol{w}=G_t^*(\boldsymbol{z})-\boldsymbol{y}}$$

$$= \frac{\partial}{\partial w_i} \left( \int_{\mathcal{Y}} (p_{t-1}(\boldsymbol{y}) - p_d(\boldsymbol{y})) \kappa(\boldsymbol{w}) \, \mathrm{d}\boldsymbol{y} \right) \bigg|_{\boldsymbol{w}=G_t^*(\boldsymbol{z})-\boldsymbol{y}} = 0, \forall \; \boldsymbol{z} \in \mathcal{Z}.$$

From the property of convolutions, we have:

$$\left( (p_{t-1} - p_d) * \kappa_i' \right)(\boldsymbol{w}) = \frac{\partial}{\partial w_i} \left( \int_{\mathcal{Y}} (p_{t-1}(\boldsymbol{w}) - p_d(\boldsymbol{w})) \kappa(\boldsymbol{y}) \, \mathrm{d}\boldsymbol{y} \right) \bigg|_{\boldsymbol{w}=G_t^*(\boldsymbol{z})-\boldsymbol{y}}$$

$$= \left( \int_{\mathcal{Y}} \left( \frac{\partial p_{t-1}(\boldsymbol{w})}{\partial w_i} - \frac{\partial p_d(\boldsymbol{w})}{\partial w_i} \right) \kappa(\boldsymbol{y}) \, \mathrm{d}\boldsymbol{y} \right) \bigg|_{\boldsymbol{w}=G_t^*(\boldsymbol{z})-\boldsymbol{y}} = 0, \forall \; \boldsymbol{z} \in \mathcal{Z}.$$

Using the identity $\dfrac{\partial p(\boldsymbol{w})}{\partial w_i} = p(\boldsymbol{w}) \dfrac{\partial \ln p(\boldsymbol{w})}{\partial w_i}$, we obtain:

$$\left( (p_{t-1} - p_d) * \kappa_i' \right)(\boldsymbol{w}) = \left( \int_{\mathcal{Y}} \left( \frac{\partial p_{t-1}(\boldsymbol{w})}{\partial w_i} - \frac{\partial p_d(\boldsymbol{w})}{\partial w_i} \right) \kappa(\boldsymbol{y}) \, \mathrm{d}\boldsymbol{y} \right) \bigg|_{\boldsymbol{w}=G_t^*(\boldsymbol{z})-\boldsymbol{y}}$$

$$= \left( \int_{\mathcal{Y}} \left( p_{t-1}(\boldsymbol{y}) \frac{\partial \ln(p_{t-1}(\boldsymbol{y}))}{\partial y_i} - p_d(\boldsymbol{y}) \frac{\partial \ln(p_d(\boldsymbol{y}))}{\partial y_i} \right) \kappa(\boldsymbol{x}-\boldsymbol{y}) \, \mathrm{d}\boldsymbol{y} \right) = 0,$$

for all $\boldsymbol{z} \in \mathcal{Z}$ and $\boldsymbol{x} = G_t^*(\boldsymbol{z})$. Rewriting the integrals as expectations yields

$$\mathbb{E}_{\boldsymbol{y} \sim p_{t-1}} \left[ \frac{\partial \ln(p_{t-1}(\boldsymbol{y}))}{\partial y_i} \kappa(G_t^*(\boldsymbol{z}) - \boldsymbol{y}) \right] - \mathbb{E}_{\boldsymbol{y} \sim p_d} \left[ \frac{\partial \ln(p_d(\boldsymbol{y}))}{\partial y_i} \kappa(G_t^*(\boldsymbol{z}) - \boldsymbol{y}) \right] = 0, \qquad \forall \; \boldsymbol{z} \in \mathcal{Z}.$$

Stacking the above, for all $i$, as a vector, we obtain:

$$\mathbb{E}_{\boldsymbol{y} \sim p_{t-1}} \left[ \nabla_{\boldsymbol{y}} \ln(p_{t-1}(\boldsymbol{y})) \kappa(G_t^*(\boldsymbol{z}) - \boldsymbol{y}) \right] - \mathbb{E}_{\boldsymbol{y} \sim p_d} \left[ \nabla_{\boldsymbol{y}} \ln(p_d(\boldsymbol{y})) \kappa(G_t^*(\boldsymbol{z}) - \boldsymbol{y}) \right] = \boldsymbol{0}, \qquad \forall \; \boldsymbol{z} \in \mathcal{Z}.$$

This completes the proof of Theorem 2.

***Explaining Denoising Diffusion GANs***: To derive a general solution to IPM-GANs (both network-based, or otherwise), consider the discriminator given at iteration $t$, $D_t(\boldsymbol{x})$. Then, the generator optimization is given by:

$$\mathcal{L}_G^{IPM}(G; D_t, G_{t-1}) = - \mathbb{E}_{\boldsymbol{z} \sim p_{\boldsymbol{z}}} \left[ D_t\left( G(\boldsymbol{z}) \right) \right] = - \int_{\mathcal{Z}} D_t(G(\boldsymbol{z})) \, p_{\boldsymbol{z}}(\boldsymbol{z}) \, \mathrm{d}\boldsymbol{z}$$

Table 3: Standard kernels considered in the GAN literature and their associated gradient fields.

| Kernel | $\kappa(\boldsymbol{x})$ | Gradient $\nabla_{\boldsymbol{x}}\kappa(\boldsymbol{x})$ |
|---|---|---|
| Radial basis function Gaussian (RBFG) ($\sigma > 0$) | $\exp\left(-\frac{1}{\sigma^2}\|\boldsymbol{x}\|^2\right)$ | $-\frac{1}{\sigma^2}\boldsymbol{x}\exp\left(-\frac{1}{\sigma^2}\|\boldsymbol{x}\|^2\right)$ |
| Mixture of Gaussians (MoG) ($\{\sigma_i > 0\}_{i=1}^{\ell}$) | $\sum_{\sigma_i}\exp\left(-\frac{1}{\sigma_i^2}\|\boldsymbol{x}\|^2\right)$ | $-\boldsymbol{x}\left(\sum_{\sigma_i}\frac{1}{\sigma_i^2}\exp\left(-\frac{1}{\sigma_i^2}\|\boldsymbol{x}\|^2\right)\right)$ |
| Inverse multi-quadric (IMQ) ($c > 0$) | $(\|\boldsymbol{x}\|^2 + c)^{-\frac{1}{2}}$ | $-\frac{1}{2}\boldsymbol{x}(\|\boldsymbol{x}\|^2 + c)^{-\frac{3}{2}}$ |
| Polyharmonic spline (PHS) ($k < 0$ or $n$ is odd) | $\|\boldsymbol{x}\|^k$ | $(k-2)\boldsymbol{x}\|\boldsymbol{x}\|^{k-2}$ |
| Polyharmonic spline (PHS) ($k \geq 0$ and $n$ is even) | $\|\boldsymbol{x}\|^k\ln(\|\boldsymbol{x}\|)$ | $\boldsymbol{x}\|\boldsymbol{x}\|^{k-2}\left((k-2)\ln(\|\boldsymbol{x}\|) + 1\right)$ |

The loss defined about the perturbed optimal generator is then given by:

$$\mathcal{L}_{G,i,\epsilon}^{IPM}(\epsilon) = -\int_{\mathcal{Z}} D_t(G_{t,i,\epsilon}^*(\boldsymbol{z}))\,\mathrm{d}\boldsymbol{z}$$

$$\Rightarrow \quad \left.\frac{\partial \mathcal{L}_{G,i,\epsilon}^{IPM}(\epsilon)}{\partial \epsilon}\right|_{\epsilon=0} = \int_{\mathcal{Z}} \left.\frac{\partial D_t(\boldsymbol{x})}{\partial x_i}\right|_{\boldsymbol{x}=G_t^*(\boldsymbol{z})} p_{\boldsymbol{z}}(\boldsymbol{z})\eta(\boldsymbol{z})\,\mathrm{d}\boldsymbol{z} = 0.$$

A similar approach, as in the case of kernel-based IPM-GANs, to simplifying the above for all $i$, results in the following optimality condition:

$$\nabla_{\boldsymbol{x}} D_t(\boldsymbol{x})\big|_{\boldsymbol{x}=G_t^*(\boldsymbol{z})} = \boldsymbol{0}, \quad \forall\, \boldsymbol{z} \in p_{\boldsymbol{z}}.$$

While the above condition is essentially the optimality condition for gradient-descent over the discriminator in the context of gradient-descent-based training of GANs, it can be used to explain the optimality of GAN based diffusion models such as Denoising Diffusion GANs (DDGAN, Xiao et al. (2022)). In DDGAN, a GAN is trained to approximate the reverse diffusion process, with time-embedding-conditioned discriminator and generator networks. While the approach results in superior sampling speeds as one only needs to sample from the sequence of generators, the underlying transformations that the generated images undergo, can be seen as the flow through the gradient field of the time-dependent discriminator as obtained above.

***Convergence of the Generator Distribution:*** Given the optimal discriminator $D^*$, Asokan & Seelamantula (2023a) showed that the generator distribution converges to the desired data distribution. For the sake of completeness, we summarize the Theorem here:

**Theorem 4.** *(Asokan & Seelamantula, 2023a) (**Optimal generator density**): Consider the minimization of the generator loss $\mathcal{L}_G$. The optimal generator density is given by $p_g^*(\boldsymbol{x}) = p_d(\boldsymbol{x})$, $\forall\, \boldsymbol{x} \in \mathcal{X}$. The optimal Lagrange multipliers are*

$$\lambda_p^* \in \mathbb{R} \quad and \quad \mu_p^*(\boldsymbol{x}) = \begin{cases} 0, & \forall\, \boldsymbol{x}\, :\, p_d(\boldsymbol{x}) > 0, \\ Q(\boldsymbol{x}) \in \mathcal{P}_{m-1}^n(\boldsymbol{x}), & \forall\, \boldsymbol{x}\, :\, p_d(\boldsymbol{x}) = 0, \end{cases}$$

*respectively, where $Q(\boldsymbol{x})$ is a non-positive polynomial of degree $m - 1$, i.e., $Q(\boldsymbol{x}) \leq 0\, \forall\, \boldsymbol{x}$, such that $p_d(\boldsymbol{x}) = 0$. The solution is valid for all choices of the homogeneous component $P(\boldsymbol{x}) \in \mathcal{P}_{m-1}^n(\boldsymbol{x})$ in the optimal discriminator.*

*Proof.* As the cost function involves convolution terms, the Euler-Lagrange condition cannot be applied readily, and the optimum must be derived using the *Fundamental Lemma of Calculus of Variations* Gel'fand & Fomin (1964), as presented by Asokan & Seelamantula (2023a). We recall a summary of the proof here for completeness. Consider the Lagrangian of the generator loss $\mathcal{L}_G$. Enforcing the first-order necessary conditions for a minimizer of the cost yields the following equation that the optimum solution $p_g^*(\boldsymbol{x})$ satisfies the equation $p_g^*(\boldsymbol{x}) = p_d(\boldsymbol{x}) + \left(\frac{\lambda_d^*}{\xi}\right)\Delta^m\mu_p^*(\boldsymbol{x})$. It is clear from the above solution that the optimum, $p_g^*(\boldsymbol{x})$, does not depend on the choice of the homogeneous component $P(\boldsymbol{x})$ in the optimal discriminator. The optimal Lagrange multipliers can be determined through dual optimization and enforcing the complementary slackness condition to obtain the result in above Theorem. $\square$

## D.2  Sample Estimate of the Discriminator Gradient

The proof follows closely the approach used in  Asokan & Seelamantula (2023a).  Consider the optimality condition along a given dimension $i$. We have:

$$\mathfrak{C}_\kappa \left( (p_{t-1} - p_d) * \kappa_i' \right)(\boldsymbol{x}) \Big|_{\boldsymbol{x} = G_t^*(\boldsymbol{z})} = 0, \qquad \forall \ \boldsymbol{z} \in \mathcal{Z}.$$

Expanding the convolution integral yields

$$\mathfrak{C}_\kappa \int_{\mathcal{Y}} (p_{t-1}(\boldsymbol{y}) - p_d(\boldsymbol{y})) \, \kappa_i'(G_t^*(\boldsymbol{z}) - \boldsymbol{y}) \, \mathrm{d}\boldsymbol{y} = 0, \qquad \forall \ \boldsymbol{z} \in \mathcal{Z}$$

$$\Rightarrow \int_{\mathcal{Y}} p_{t-1}(\boldsymbol{y}) \, \kappa_i'(G_t^*(\boldsymbol{z}) - \boldsymbol{y}) \, \mathrm{d}\boldsymbol{y} - \int_{\mathcal{Y}} p_d(\boldsymbol{y}) \, \kappa_i'(G_t^*(\boldsymbol{z}) - \boldsymbol{y}) \, \mathrm{d}\boldsymbol{y} = 0, \qquad \forall \ \boldsymbol{z} \in \mathcal{Z}$$

$$\Rightarrow \mathop{\mathbb{E}}_{\boldsymbol{y} \sim p_{t-1}} \left[ \kappa_i'(G_t^*(\boldsymbol{z}) - \boldsymbol{y}) \right] - \mathop{\mathbb{E}}_{\boldsymbol{y} \sim p_d} \left[ \kappa_i'(G_t^*(\boldsymbol{z}) - \boldsymbol{y}) \right] = 0, \qquad \forall \ \boldsymbol{z} \in \mathcal{Z}.$$

Replacing the expectations with their sample estimates yields

$$\sum_{\boldsymbol{y}_\ell \sim p_{t-1}} \kappa_i'(G_t^*(\boldsymbol{z}) - \boldsymbol{y}_\ell) = \sum_{\boldsymbol{y}_\ell \sim p_d} \kappa_i'(G_t^*(\boldsymbol{z}) - \boldsymbol{y}_\ell), \qquad \forall \ \boldsymbol{z} \in \mathcal{Z}.$$

Evaluating the above at a sample level, for $G_t^*(\boldsymbol{z}_t) = \boldsymbol{x}_t$, and stacking for all $i$, we get the desired $N$-sample estimate of the discriminator gradient for the closed-form discriminator:

$$\nabla_{\boldsymbol{x}} D_t^*(\boldsymbol{x}_t) = \mathfrak{C}_k' \sum_{\boldsymbol{g}^j \sim \{\boldsymbol{x}_{t-1}\}} \nabla_{\boldsymbol{x}} \kappa(\boldsymbol{x}_t - \boldsymbol{g}^j) - \mathfrak{C}_k' \sum_{\boldsymbol{d}^i \sim p_d} \nabla_{\boldsymbol{x}} \kappa(\boldsymbol{x}_t - \boldsymbol{d}^i). \qquad (12)$$

## D.3  Convergence of Discriminator-guided Langevin Diffusion

An in-depth analysis of the convergence of discriminator-guided Langevin diffusion from the perspective of stochastic differential equations (SDEs) is outside the scope of this paper. However, (Lunz et al., 2018), in the context of adversarial regularization for inverse problems, have extensively analyzed the following iterative algorithm:

$$\boldsymbol{x}_{t+1} = \boldsymbol{x}_t - \eta \nabla_{\boldsymbol{x}} D_{t,\theta}^*(\boldsymbol{x}),$$

where $\eta$ is the learning rate, and $D_{t,\theta}^*(\boldsymbol{x})$ denotes the optimal discriminator at time $t$ parameterized by $\theta$. In particular, they show that (Lunz et al. (2018), Theorem 1):

$$\frac{\partial}{\partial \eta} \mathcal{W}(p_d, p_t) = - \mathop{\mathbb{E}}_{\boldsymbol{x} \sim p_{t-1}} \left[ \|\nabla_{\boldsymbol{x}} D_{t,\theta}^*(\boldsymbol{x})\|_2^2 \right],$$

where $\mathcal{W}$ denotes the Wasserstein-1 or Earthmover's distance. This shows that, the updated distribution $p_t$ is closer in Wasserstein distance to the target distribution $p_d$, in comparison to $p_{t-1}$. For functions with $\|\nabla_{\boldsymbol{x}} D_{t,\theta}^*(\boldsymbol{x})\| = 1$, which is the condition under which the gradient-regularized GANs have been optimized, we have the decay $\frac{\partial}{\partial \eta} \mathcal{W}(p_d, p_t) = -1$. While we consider the updates

$$\boldsymbol{x}_{t+1} = \boldsymbol{x}_t - \alpha_t \nabla_{\boldsymbol{x}} D_t^*(\boldsymbol{x}_t) + \gamma_t \boldsymbol{z}_t$$

in discriminator-guided Langevin diffusion, we will show, experimentally, that the update scheme $\boldsymbol{x}_{t+1} = \boldsymbol{x}_t - \alpha_0 \nabla_{\boldsymbol{x}} D_t^*(\boldsymbol{x}_t)$ indeed performs the best, on image datasets (cf. Appendix E).

# E ADDITIONAL EXPERIMENTAL RESULTS ON DISCRIMINATOR-GUIDED LANGEVIN SAMPLING

We present additional experimental results on generating 2-D shapes, and images using the discriminator-guided Langevin sampler.

## E.1 ADDITIONAL RESULTS ON SYNTHETIC DATA LEARNING

On the 2-D learning task, we present additional combinations on the *shape morphing experiment*.

***Training Parameters***: All samplers are implemented using TensorFlow (Abadi et al., 2016) library. The discriminator gradient is built as a custom radial basis function network, whose weights and centers are assigned at each iteration. At $t = 0$, the centers $\boldsymbol{g}^j \sim p_{t-1}$ are sampled from the unit Gaussian, *i.e.*, $p_{-1} = \mathcal{N}(\boldsymbol{0}, \mathbb{I})$. In subsequent iterations, the batch of samples from time instant $t-1$ serve as the centers for $D_t^*$. Based on experiments presented in Appendix E.2, we set $\gamma_t = 0$ and $\alpha_t = 1 \ \forall \ t$. The input and target distributions are created following the approach presented by (Mroueh & Rigotti, 2020). Figure 5 shows the supports of the input/output distributions (black denotes the support). For grayscale images, the support corresponds to regions with pixel intensities below the threshold of 128.

***Experimental Results***: We consider the *Heart* and *Cat* shapes as the target, while considering various input shapes, corresponding to varying levels of difficulty in matching the target distribution. In the case of learning the *Heart* shape, for input shapes that do not contain *gaps/holes*, the convergence is relatively fast, and shape matching occurs in about 100 to 250 iterations. For more challenging input shapes, such as the *Cat* logo, the discriminator-guided Langevin sampler converges in about 500 iterations. This is superior to the reported 800 iterations in the Unbalanced Sobolev descent formulation. The results are similar in the case where the *Cat* image is the target (cf. Figure 7).

## E.2 ADDITIONAL RESULTS ON IMAGE LEARNING

We present ablation experiments on generating images with the discriminator-guided Langevin sampler to determine the choice of $\alpha_t$ and $\gamma_t$ in the update regime. We also provide additional images pertaining to the experiments presented in the *Main Manuscript*.

***Choice of coefficients*** $\alpha_t$ ***and*** $\gamma_t$: For the ablation experiments, we consider MNIST, SVHN, and 64-dimensional CelebA images. Based on the analysis presented in Asokan & Seelamantula (2023a), we consider the kernel-based discriminator with the polyharmonic spline kernel in all subsequent experiments. Recall the update scheme:

$$\boldsymbol{x}_t = \boldsymbol{x}_{t-1} - \alpha_t \nabla_{\boldsymbol{x}} D_t^*(\boldsymbol{x}_t; \ p_{t-1}, p_d) + \gamma_t \boldsymbol{z}_t, \quad \text{where} \quad \boldsymbol{z}_t \sim \mathcal{N}(\boldsymbol{0}, \mathbb{I}).$$

Based on the observations made by Karras et al. (2022), to ascertain the optimal choice of the coefficients, we consider the following scenarios:

- **The ordinary differential equation (ODE) formulation**, wherein the noise perturbations are ignored, giving rise to an ODE that the samples are evolved through. Here $\gamma_t = 0, \ \forall \ t$.
- **The stochastic differential equation (SDE) formulation**, wherein we retain the noise perturbations. Based on the links between score-based approaches and the GANs, we consider the approach presented in noise-conditioned score networks (NCSNv1) (Song & Ermon, 2019), with $\gamma_t = \sqrt{2\alpha_t}$.

Within these two scenarios, we further consider the following cases:

- **Unadjusted Langevin dynamics (ULD)**, wherein $\alpha_t$ is fixed, *i.e.*, $\alpha_t = \alpha_0, \ \forall \ t$.
- **Annealed Langevin dynamics (ALD)**, wherein $\alpha_t$ decays according to a schedule. While various approaches have been proposed for scaling (Song & Ermon, 2019; 2020; Song et al., 2021b; Jolicoeur-Martineau et al., 2021; Karras et al., 2022), we consider the geometric decay considered in NCSNv1 (Song & Ermon, 2019).

For either case, we present results considering $\alpha_0 \in \{100, 10, 1\}$.

Figures 8–10 show the images generated by the discriminator-guided Langevin sampler on MNIST, SVHN and CelebA, respectively, for the various scenarios considered. Across all datasets, we observe that annealing the coefficients results in poor convergence. We attribute this to the fact that the polyharmonic kernel, being a distance function, decays *automatically* as the iterates converge, *i.e.,* as $p_t$ approaches $p_d$. Consequently, the magnitude of the discriminator gradient, in the case when $\alpha_t$ is decays, is too small to significantly move the particles along the discriminator gradient field. Next, we observe that for relatively small $\alpha_0 \leq 10$, the samplers converge to realistic images. When $\alpha_0$ is large, the resulting *gradient explosion* during the initial steps of the sampler results in *mode-collapse* in all scenarios. Thirdly, in choosing $z_t$, the experimental results indicate that the model converges to visually superior images when $z_t = 0$. For the scenarios where $\alpha_t$, the coefficient of $\nabla_x D_t^*$, is kept constant, but the coefficient $\gamma_t$ decays with $t$ as in the baseline setting. When $z_t$ is non-zero, the generated images are noisy. We attribute the convergence of the discriminator-guided Langevin sampler to unique samples even in scenarios when $z_t$ is zero, to the implicit randomness of the centers of the radial basis function kernels introduced by the sample estimates in the discriminator $D_t^*$.

The superior convergence of the proposed approach is further validated by the *iterate convergence* presented in Figure 6. We compare discriminator-guided Langevin sampler, with $\alpha_t = \alpha_0 = 10$, with and without noise perturbations $z_t$, against the base NCSN model, owing to the links to the score-based results derived in ScoreGANs and FloWGANs. We plot $\|x_t - x_{t-1}\|_2^2$ as a function of iteration $t$ for the MNIST learning task. In NCSN, the iterates converge at each noise level, and subsequently, when the noise level drops, the sample quality improved. This is consistent with the observations made by Song & Ermon (2020), who showed that the score network $S_\theta$ implicitly scales its output by the noise variance $\sigma$. The proposed approach, with $z_t = \mathbf{0}$, performs the best.

***Uniqueness of generated images***: As the kernel-based discriminator operates directly on the target data, drawing batches of samples as centers in the RBF interpolator, an obvious question to ask is whether the discriminator-guided Langevin iterations converge to unique samples *not seen in the dataset*. To verify this, we perform a $k$-nearest neighbor analysis, considering $k = 9$ in the experiments. Figures 11– 13 present the top-$k$ neighbors of samples generated by the proposed images from each digit class of MNIST, SVHN, and CelebA datasets. The neighbors are found across all *digit* classes in the case of MNIST and SVHN. It is clear from these results that the proposed approach **does not** memorize the dataset. In the case of SVHN, considering the samples generated from *digit class 5* of *digit class 9*, we observe that the nearest neighbor is from a different class, indicative of the sampler's ability to interpolate between the classes seen as part of discriminator centers during sampling.

***Details on the experiment presented in Section 4 of the Main Manuscript***: Figure 14 presents the images, considering the Langevin sampler with $\alpha_t = \alpha_0 = 10$ with $z_t = 0$. Across all three datasets, we observe that the models converge to nearly realists samples in about $t = 500$ iterations, while subsequent iterations serve to *denoise* the images. Animations pertaining to these iterations are provided as part of the Supplementary Material.

***Images for experiments presented in Section 5 of the Main Manuscript***: Figures 17 and 18 provide additional comparisons between the baseline and proposed LDM variants on the CelebA-HQ and FFHQ datasets, respectively.

***Ablation on the choice of the sampler:*** The proposed discriminator guidance term is orthogonal to baselines such as Lu et al. (2022); Zhou et al. (2024), wherein better ODE solvers are used to accelerate sampling. As such, the closed-form discriminator guidance (+DG*) can be combined with these techniques as well. As a proof of concept, we present an ablation on CelebA-HQ, considering the DPM solver (Lu et al., 2022), with and without +DG*. Table 5 presents the evaluation metrics for this experiment. We observe that including discriminator guidance allows us to further accelerate the sample generation process, with the DPM+DG* sampler achieving comparable performance in $T = 15$ (1 discriminator step with 14 DPM solver steps) steps, as the baseline DPM model with $T = 20$. On the other hand, the DPM+DG* with $T = 20$ outperforms the baseline for the same $T$.

Table 4: Ablations of the proposed closed-form discriminator guidance for DPM Solver (DPM+DG$^*$) on the CelebA-HQ dataset, in terms of the Clean-FID, CLIP-FID and KID metrics. We observe that including discriminator guidance allows us to further accelerate the sample generation process, with the DPM+DG$^*$ sampler achieving comparable performance in $T = 15$ (1 discriminator step with 14 DPM solver steps) steps, as the baseline DPM model with $T = 20$. ‡ denotes that the metric is computed via Clean-FID (Parmar et al., 2021).

| | Method | Clean-FID‡ | CLIP-FID‡ | KID‡ |
|---|---|---|---|---|
| DPM | $T = 20$ | 24.54 | 9.50 | 0.0231 |
| | $T = 15$ | 26.63 | 10.07 | 0.0262 |
| DPM+DG$^*$ | $T = 20,\ T_D = 20,\ w_{dg} = 1.0$ | 24.10 | 9.28 | **0.0230** |
| | $T = 20,\ T_D = 2,\ w_{dg} = 1.0$ | **24.07** | **9.22** | 0.0235 |
| | $T = 20,\ T_D = 2,\ w_{dg} = 0.5$ | 24.67 | 9.28 | 0.0235 |
| | $T = 15,\ T_D = 1,\ w_{dg} = 1.0$ | 24.64 | 9.71 | 0.0233 |
| | $T = 15,\ T_D = 1,\ w_{dg} = 0.5$ | 24.44 | 9.66 | 0.0232 |
| | $T = 10,\ T_D = 1,\ w_{dg} = 1.0$ | 31.82 | 11.48 | 0.0320 |
| | $T = 10,\ T_D = 1,\ w_{dg} = 0.5$ | 31.81 | 11.42 | 0.0328 |

Table 5: Performance evaluation of WANDA, in terms of Clean-FID and CLIP-FID (Parmar et al., 2021) when ablations are carried out on the choice of the cut-off time $T_D$ and guidance weight $w_{dg}$. In general, we observe that, running discriminator guidance for about 10% of the initial iterations, with the guidance weight $w_{dg} \in (0.5, 1)$ leads to the best performance.

| | Method | Clean-FID‡ | CLIP-FID‡ |
|---|---|---|---|
| | Baseline | 12.95 | 3.78 |
| | $T_D = 50,\ w_{dg} = 25$ | 22.85 | 5.48 |
| | $T_D = 50,\ w_{dg} = 20$ | 19.92 | 5.01 |
| | $T_D = 50,\ w_{dg} = 10$ | 15.41 | 4.22 |
| $T = 50$ | $T_D = 10,\ w_{dg} = 10$ | 15.37 | 4.18 |
| | $T_D = 5,\ w_{dg} = 10$ | 14.04 | 4.14 |
| | $T_D = 5,\ w_{dg} = 5$ | 12.79 | 3.90 |
| | $T_D = 5,\ w_{dg} = 2$ | 12.24 | 3.81 |
| | $T_D = 5,\ w_{dg} = 1$ | 12.13 | 3.79 |
| | $T_D = 5,\ w_{dg} = 0.5$ | **12.04** | **3.72** |
| | Baseline | 9.30 | 3.02 |
| | $T_D = 100,\ w_{dg} = 25$ | 15.37 | 4.16 |
| | $T_D = 100,\ w_{dg} = 15$ | 11.93 | 3.51 |
| $T = 100$ | $T_D = 10,\ w_{dg} = 10$ | 10.70 | 3.26 |
| | $T_D = 10,\ w_{dg} = 5$ | 9.88 | 3.11 |
| | $T_D = 10,\ w_{dg} = 1$ | 9.39 | 3.06 |
| | $T_D = 5,\ w_{dg} = 5$ | 9.27 | 3.01 |
| | $T_D = 5,\ w_{dg} = 1$ | **9.07** | **2.94** |

Table 6: Performance of LDM+DG* on the LSUN-Churches 256-dimensional dataset. ‡ denotes that the metric is computed via Clean-FID (Parmar et al., 2021).

| Method | Clean-FID‡ | CLIP-FID‡ | KID‡ |
|---|---|---|---|
| $T = 200$ | 6.67 | 4.89 | 0.0039 |
| $T = 200, \ T_D = 20, \ w_{dg} = 2.0$ | 6.99 | 4.96 | 0.0044 |
| $T = 200, \ T_D = 10, \ w_{dg} = 0.5$ | 6.43 | 4.73 | 0.0037 |
| $T = 200, \ T_D = 10, \ w_{dg} = 0.1$ | **6.50** | **4.80** | **0.0032** |

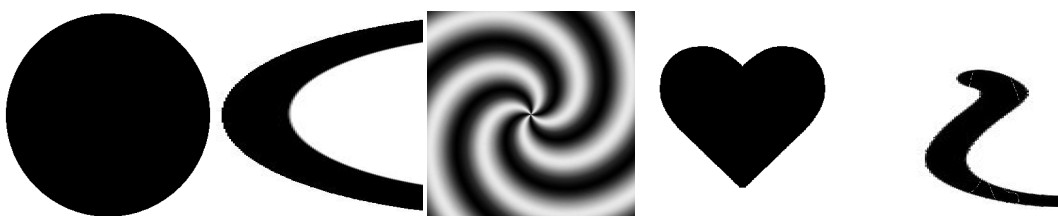

Figure 5: (♣ Color online) Images considered in generating the source and target in the *Shape morphing* experiment.

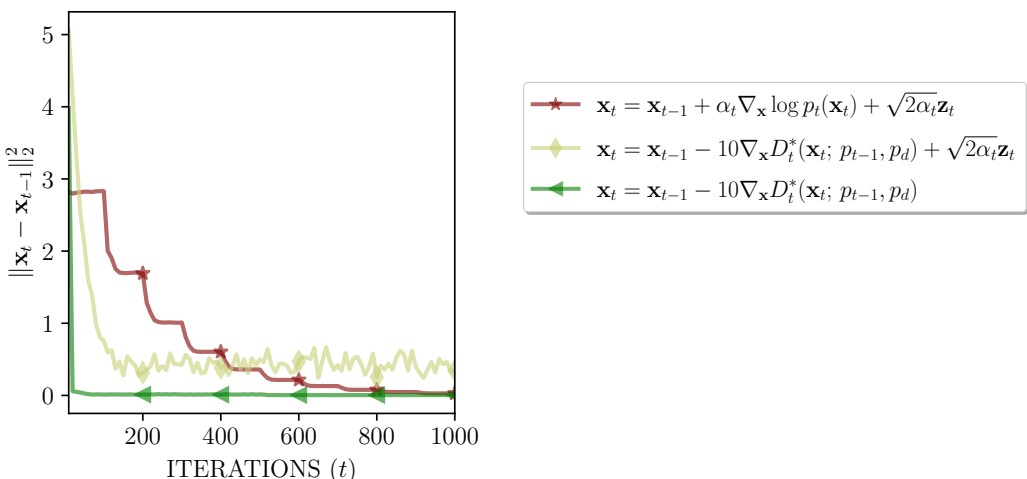

Figure 6: (♣ Color online) Plot comparing the *iterate convergence* of the discriminator-guided Langevin diffusion model, compared against the baseline NCSNv1 (Song & Ermon, 2019) model. The score in NCSN is replaced with the output of a score network $S_\theta$. The norm of the iterate-differences decays as the noise-scale in the case of NCSN. This is consistent with the observations made by Song & Ermon (2020), who showed that the score network $S_\theta$ implicitly scales its output by the noise variance $\sigma$. In discriminator-guided Langevin diffusion, adding noise results in poorer performance, while the unadjusted Langevin sampler performs the best.

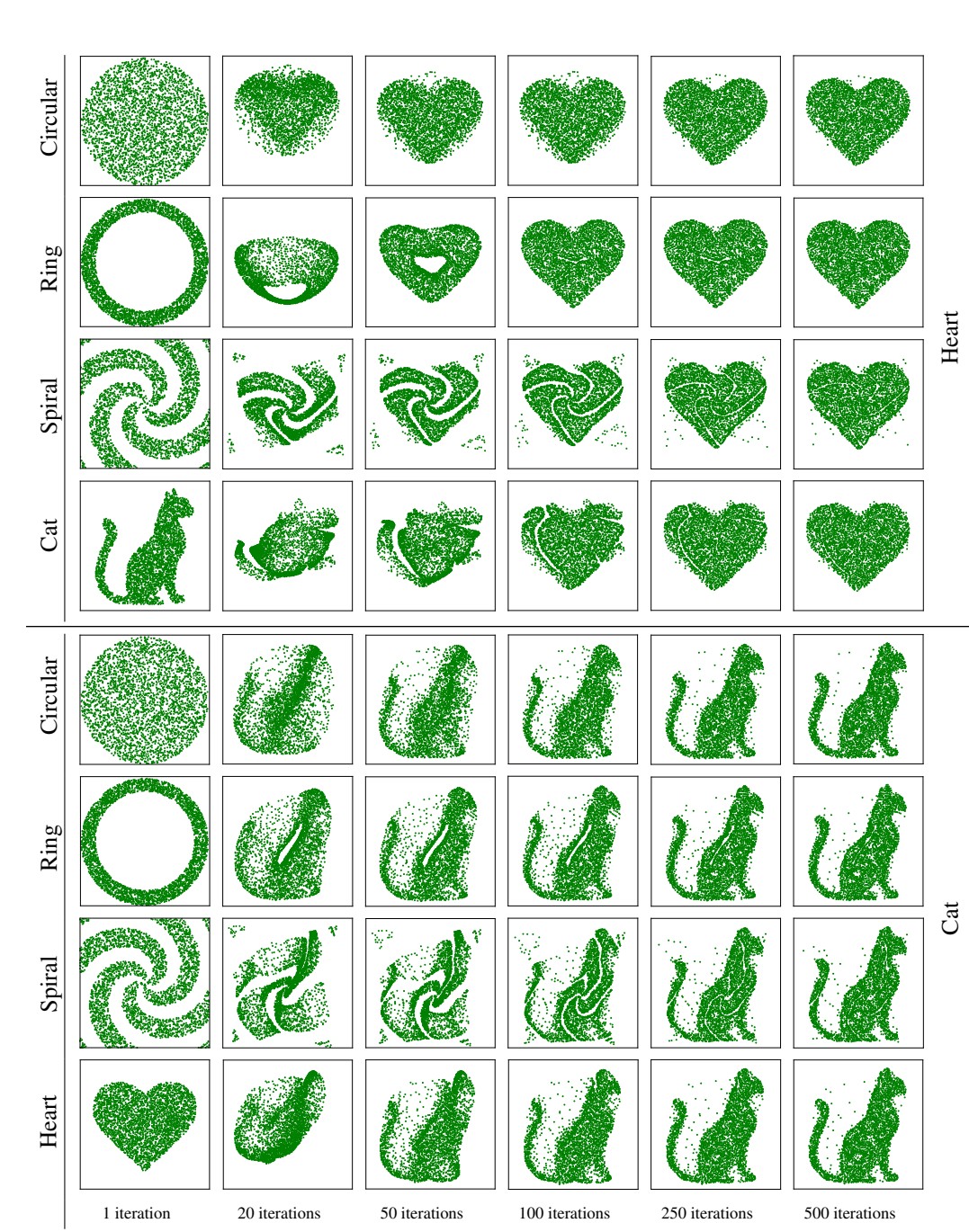

Figure 7: (🎨 Color online) Samples evolving with iterations for the discriminator-guided Langevin sampler, considering various shapes of the initial uniform distributions, given a target uniform distribution shaped like a *Heart*, or a *Cat* as indicated. For relatively simpler input shapes, such as the circular pattern, the sampler converges in about 100 iterations, while in the spiral case, the sampler converges in about 250 steps.

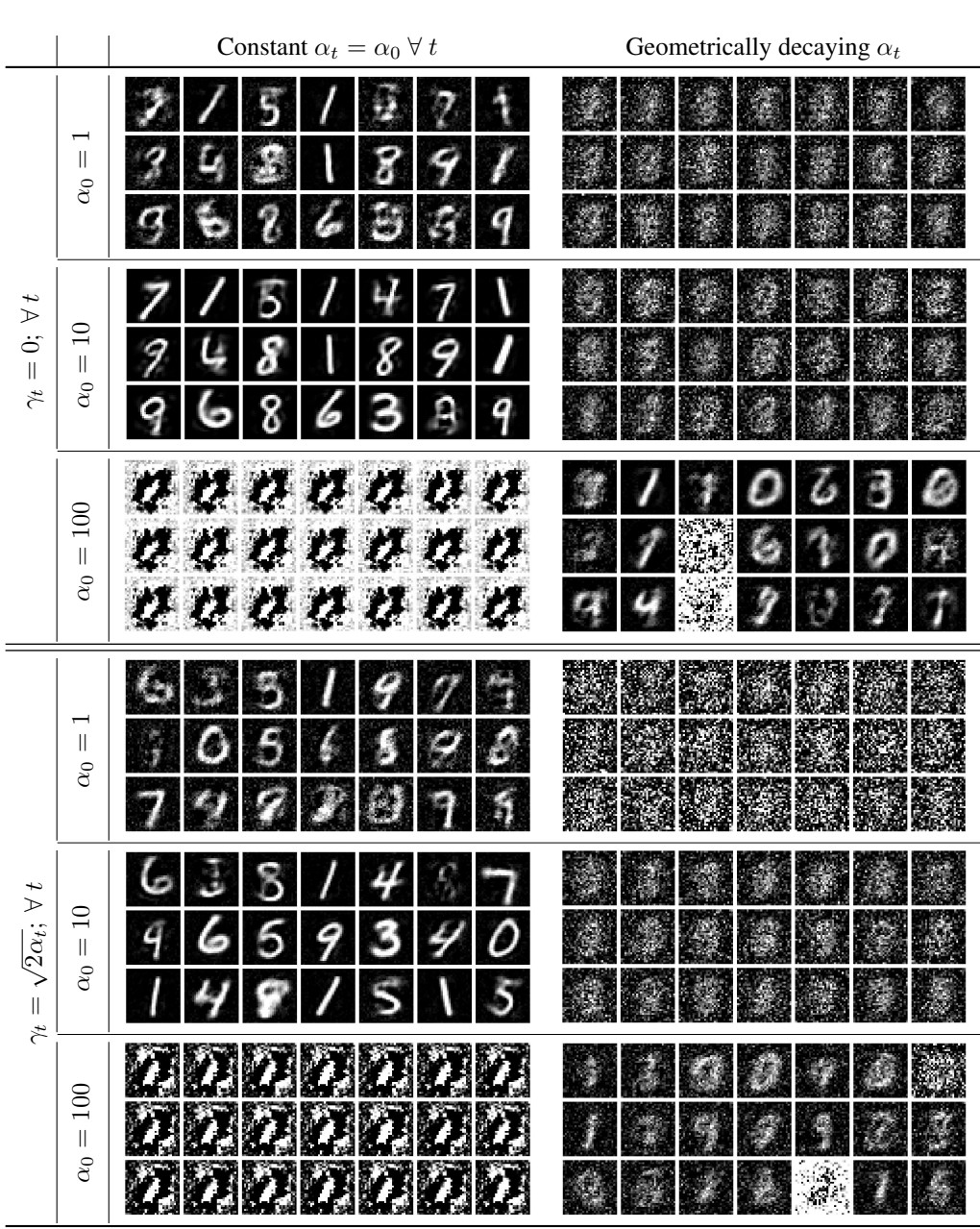

Figure 8: (🌀 Color online) Images generated using the discriminator-guided Langevin sampler with MNIST as the target. The model fails to converge when $\alpha_t$ decays, for small $\alpha_0 \leq 10$. When $\alpha_0 = 100$, some samples diverge due to gradient explosion. We observe that $\alpha_0 = 10$, with $z_t = 0$ yields the best performance.

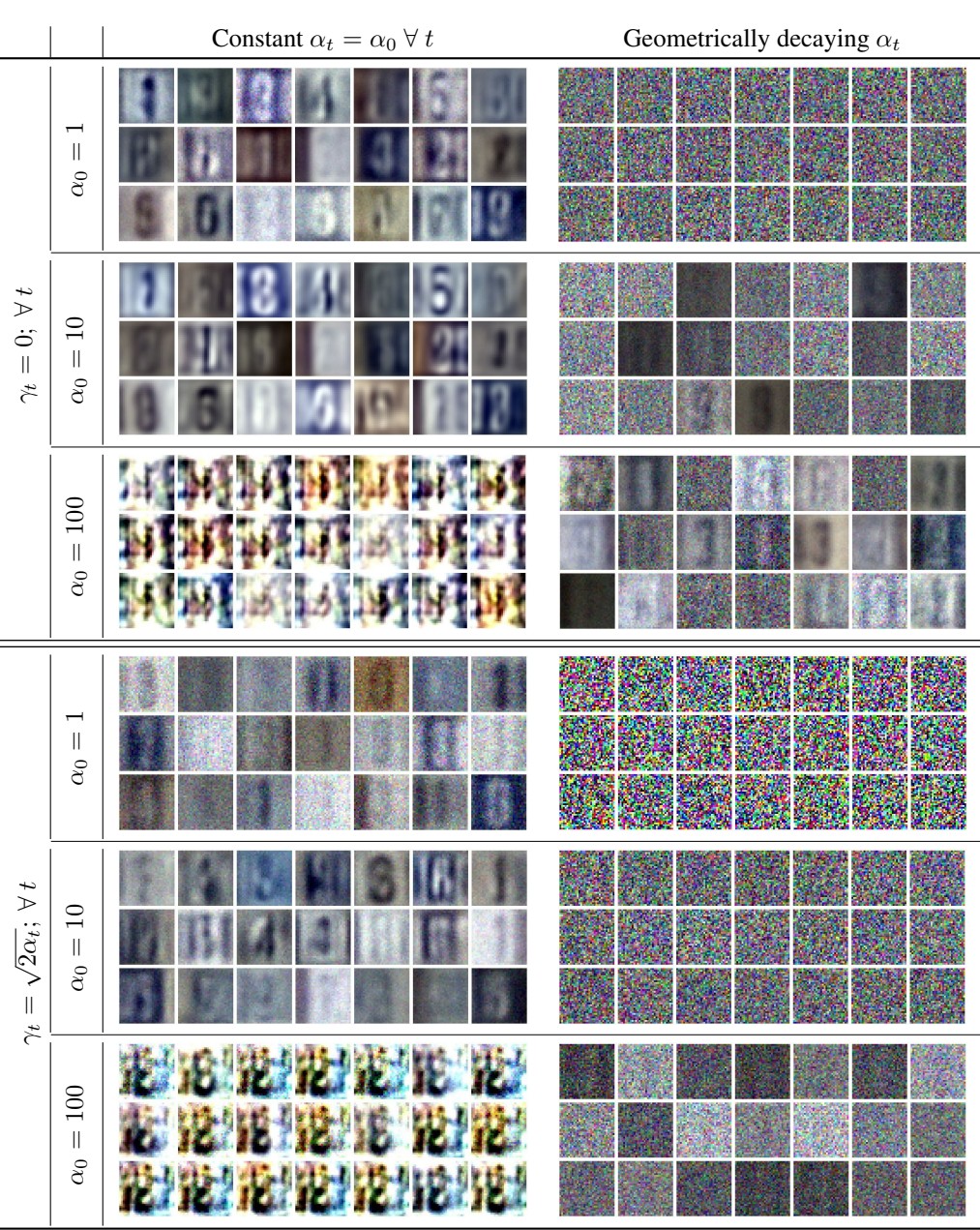

Figure 9: (🌑 Color online) Images generated using the discriminator-guided Langevin sampler with SVHN as the target. The model fails to converge with geometrically decaying $\alpha_t$, or when $\boldsymbol{z}_t$ is not the zero vector. As in the case of MNIST, observe that $\alpha_0 = 10$, with $\boldsymbol{z}_t = 0$ yields the best performance. Setting $\alpha_0 = 1$ with $\boldsymbol{z}_t = 0$ results in slow convergence.

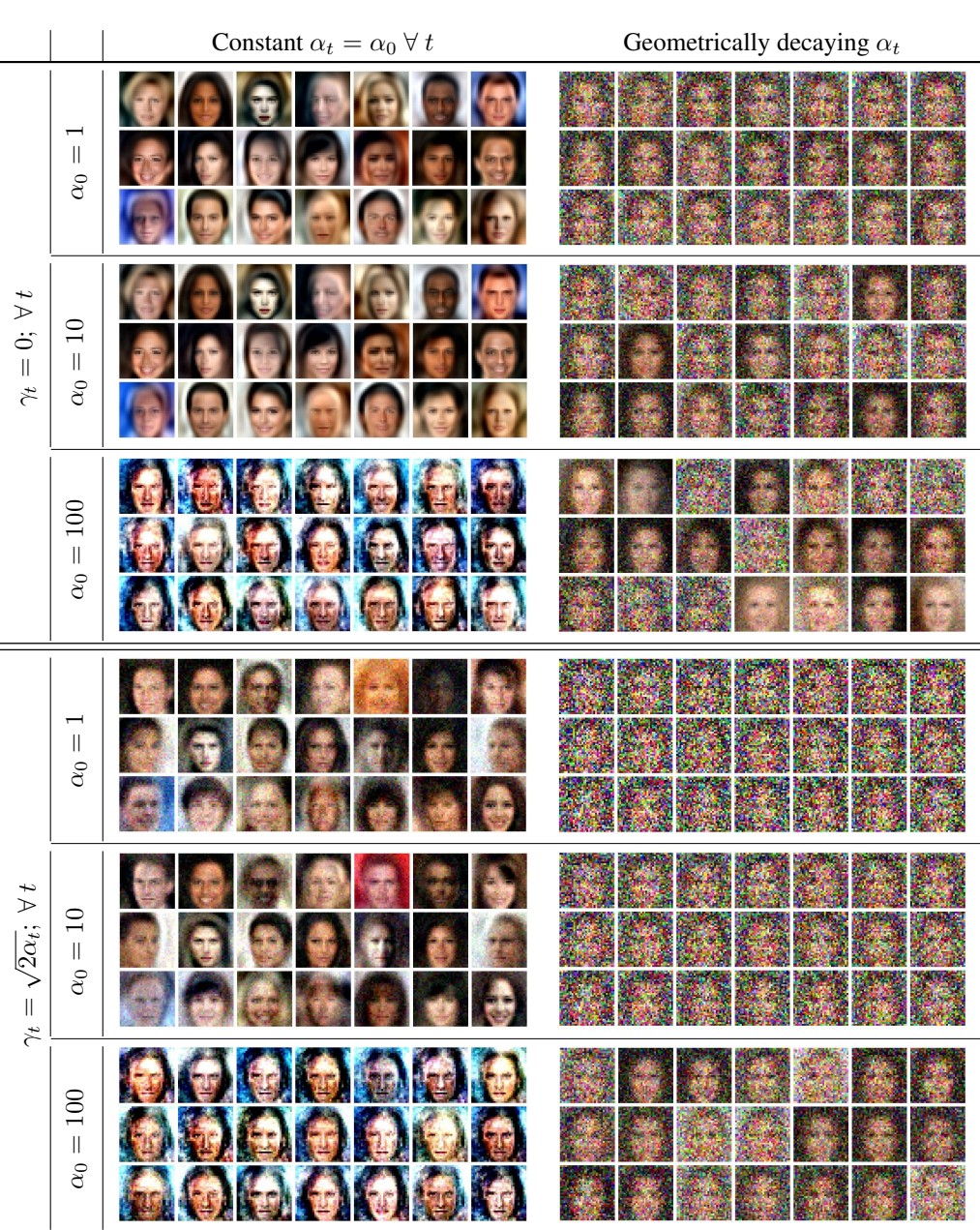

Figure 10: (🌀 Color online) Images generated using the discriminator-guided Langevin sampler with CelebA as the target. The model fails to converge when $\alpha_t$ decays geometrically, or when $z_t \neq 0$. Setting $\alpha_0 \in [1, 10]$, with $z_t = 0$ results in the sampler generating realistic images. For these choices of $\alpha_0$, when $z_t \neq 0$, the generated images are noisy.

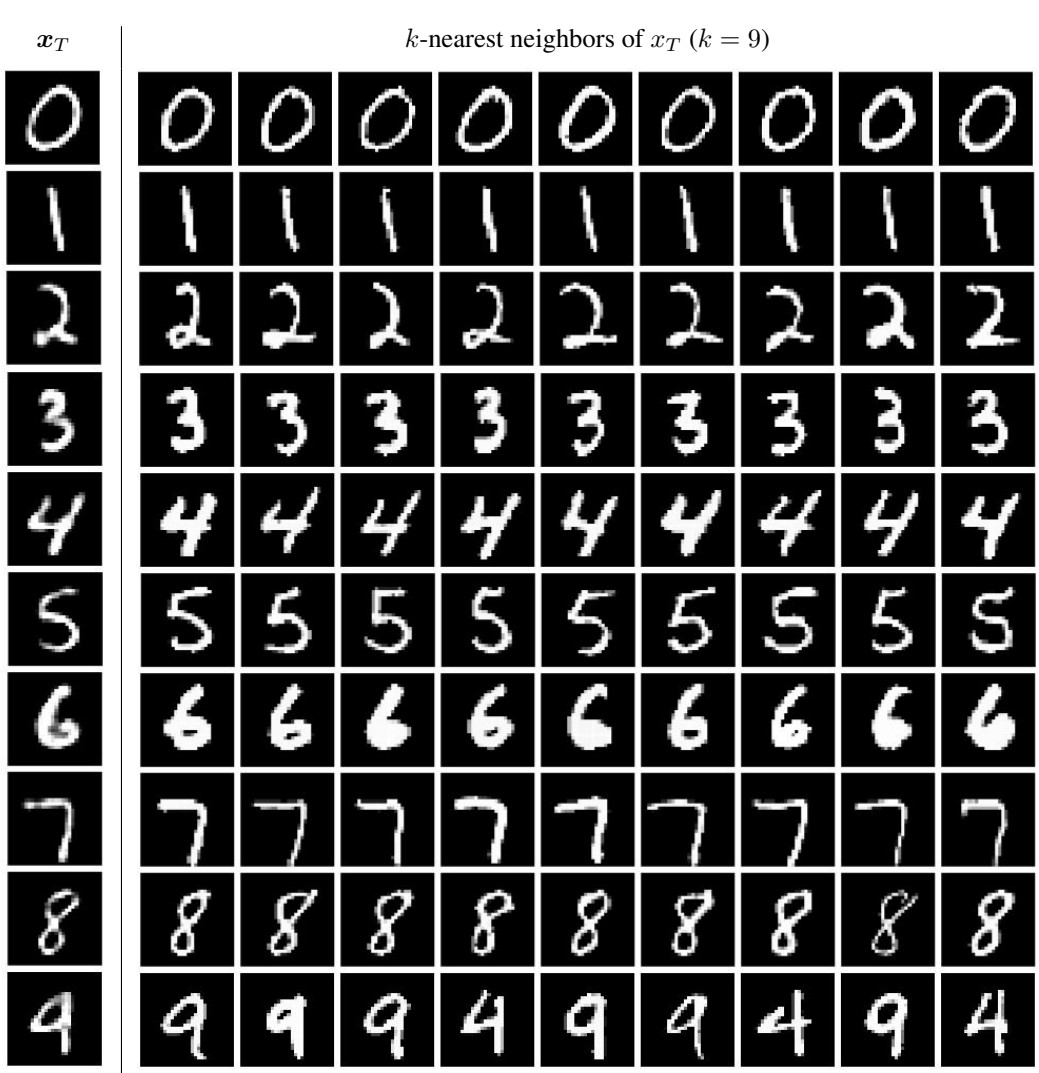

Figure 11: (🔴 Color online) The $k$-nearest neighbor ($k$-NN) test performed on images generated by the discriminator-guided Langevin sampler, when $\alpha_t = \alpha_0 = 10$ and $\boldsymbol{z}_t = 0$, on the MNIST dataset. We observe that the generated images are unique and distinct from the top-9 neighbors drawn from the target dataset, indicating that the sampler **does not memorize** the images seen as part of the interpolating RBF discriminator's centers.

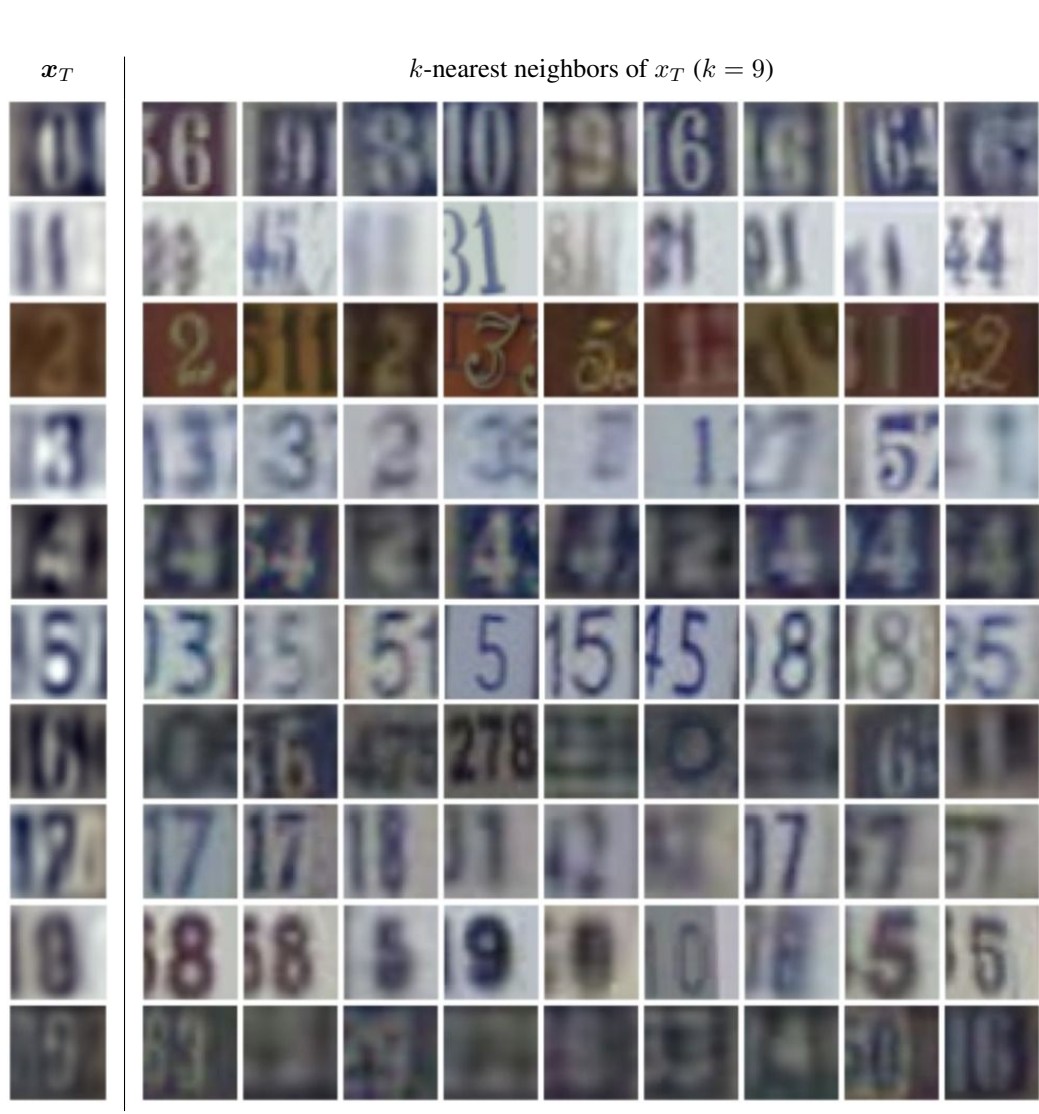

Figure 12: (🔴 Color online) The $k$-nearest neighbor (kNN) test performed on images generated by the discriminator-guided Langevin sampler, when $\alpha_t = \alpha_0 = 10$ and $z_t = 0$, on the SVHN dataset. We observe that the generated images are unique, compared to the top-9 neighbors drawn from the target dataset. For generated samples such as the *digit 9* or *digit 5*, we observe that the top $k$-NN images are from classes different from that of the generated image, indicative of the model's ability to interpolate between the classes seen as part of discriminator centers during sampling.

1674
1675
1676
1677
1678
1679
1680
1681
1682
1683
1684
1685
1686
1687
1688
1689
1690
1691
1692
1693
1694
1695
1696
1697
1698
1699
1700
1701
1702
1703
1704
1705
1706
1707
1708
1709
1710
1711
1712
1713
1714
1715
1716

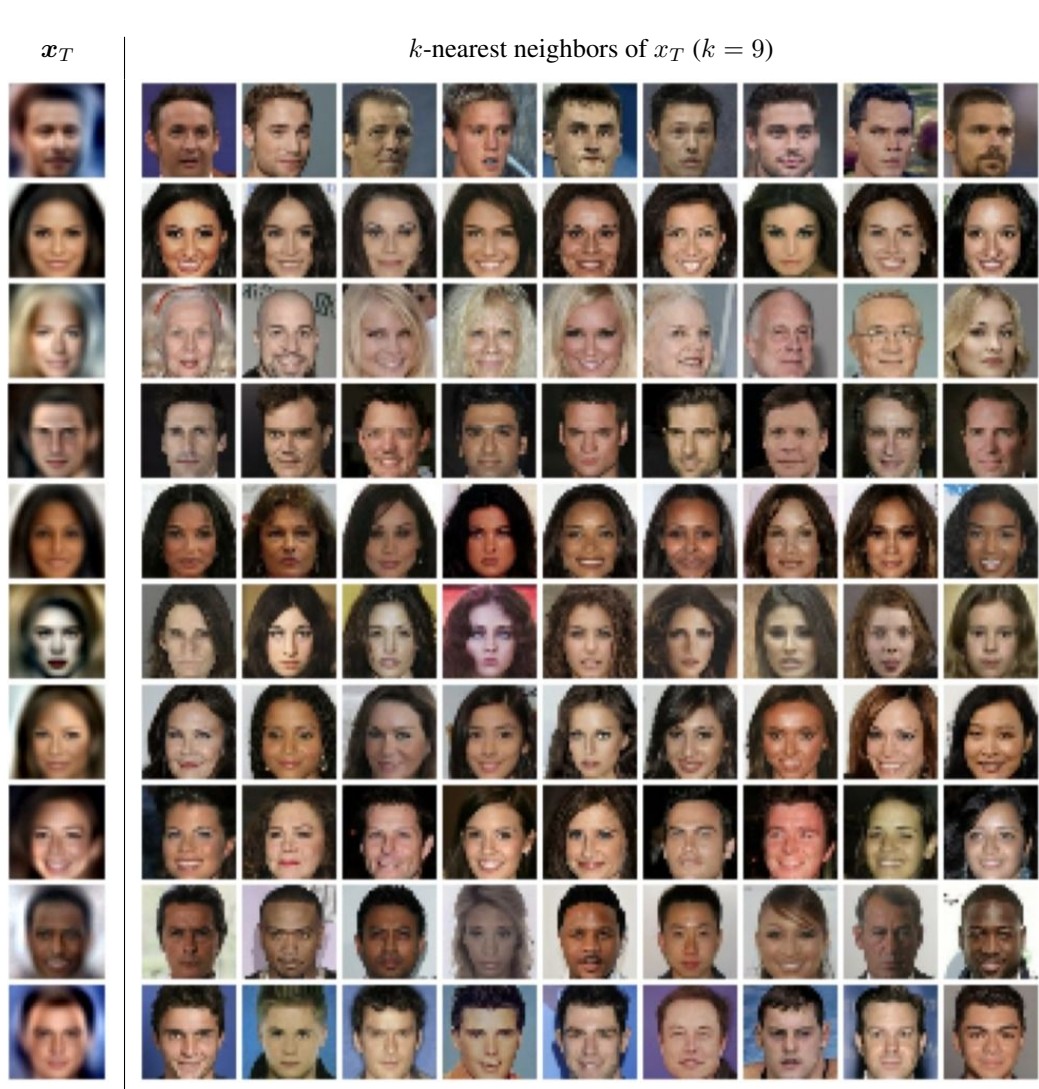

Figure 13: (🌀 Color online) The $k$-nearest neighbor (kNN) test performed on images generated by the discriminator-guided Langevin sampler, when $\alpha_t = \alpha_0 = 10$ and $z_t = \mathbf{0}$, on the CelebA dataset. The generated images are unique and distinct from the top-9 neighbors drawn from the target dataset, which suggests that the proposed approach does not memorize data.

1717
1718
1719
1720
1721
1722
1723
1724
1725
1726
1727

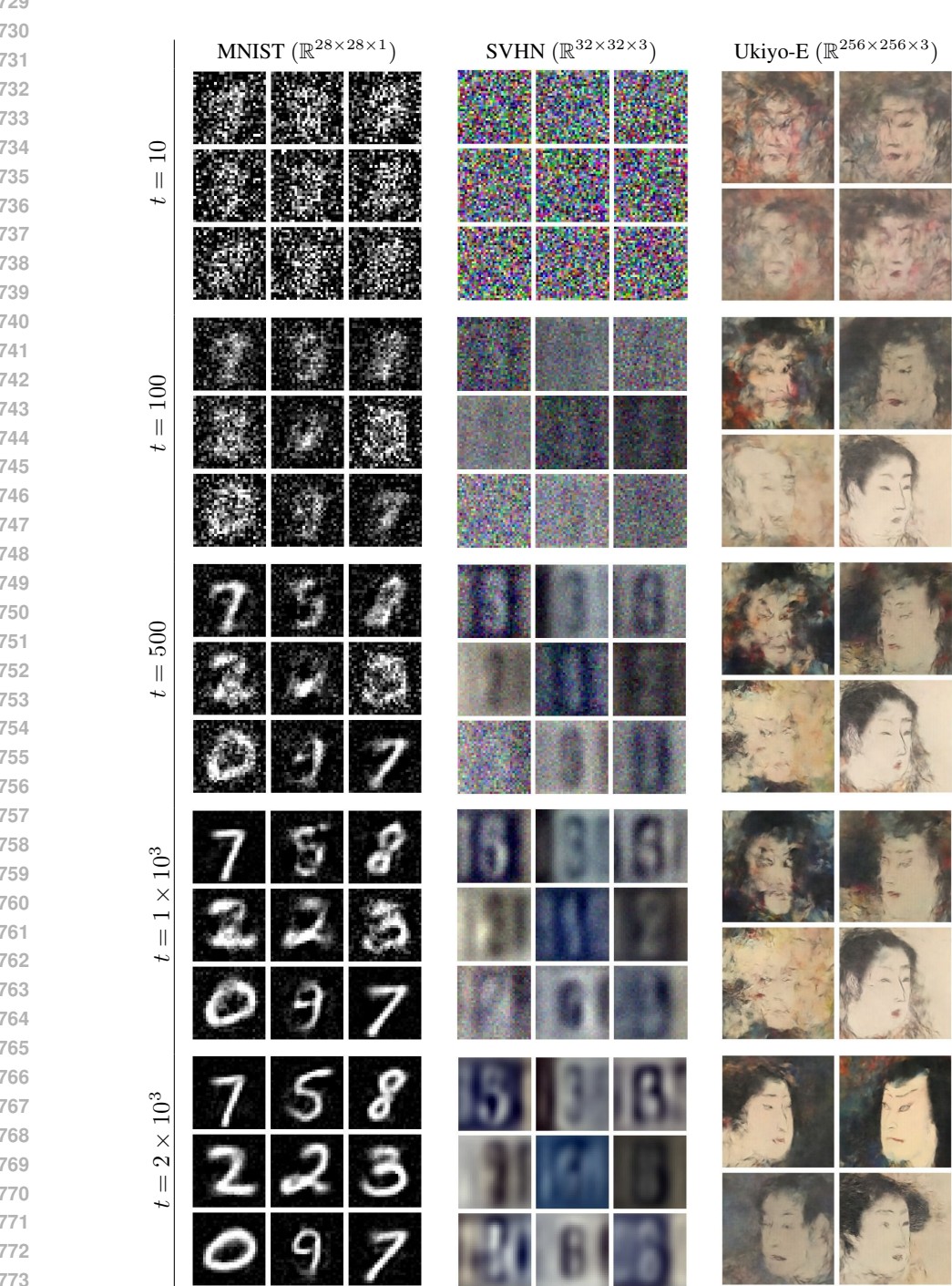

Figure 14: (🎨 Color online) Images generated using the discriminator-guided Langevin sampler. The score in standard diffusion models is replaced with the gradient field of the discriminator, obviating the need for any trainable neural network, while generating realistic samples.

EDM + Heun Sampler (128 steps)            **Ours** + Heun Sampler (40 steps)

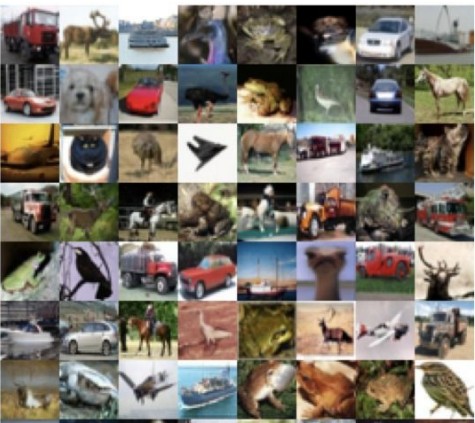 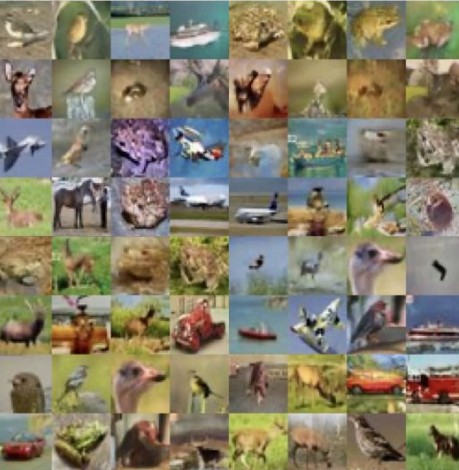

Figure 15: (🎨 Color online) Samples generated by the proposed discriminator-guided Langevin diffusion, compared against the baseline EDM (Karras et al., 2022), on the CIFAR-10 dataset. Both approaches are sampled using the Heun second-order sampler, with sampling parameters as described by Karras et al. (2022). While the baseline model requires 128 iterations, the proposed sampler generates realistic images in about 40 iterations.

EDM + EDM Sampler (256 steps)            **Ours** + EDM Sampler (80 steps)

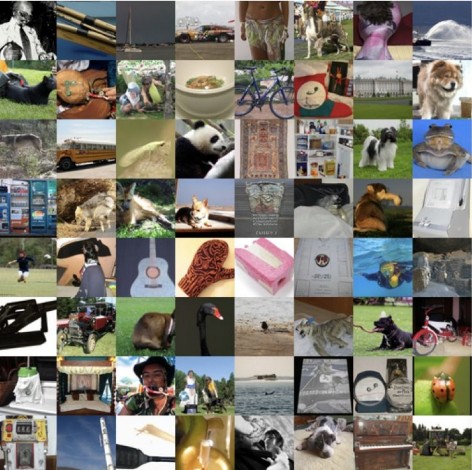 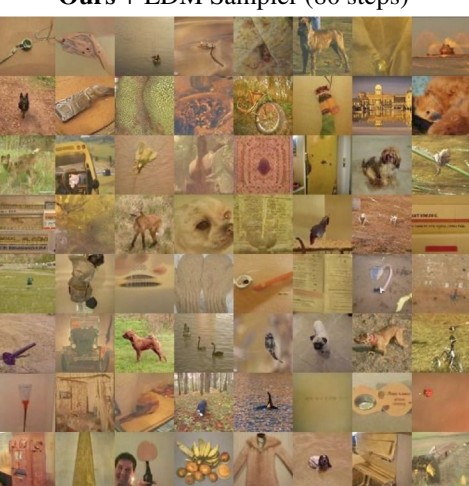

Figure 16: (🎨 Color online) Samples generated by the proposed discriminator-guided Langevin diffusion, compared against the baseline EDM approach proposed by Karras et al. (2022), on the ImageNet-64 dataset, using the EDM sampler, with sampling parameters as described by Karras et al. (2022) for the baseline. The baseline model requires 256 iterations, while the proposed discriminator-guided Langevin sampler converges in about 80 steps. The images generated by discriminator-guided Langevin diffusion lack significant color diversity, but were obtained entirely from kernel-guided sampling, without the need for training a score network. The issue of lack of sufficient color diversity on ImageNet-64 dataset requires further investigation.

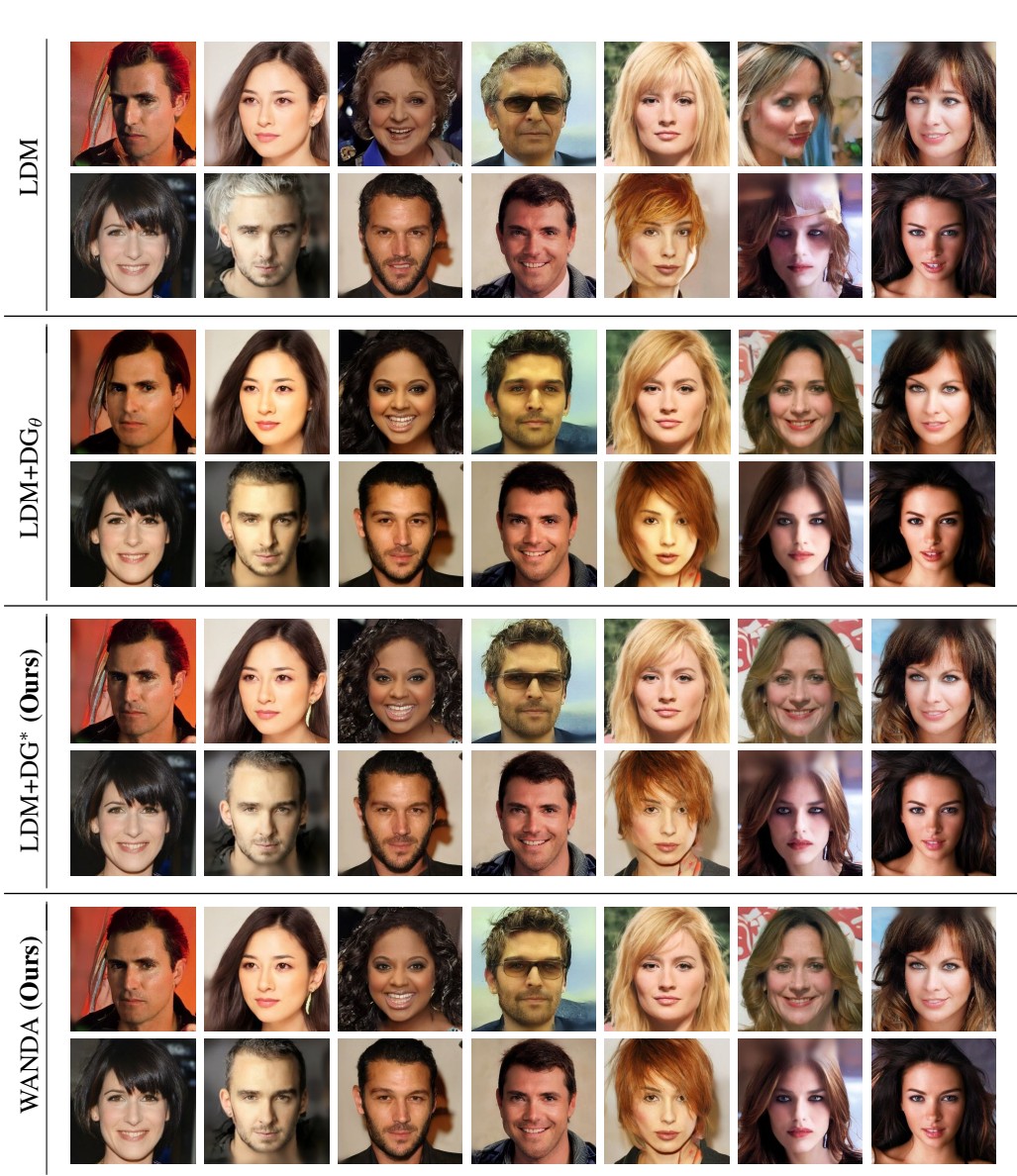

Figure 17: A comparison of the 256-dimensional CelebA-HQ images generated (given the same input) by the baseline latent diffusion model (LDM), and the proposed closed-form discriminator guidance models with and without time-step-shifted sampling (WANDA and LDM-DG*, respectively). Images generated by LDM+DG$_\theta$ are oversmooth. The discriminator guidance in LDM-DG* significantly improves the quality of the images generated, by removing artifacts. WANDA is capable of generating images with a quality comparable to that of LDM-DG*, with relatively fewer function evaluations.

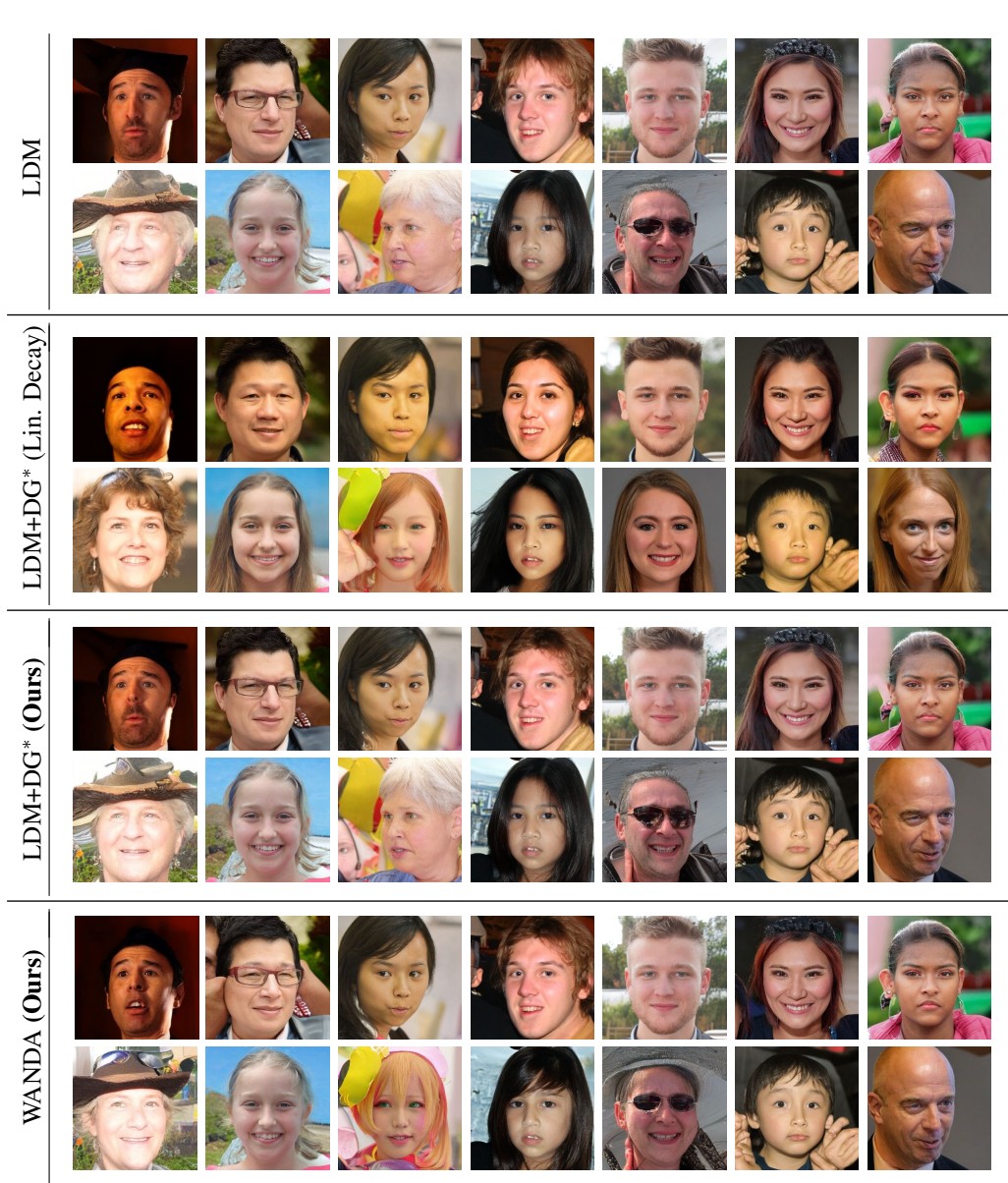

Figure 18: A comparison of the 256-dimensional FFHQ images generated (given the same input) by the baseline latent diffusion model (LDM), and the proposed closed-form discriminator guidance models with and without time-step-shifted sampling (WANDA and LDM-DG*, respectively). Images generated by LDM+DG* with the linear decay (Lin. Decay) on $w_{dg,t}$ are either oversmooth or have saturated colors, which we attribute to the discriminator guidance not decaying sufficiently fast. The discriminator guidance in LDM-DG* significantly improves the quality of the images generated, by removing artifacts. WANDA is capable of generating images with a quality comparable to that of LDM-DG*, with relatively fewer function evaluations.

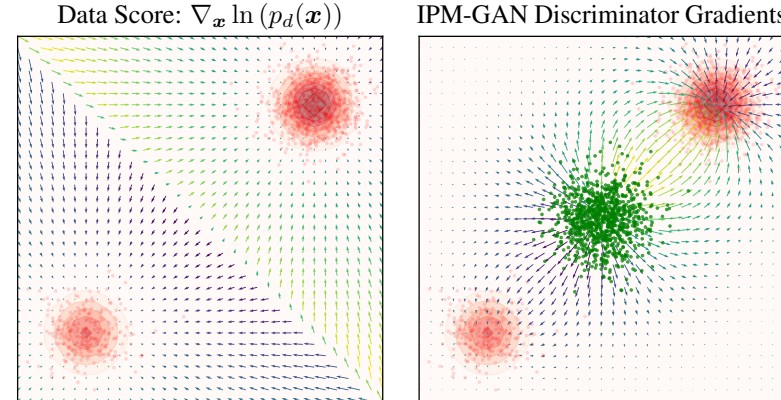

Figure 19: (♣ Color online) The loss landscape of the closed-form IPM-GAN discriminator, juxtaposed against the *(Stein) score* of the target data, for a Gaussian mixture $p_d = \frac{1}{5}\mathcal{N}(-5\mathbf{1}_2, \mathbb{I}_2) + \frac{4}{5}\mathcal{N}(5\mathbf{1}_2, \mathbb{I}_2)$. The starting distribution, $p_T$ for the T-step diffusion process, is the standard normal Gaussian. All integral probability metric (IPM) minimizing GANs minimize the gradient field of the density difference $p_d - p_g$ convolved with a kernel $\kappa$, which corresponds to a kernel-convolved version of the score. The repulsive nature of the gradient field of the Discriminator improves stability and accelerated sampling in the proposed closed-form discriminator-guided diffusion.

## F    WAVELET-BASED NOISE VARIANCE ESTIMATION

To estimate the variance $\sigma^2$ of the noise $W[t]$ from the data $X[t] = W[t] + f[t]$ where $X[t]$ is $x_t$, we need to suppress the influence of $f[t]$. When $f$ is piecewise smooth, a robust estimator is calculated from the median of the finest-scale wavelet coefficients.

A signal $X$ of size $N$ has $N/2$ wavelet coeffecients $\{\langle X, \psi_{l,m}\rangle\}_{0 \leq m < N/2}$ at the finest-scale $2^l = 2N^{-1}$. The coefficient $|\langle f, \psi_{l,m}\rangle|$ is small if $f$ is smooth over the support of $\psi_{l,m}$, in which case $\langle X, \psi_{l,m}\rangle \approx \langle W, \psi_{l,m}\rangle$. In contrast, $|\langle f, \psi_{l,m}\rangle|$ is large if $f$ has sharp transitions in the support of $\psi_{l,m}$. A piece-wise regular signal has few sharp transitions, and thus produces a number of large coefficients that is small compared to $N/2$. At the finest scale, the signal $f$ thus influences the value of a small portion of large-amplitude coefficients $\langle X, \psi_{l,m}\rangle$ that are considered to be "outliers." All others are approximately equal to $\langle W, \psi_{l,m}\rangle$, which are independent Gaussian random variables of variance $\sigma^2$.

A robust estimator of $\sigma^2$ is calculated from the median of $\langle X, \psi_{l,m}\rangle_{0 \leq m < N/2}$. The median of $P$ coefficients $\text{Med}(\alpha_p)_{0 \leq p < P}$ is the value of the middle coefficient $\alpha_{n_0}$ of rank $P/2$. As opposed to an average, it does not depend on the specific values of coefficients $\alpha_p \geq \alpha_{n_0}$. If $M$ is the median of the absolute value of $P$ independent Gaussian random variables of zero mean and variance $\sigma_0^2$, then one can show that

$$E\{X\} \approx 0.6745\sigma_0 \tag{13}$$

The variance $\sigma^2$ of the noise $W$ is estimated from the median $M_X$ of $\{\langle X, \psi_{l,m}\rangle\}_{0 \leq m < N/2}$, by neglecting the influence of $f$:

$$\tilde{\sigma} = \frac{M_X}{0.6745} \tag{14}$$

Indeed, $f$ is responsible for few large-amplitude outliers, and these have little impact on $M_X$.

$\boldsymbol{x}_T$          $k$-nearest neighbors of $x_T$ ($k = 9$)

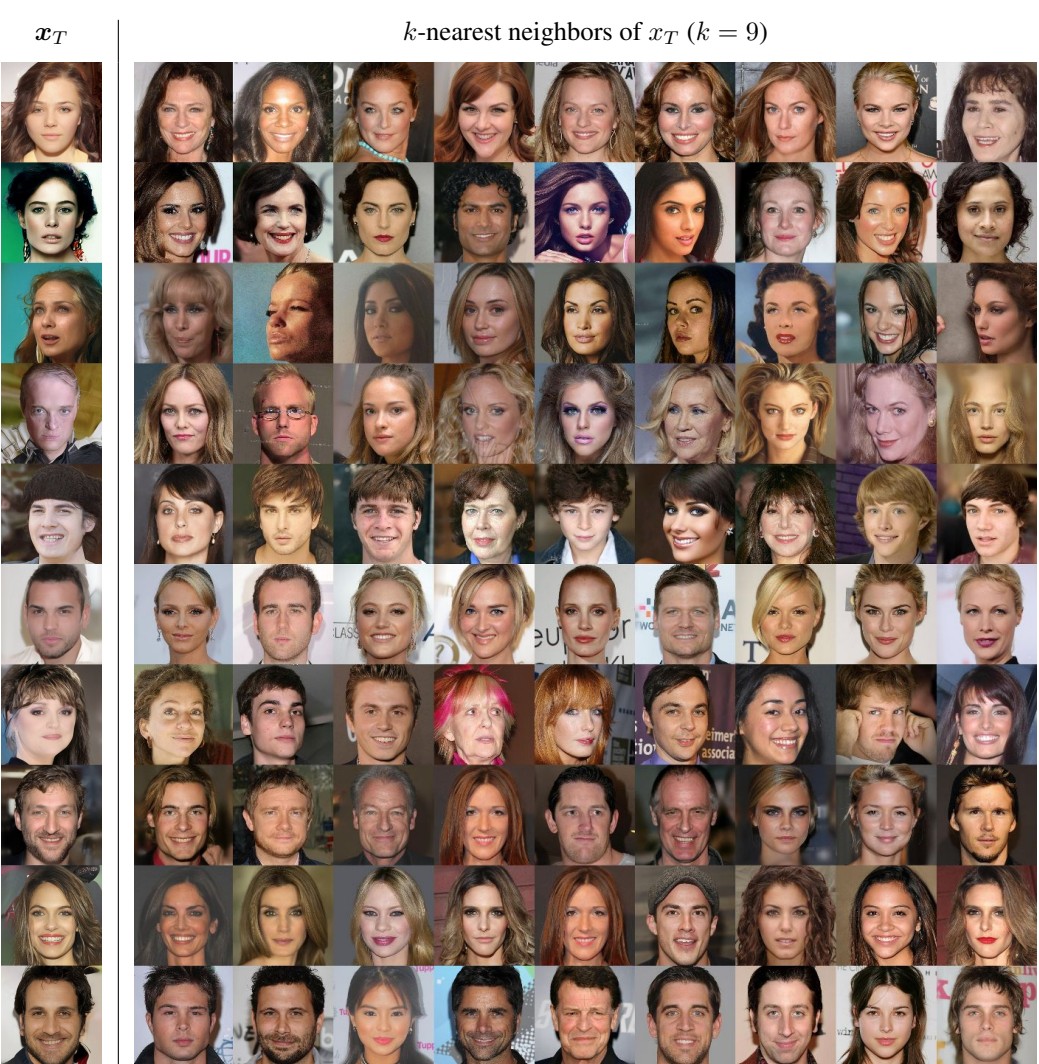

Figure 20: (🔴 Color online) The $k$-nearest neighbor (kNN) test performed on images generated by the discriminator-guided DPM sampler, on the CelebA-HQdataset. The generated images are unique and distinct from the top-9 neighbors drawn from the target dataset, which suggests that the proposed approach does not memorize data.

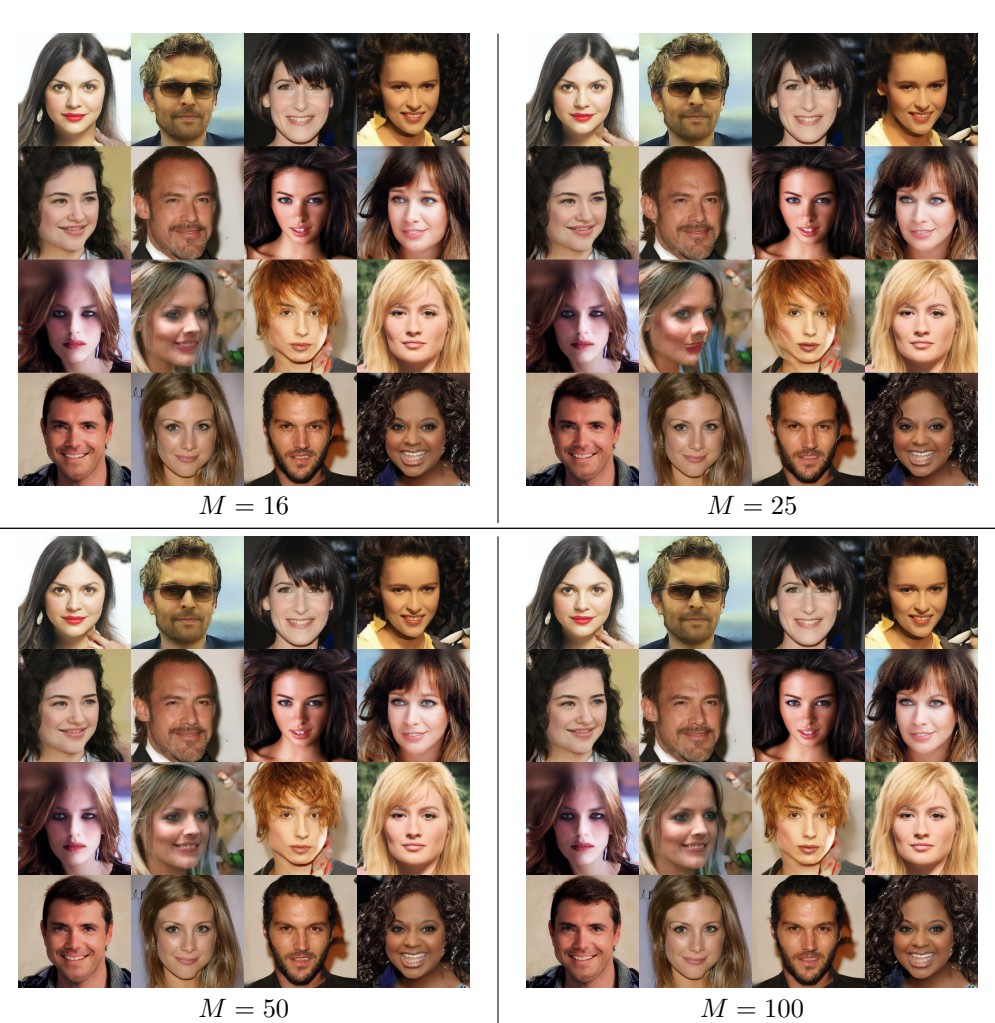

Figure 21: (🔴 Color online) A comparison of the images generated for varying numbers of centers $M$ considered in the closed-form discriminator. We observe that the performance is generally unaffected by this choice, and using $M = 50$ is preferred, to ensure statistically, that the sample estimates converge.

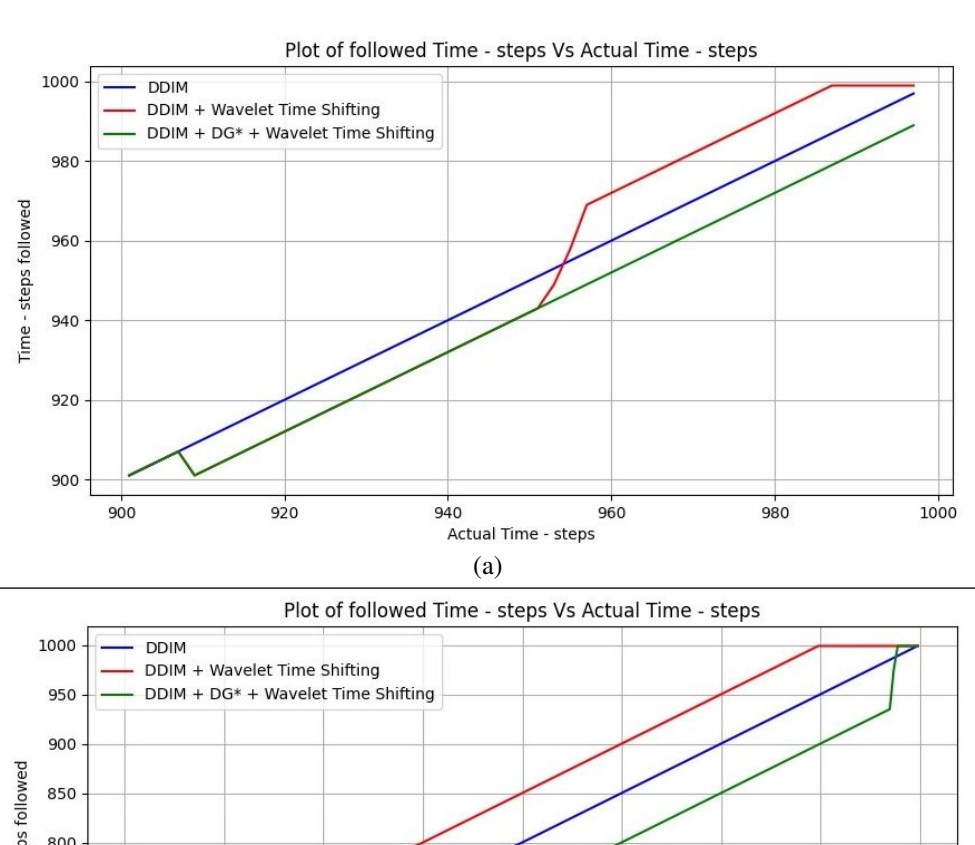

(a)

(b)

Figure 22: (♣ Color online) A comparison of the predicted and actual time step $t$ in WANDA, and the baseline DDIM variants for (a) $T_D = 900$ and (b) $T_D = 600$, respectively, with $T = 1000$. We observe that the the discriminator guidance term introduces a jump (a sharp drop in the *time step followed* for the green curve) of 2-10% of the steps is either setting.

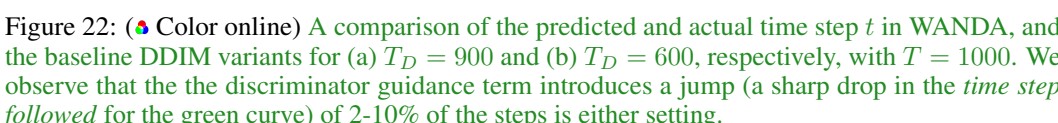

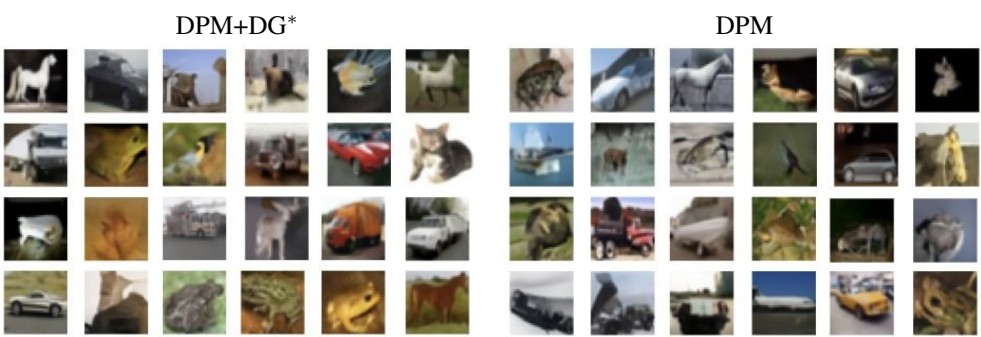

Figure 23: (♣ Color online) Samples generated by the proposed DPM+DG* sampler, compared against the DPM sampler on the CIFAR-10 dataset.

