# OpenReview forum: "Accelerated Diffusion using Closed-form Discriminator Guidance"
_ICLR.cc/2025/Conference — Submitted to ICLR 2025_

### Official Review · Reviewer_QDuM · 2024-10-22

**Soundness:** 2
**Presentation:** 3
**Contribution:** 2
**Rating:** 5
**Confidence:** 4

**Summary:**

This paper derives a closed form expression for the IPM-GAN discriminator, whose gradient is then used to perform Langevin sampling or guide diffusion generation.

**Strengths:**

**Originality** : this paper is original in the aspect that it uses closed-form expressions for the optimal discriminator instead of discriminators parametrized by neural nets. It is also, to the best of my knowledge, the first paper to use IPM-GAN discriminators for diffusion sampling and guidance.

**Quality** : this paper provides some theoretical grounds for using IPM-GAN discriminators over f-GAN discriminators for diffusion. Some experiments on toy and image datasets are also provided as evidence that the proposed discriminator guidance improves sample quality.

**Clarity** : motivation as well as main theoretical results are presented clearly in the main text. In particular, in Section 3.1, the authors do a good job of explaining how IPM-GAN discriminator guidance could be better than f-GAN discriminator guidance.

**Significance** : this paper provides some theoretical grounds for how diffusion and GANs could be used synergistically to enhance generative performance.

**Weaknesses:**

**Missing theory** : there is no guarantee that the proposed discriminator-based generator process correctly samples from the data distribution.

**Missing baselines** : while there may be other relevant baseline methods, the paper is missing comparison with the most important baseline - discriminator guidance [1]. The authors *must* compare the proposed method with [1] if they wish to support the claim that IPM-GAN discriminator guidance provides improvement over f-GAN discriminator guidance. In addition, if the authors wish to claim that the proposed method accelerates diffusion sampling, they must compare with additional relevant baselines such as [2,3].

**Missing ablation** : there is no ablation w.r.t. number of data samples used when approximating the optimal discriminator. Is this because a large number of samples is needed to obtain reasonable results?

[1] Refining Generative Process with Discriminator Guidance in Score-based Diffusion Models, ICML 2023

[2] DPM-Solver: A Fast ODE Solver for Diffusion Probabilistic Model Sampling in Around 10 Steps, NeurIPS 2022

[3] Fast ODE-based Sampling for Diffusion Models in Around 5 Steps, CVPR 2024

**Questions:**

**Q1** : Why does the proposed method accelerate diffusion sampling? My understanding is that IPM-GAN discriminator guidance mitigates the mismatch between model distribution and the true data distribution at each time $t$. Sampling acceleration is related to minimizing the truncation error during numerical integration of the diffusion SDE/ODE as we reduce the number of function evaluations.

**Q2** : At lines 254-255, what do the authors mean by *the kernel $\kappa$ is the Green's function to the differential operator governing the optimal discriminator*? Please provide a concrete equation of $\kappa$ in relation to $D$.

**Q3** : Does the generative process defined by closed form discriminator gradients generalize beyond training data? In other words, how do we know whether the generative model is not simply memorizing the training dataset?

---

> ### Author Response · Authors · 2024-11-22
> **Response to Reviewer's Comments**
>
> Thank you for your valuable feedback! Please find a detailed response to your questions below.
>
> - **Theory for discriminator-based generator process**: The optimality of the GAN generator training, given the closed-form Coulomb and polyharmonic-spline kernel-based discriminator, has already been explored by Asokan and Seelamantula 2023a, 2023b. A similar analysis for neural-tangent-kernel-based discriminators was carried out by Franceschi et al. 2022. In summary, these results show that, given the optimal discriminator, the generator training converges to the optimal solution $p_g = p_d$. In consideration of your comment, to improve clarity, we have included a summary of this proof in Appendix D.1 of the revised submission. In the context of discriminator-guided Langevin diffusion, Lunz et al., 2018 have already proved the convergence of iterates involving $\nabla D_t^*({x}_t)$. We already discuss the application of this convergence result to our solution in Appendix D.3.
>
> - **Comparisons against Kim et al., 2023 [1]**: We already have this comparison (LDM+$DG_\theta$ in Table 2). [1] consider discriminator guidance in both the latent space (with LSGM-G++, on CIFAR-10) and in the pixel space (with a pre-trained encoder and the ADM classifier). Since we present experiments on latent-space generation on CelebA-HQ (cf. Sec. 5), we reproduced the LSGM-G++ setting (from Table 8 in [1]) for this dataset, which is the LDM+DG$_\theta$. This is already discussed in Lines 480-485.
>
> - **Baselines for accelerated diffusion sampling**: The proposed discriminator guidance term is orthogonal to baselines such as [2,3] wherein better ODE solvers are used to accelerate sampling. As such, the closed-form discriminator guidance (+DG$^*$) can be combined with these techniques as well. Given the limited rebuttal timeline, we also include experiments on CelebA-HQ, consider the suggested DPM solver [2], with and without +DG$^*$ (cf. Appendix E.2, Table 4 of the revised manuscript). We observe that including discriminator guidance allows us to further accelerate the sample generation process, with the DPM+DG$^*$ sampler achieving comparable performance in $T=15$ steps (1 discriminator step and 14 DPM-solver steps, CLIP-FID = 9.66), as the baseline DPM model with $T=20$ (CLIP-FID = 9.50). On the other hand, the DPM+DG$^*$ with $T=20$ outperforms (CLIP-FID = 9.22) the baseline for the same $T$. We will include comparisons with the Fast-ODE solver [3] in the camera-ready version of the manuscript.
>
> - **Ablation w.r.t number of samples**: The closed-form discriminator is computed as a sample estimate using a batch of $M \leq B$ centers, where $B$ is the batch size, set to 80 in our experiments, and can be selected based on the available compute budget. However, since the discriminator guidance only affects the initial stages of sampling, the choice of $M$ does not affect the generated image quality. To illustrate this effect we now include Figure 21 in the Appendix, which presents images generated for varying numbers of centers \(M\) considered in the closed-form discriminator.
>
> - **Closed-form discriminator guidance and accelerated sampling**: As we discuss in Sec 3.1 (Lines 297-304), unlike the score, DG$^*$ provided a strong *repulsive* force away from the initial distribution, pushing the samples towards the target $p_d$. This repulsive force accelerated the sampling process in earlier iterations. To validate this, as we show in Fig. 6 of the Appendix, on  comparing the iterate convergence of the closed-form discriminator-guided diffusion (DG$^*$) against the baseline NCSNv1, we observe that the iterate convergence (the value of $\|\mathbf{x}_{t-1} - \mathbf{x}_t\|$ as t decreases) is much quicker for DG$^*$. To add intuition, we have included Figure 19 in the Appendix, to illustrate this *push-pull* gradient field on toy data. In Summary, in addition to mitigating the distribution mismatch (as seen in standard discriminator guidance), the closed form discriminator also accelerates sampling by providing stronger gradients in earlier training steps. Please also see our response to Reviewer  Z719 on the jump seen with DG$^*$.
>
> - **Expression for $\kappa$**: We already present expressions for various choices for $\kappa$
> In Table 8 of the Appendix. We have included the expression of an example kernel in Section 2.2 of the revised Manuscript.
>
> - **On memorizing the training dataset**: Using DG$^*$ *does not* result in memorizing the training data. We have already provided ablation on this, for the MNIST, SVHN and CelebA datasets in Appendix E.2 of the submission (cf. Fig 11-13). We have also included similar comparisons on CelebA-HQ as part of the revised manuscript (cf. Fig. 20 of the revised manuscript)
>
> Please let us know if any additional clarity is needed. We hope that we were able to address your concerns regarding our submission and that you'll consider raising your score in case the questions were resolved.

---

> > ### Comment · Reviewer_QDuM · 2024-11-26
> >
> > Thank you for the reply. While the authors have addressed my concerns regarding experiments, I still have major doubts about the theoretical correctness of the proposed approach, and whether it truly accelerates sampling. Hence, I still keep my rating reject. Please see below for specifics.
> >
> > **The authors' reply does not address my concern regarding whether the proposed discriminator guided sampling converges to the data distribution.**
> >
> > The authors write in the reply that
> >
> > > The optimality of the GAN generator training, given the closed-form Coulomb and polyharmonic-spline kernel-based discriminator, has already been explored by Asokan and Seelamantula 2023a, 2023b. A similar analysis for neural-tangent-kernel-based discriminators was carried out by Franceschi et al. 2022.
> >
> > But these results are not regarding discriminator guided diffusion sampling process proposed by the authors. In fact, in the experiments, the authors turn off discriminator guidance after the first few sample update steps (see Section 5.1), deviating even further from the setup of the referred works which require discriminator guidance at all sample update steps. The authors also write that
> >
> > > In the context of discriminator-guided Langevin diffusion, Lunz et al., 2018 have already proved the convergence of iterates involving $\nabla D_t^*(x_t)$. We already discuss the application of this convergence result to our solution in Appendix D.3.
> >
> > However, the work by Lunz et al. is about gradient flows, which is a deterministic (noiseless) sampling procedure, and their theoretical results do not apply to stochastic sampling procedures such as Langevin diffusion.

---

> ### Author Response · Authors · 2024-11-27
>
> Thank you for your response and constructive feedback. We would like to provide additional clarity on the closed-form discriminator-guided sampling, and its links to Langevin Sampling.
>
>  - **Does closed-form discriminator guidance truly accelerate sampling?** We would like to reiterate that, as we mentioned in our response, **The acceleration achieved by the proposed discriminator guidance and accelerated sampling approaches are orthogonal to one another.** in the context of the discriminator guidance, acceleration is defined by means of the strong push-pull nature of the gradient field of the discriminator, which pushes the initial distribution towards $p_d$ with significantly stronger gradients as compared to the standard score -- the strength being defined via the chosen kernel.  In accelerated sampling, as defined in the context of [1,2], Kim et al., etc, acceleration is in terms of improved discretization of the SDEs, or in better approximations of the score. *The closed-form discriminator does not improve SDE discretization or score approximations. It provides an alternative gradient field, which is stronger than the score field when far away from the data distribution during early stages of sampling.*
>
>
>  - **Applicability of  Lunz et al.**: **The results of Theorem 2 and Lemma 3 indicate that the convergence of a sampler (such as the GAN generator, or Line 342) with access to the closed-form optimal discriminator, and in the absence of noise, is guaranteed in 1 step,** i.e., if one had access to $D_{t-1}^{\*}$ and $p_d$ and $p_{t-1}$, then $G_{t}(z_t) = x_t \sim p_t = p_d$. However, this is not achievable in practice, both due to the inaccessibility of $p_{t-1}$, and sample approximations of $D_t^{*}$. The results of Lunz et al. [3], covers a scenario wherein access to the optimal discriminator is not available, which is true in the case of DG$^{\*}$ approximated with a sample estimate. We show, via the experiments in Section 4 and ablation in Appendix E.2 (Figures 8-10), even in the case of the proposed closed-form discriminator-based Langevin sampling without the score, the optimal performance is indeed achieved in the noise-free setting.
>
> - **Provided references and discriminator guided diffusion**: While we agree with the reviewer that there are primarily GAN works, *as we discuss in Sec. 1, approaches such as Franceschi et al. 2023 [4], Zhang et al., 2023 [5], have extended this analysis to the diffusion/generative-particle-model settings.* Additionally, [4] has already shown that score-based diffusion models, GANs, and gradient flows can be unified as a family of generative particle models, and derive convergence results for Langevin sampling with neural-network-based discriminators (Finding 8 of [4]). We already discussed these approaches in Lines 122-128). Our results can be viewed as a novel extension of their setting, wherein the convergence would be guaranteed in 1 step, given access to the closed-form $D_t^*$ and $p_{t-1}$ (Theorem 2, Lemma 3 of the submission).
>
> - **Convergence analysis of DG**$^{\*}$ **in Section 5.1**: We agree that the neither SDE-based convergence/acceleration analysis, nor Theorem 2/Lemma 3 readily apply to the Langevin formulation presented in Section 5 and we are currently exploring approaches to extending the existing results (such as those present in Kim et al., 2023) to this setting. However, as we discuss in Sec 5 (Lines 466-472), and as shown by Li et al. (2024) [6], diffusion models contain two stages: (a) The first stage representing the movement of the Gaussian of diffusion models input towards the image distribution, during which no modes are present, and (b) the second stage, wherein patterns and structure emerge from latching onto a specific image to generate from the target distribution. The proposed DG$^*$ framework, akin to time-step shifting (Li et al., 2024), can be viewed as improving the convergence of the first stage, while the standard score-based sampling (and the associated convergence guarantees) hold for the second stage.
>
> [4] Franceschi et al. “Unifying GANs and Score-Based Diffusion as Generative Particle Models”, NeurIPS 2023
>
> [5] Zhang et al. “Diffflow: A unified SDE for score-based diffusion models and generative adversarial networks.” arXiv 2023
>
> [6] Li et al., “Alleviating exposure bias in diffusion models through sampling with shifted time steps,” ICLR 2024
>
> Please let us know if any additional clarity is needed.

---

> > ### Comment · Reviewer_QDuM · 2024-11-27
> >
> > Thank you for the discussion. However, I am still not convinced by the authors' reply. Below, I provide feedback for each of the authors comments (in block quotes).
> >
> > > The closed-form discriminator does not improve SDE discretization or score approximations. It provides an alternative gradient field, which is stronger than the score field when far away from the data distribution during early stages of sampling.
> >
> > That is exactly why I am concerned with the proposed method. How are the authors certain the alternative gradient field, when combined with noise, guides the prior distribution into the data distribution?
> >
> > > Applicability of Lunz et al.: The results of Theorem 2 and Lemma 3 indicate that the convergence of a sampler (such as the GAN generator, or Line 342) with access to the closed-form optimal discriminator, and in the absence of noise, is guaranteed in 1 step
> >
> > Again, if previous results only apply in the noiseless scenario, how do the authors justify sampling with noise?
> >
> > > We show, via the experiments in Section 4 and ablation in Appendix E.2, even in the case of the proposed closed-form discriminator-based Langevin sampling (without the score), the optimal performance is indeed achieved in the noise-free setting.
> >
> > This result only enhances my concern about this work. The fact that the performance of the proposed method degrades with added noise enlarges my suspicion that the method is not guaranteed to converge to the data distribution when sampling with noise.
> >
> > Also, I would like to clarify that Langevin sampling without noise it not Langevin sampling. Langevin sampling converges to the stationary data distribution only when noise is injected into each sample update step. Otherwise, the samples collapse into local maxima. I suggest the authors refrain from using the term "Langevin sampling" to refer to "flow / ODE based sampling".
> >
> > > Provided references and discriminator guided diffusion: While we agree with the reviewer that there are primarily GAN works, as we discuss in Sec. 1, approaches such as Franceschi et al. 2023 [4], Zhang et al., 2023 [5], have extended this analysis to the diffusion/generative-particle-model settings. Additionally, [4] has already shown that score-based diffusion models, GANs, and gradient flows can be unified as a family of generative particle models, and derive convergence results for Langevin sampling with neural-network-based discriminators (Finding 8 of [4]). We already discussed these approaches in Lines 122-128). Our results can be viewed as a novel extension of their setting...
> >
> > Finding 8 of [4] again is about noiseless sampling, so it doesn't apply to the authors' method, discriminator guided Langevin diffusion.
> >
> > > Convergence analysis of DG in Section 5.1: We agree that the neither SDE-based convergence/acceleration analysis, nor Theorem 2/Lemma 3 readily apply to the Langevin formulation presented in Section 5 and we are currently exploring approaches to extending the existing results (such as those present in Kim et al., 2023) to this setting. However, as we discuss in Sec 5 (Lines 466-472), and as shown by Li et al. (2024) [6], diffusion models contain two stages: (a) The first stage representing the movement of the Gaussian of diffusion models input towards the image distribution, during which no modes are present, and (b) the second stage, wherein patterns and structure emerge from latching onto a specific image to generate from the target distribution. The proposed DG framework, akin to time-step shifting (Li et al., 2024), can be viewed as improving the convergence of the first stage, while the standard score-based sampling (and the associated convergence guarantees) hold for the second stage.
> >
> > I find this explanation to be quite hand-wavy, as it is not supported with any theoretical analysis.

---

> > > ### Author Response · Authors · 2024-12-01
> > >
> > > Thank you for your response and insightful questions. We would like to provide additional clarity to your questions, and insights obtained via additional results derived.
> > >
> > > - **"Langevin Sampling" without Noise**: We apologise for any confusion that could have occurred in the rebuttal. In the submission (Appendix E.2, Lines 1172-1179), we indeed refer to the noise-free setting when $\gamma_t=0$ as the ODE flow.
> > >
> > > - **Results on Noiseless Sampling**: In this submission, we propose three approaches to closed-form discriminator guidance: (a) Noise-free discriminator-only ODE flow; (b) Discriminator-only Langevin flow, and (c) Closed-form discriminator guidance for score-based Langevin diffusion in the latent space (applied with and without accelerated sampled strategies). While methods proposed in (a) and (b) converge to realistic samples and require no training, they are relatively harder to scale, and do not yield state-of-the-art performance. While the theoretical analysis of Lunz et al., Franceschi et al. apply directly to (a), Franceschi et al. also propose an extension to setting (b) in Appendix A of their paper, which could be applied to the our work as well. Nevertheless, we agree with the reviewer than an analysis of setting (c) is most interesting.
> > >
> > >  - **Analyzing convergence of Closed-form IPM-discriminator-based Guidance**: We would first like to point out that an in-depth convergence analysis of the proposed discriminator guidance is not straightforward. In particular, the closed-form discriminator we derived is optimal in the Wasserstein-2 (W-2) sense, with respect to a varying choice of constraint spaces, the constraint enforced via a regularization, and thereby, the associated kernel in the closed-form solution. However, existing approaches to analyze the optimality of score-based diffusion rely on convergence in terms of the KL-divergence [1,2,3,4,5], just to name a few, and extension to W-2 based bounds are a recent direction for research in the diffusion model setting [6,7,8]. Consequently, the derived bound, in terms of $f$-divergences, although consistent with literature, will be suboptimal. We believe that an extension of Wasserstein-metric-based convergence results to discriminator-guided settings is non-trivia [9], and an interesting direction for future research.
> > >
> > > Nevertheless, to validate the convergence of the proposed approach, we present the following Lemma, which bounds the KL divergence of the learnt and target data distribution in terms of the mismatch in the score estimate, and the discriminator gradient.
> > >
> > > [1] Kim et al., “Refining Generative Process with Discriminator Guidance in Score-based Diffusion Models,” ICML 2023
> > >
> > > [2] Zhang et al., “Convergence of Score-Based Discrete Diffusion Models: A Discrete-Time Analysis,” arXiv 2024
> > >
> > > [3] Benton et al., “Nearly d-linear convergence bounds for diffusion models via stochastic localization,” ICLR 2024
> > >
> > > [4] Lou and Ermon, “Reflected Diffusion Models,” ICML 2023
> > >
> > > [5] Hussain and Nock, “Generalization for Discriminator-Guided Diffusion Models via Strong Duality,” Openreview, 2024
> > >
> > > [6] Backhoff-Veraguas et al., “Adapted Wasserstein Distance between the Laws of SDEs,” arXiv 2024
> > >
> > > [7] Deng et al., “Optimal Wasserstein-1 distance between SDEs driven by Brownian motion and stable processes,” arXiv 2024
> > >
> > > [8] Xia et al., “A Wasserstein-2 Distance for Efficient Reconstruction of Stochastic Differential Equations,” Openreview 2023
> > >
> > > [9] Lavenant and Santambrogio, “The flow map of the Fokker–Planck equation does not provide optimal transport”, SIAM Journal on Mathematical Analysis, 2020.

---

> > > > ### Author Response · Authors · 2024-12-01
> > > > **Convergence of Discriminator-guided Langevin Diffusion**
> > > >
> > > > > *Lemma*:    Consider the reverse diffusion processes associated with the base score-based approach, and the proposed closed-form discriminator guidance model. Let the probability densities associated with these two processes be $p^\*\_t$ and $p\_t$, with $p\_T^\*  = \mathcal{N}(\mathbf{0},\mathbb{I})$, the standard Gaussian distribution and $p\_0^\*=p\_d$ and $p\_0 = p_m$, denoting the data distribution and the \textit{modeled} data distribution, respectively. The, we have:
> > > > $$
> > > >         \mathcal{D}\_{KL,\mathrm{DG}^\*}(p\_d\|p\_m) \leq \mathcal{D}\_{KL}(p\_T^\*\|\pi) + \varepsilon\_{D^\*},
> > > > $$
> > > >     where
> > > > $$
> > > >         \varepsilon\_{D^\*} = \frac{1}{2} \mathbb{E}\_{p^\*\_t}\left[\int g^2(t)\Big\Vert E\_{S^\*} - h(t) \nabla\_{\mathbf{X}} D\_t^\*(\mathbf{X}\_t)\Big\Vert^2\mathrm{d}t\right] = \frac{1}{2} \mathbb{E}\_{p^\*\_t}\left[\int g^2(t)\Big\Vert \nabla D\_{SGAN,t}^\*(\mathbf{X}\_t) - \nabla D\_{t}^\*(\mathbf{X}\_t)\Big\Vert^2\mathrm{d} t\right],
> > > > $$
> > > >     where in turn, $E_{S^\*} = \nabla \ln p_t^*(\mathbf{X}\_t) - {\epsilon}\_{\theta}(\mathbf{X}\_t)$, which is the approxmation error term present in the standard score-based Langevin sampler, and $D\_{SGAN,t}^\*(\mathbf{X}\_t) = \ln \frac{p\_t^\*}{p\_t}$ is the optimal SGAN discriminator.
> > > >
> > > > *Insights*: While we can no longer update the PDF of the submission, we include a proof that follows the approach presented by Kim et al., in the anoymous [Github Link](https://github.com/OrangeIguana/ICLR2025_WANDA/blob/main/WANDA_ICLR2025_Convergence.pdf). The proof follows by defining the stochastic discriminator-guided process applying the Girsanov theorem. During sampling, we show that the gain in $\varepsilon\_{D^\*}$ is the value of the gradient of the discriminator, and positive, owing to the larger support of the initial distribution $p\_t$ for small $t$, this distance between $\mathbf{X}\_t\sim p\_t$ and $p\_{t-1}$ the choice of radially-symmetric and decaying kernels $\kappa$ and the convolution form of the discriminator. The above lemma shows that the use of discriminator guidance yields a positive gain over the standard score-based sampler with error $E\_{S^\*}$, and that the gain can also be bounded by the error between the divergence-based and IPM-based GAN discriminators.
> > > >
> > > > This reuslts show that the **inclusion of DG$^\*$ does indeed result in convergnce to the desired target distribution, with an improved KL-divergence between the true and the learnt data distributions**. However,  **we reiterate that an in-depth analysis of this result and extensions to convergence in terms of the Wasserstein metric, warrants a more thorough theoretical analysis and would be a paper in its own right.**
> > > >
> > > > We hope that the above results/discussion, the theoretical results derived in the context of IPM-GAN generator optimality, links to ODE flows and Langevin diffusion, and strong experimental results and ablations validate the strength of this submission.

---

> ### Comment · Reviewer_QDuM · 2024-12-02
>
> I thank the authors for coming up with a proof in such a brief notice. I find that it strengthens the proposed method significantly, and I have **increased the score from 3 to 5**. I am not sure about increasing the score further for three reasons:
>
> - I feel the paper would have to undergo a major restructuring in order to incorporate the additional theory,
> - the provided proof does not shed any light on the acceleration effect of discriminator guided sampling,
> - there is a discrepancy between theory and practice -- the provided Lemma shows one can apply discriminator guidance to all sampling steps without reduction in sample quality. However, Table 2 in the main paper shows increasing the number of discriminator steps during sampling results in worse FID.

---

> > ### Author Response · Authors · 2024-12-02
> >
> > Thank you for your continued engagement, and for improving your rating of our work. Our responses to your questions are given below.
> >
> > - **Incorporation of theory into the manuscript**: The new insights and results derived as part of the rebuttal can be incorporated into the manuscript without significant change. More precisely, the above lemma and insights can be included in Section 5. The proof of the new lemma has already been incorporated in the Appendix. **We believe that these results do not require a major restructuring as the main claims and results of the papers remain unchanged.**
> >
> >
> > - **Application of the Lemma to WANDA**: The result does not make any assumption on $h(t)$, which is the coefficient of the discriminator gradient. In the WANDA setting, wherein the discriminator is *turned off* after $T\_D$, setting $h(t) = h\_1(t)H\_{T\_D}(t)$, where $H\_{T\_D}(t)$ is the Heaviside/unit-step function with the step at $T\_D$. Furthermore, to understand the convergence result in WANDA, we can simplify the gain derived in the preceding lemma as follows:
> > $$
> > \Big\Vert E\_{S^\*} - h(t) \nabla\_{\mathbf{X}} D\_t^\*(\mathbf{X}\_t)\Big\Vert^2
> > $$
> >
> > $$
> > = \Big\Vert E\_{S^\*} - h(t)\nabla\_{\mathbf{X}} D\_t(\mathbf{X}\_t) + h(t)\nabla\_{\mathbf{X}} D\_t(\mathbf{X}\_t) -  h(t) \nabla\_{\mathbf{X}} D\_t^\*(\mathbf{X}\_t)\Big\Vert^2
> > $$
> >
> > $$
> > = \Big\Vert E\_{S^\*} - h\_1(t)H\_{T\_D}(t)\nabla\_{\mathbf{X}} D\_t(\mathbf{X}\_t) + h\_1(t)H\_{T\_D}(t)\varepsilon\_{\nabla D}\Big\Vert^2
> > $$
> >
> > where $D\_t(\mathbf{X}\_t)$ denotes the sample estimate of the optimal discriminator defined in Equation 8 (L346) of the submission, and $\varepsilon\_{\nabla D}$ denotes the error in estimating the true closed-form discriminator via the sample estimate. The discriminator guidance phase can now be defined as a choice of $T\_D$ such that the gains obtained by the closed-form discriminator remain positive, *i.e.,* select $T\_D$ such that $[ \nabla\_{\mathbf{X}} D\_t(\mathbf{X}\_t) - \varepsilon\_{\nabla D}]_i > 0\~\forall\~i$ (element-wise inequality). However, computing $T\_D$ in closed-form via this approach is impractical as we do not have access to the form or characteristics of $p\_d$ or $p\_{t-1}$ in practice. As discussed in the ablations, this value was found empirically to be around 10% of the total number of iterations, $T$.

---

> > > ### Author Response · Authors · 2024-12-02
> > >
> > > - **Acceleration provided by the closed-form discriminator**: To build intuition, we show that the proposed guidance framework can be viewed as effectively resulting in a second-order update scheme, owing to the form of the discriminator kernel graident. The second-order update resembles Polyak heavy-ball momentum update found in the literature [1,2,3], and can be attributed to being the source for the observed acceleration. Two key contrbuting factor in this analysis are (a) The explicit dependence of the discriminator gradient at time $t$, on the generated distribution at time $t-1$ (appearing in the form of the convolution with $p\_{t-1}$); and (b) the radial symmetry of the kernel ($\kappa(\Vert \mathbf{x}\Vert)$), which always yields a gradient of the form $\mathfrak{c} \mathbf{x} \kappa^{\prime}(\Vert\mathbf{x}\Vert)$. In particular, consider a setting wherein the kernel is a polyharmonic spline kernel of order $k=1$ (cf. Table 3), with the discriminator gradient given by the sample estimate:
> > > $$
> > > \nabla D\_t(\mathbf{X}_t) =   \mathfrak{C}^{\prime}\_k\sum\_{\mathbf{g}^j \sim \{\mathbf{X}\_{t-1}\}} \nabla\_{\mathbf{X}}\kappa(\mathbf{X}\_t-\mathbf{g}^j)- \mathfrak{C}^{\prime}\_k\sum\_{\mathbf{d}^i \sim p\_d} \nabla\_{\mathbf{X}}\kappa(\mathbf{X}\_t-\mathbf{d}^i).
> > > $$
> > >
> > > A simplified single-sample approximation gives
> > >
> > > $$
> > > \nabla D\_t(\mathbf{X}_t) =  \mathfrak{C}^{2}\_k \frac{\mathbf{X}\_{t} - \mathbf{X}\_{t-1}}{\Vert \mathbf{X}\_{t} - \mathbf{X}\_{t-1} \Vert} - \mathfrak{C}^{2}\_k  \frac{\mathbf{X}\_{t} - \mathbf{d}}{\Vert \mathbf{X}\_{t} - \mathbf{d} \Vert},
> > > $$
> > > where $\mathbf{d}$ is a random sample drawn from the target data distibution. Consider the standard closed-form discriminator guided Diffusion update:
> > > $$
> > > \mathbf{X}\_{t+1} = \alpha\_{1,t} \mathbf{X}\_t - \alpha\_{2,t} \epsilon\_{\theta}(\mathbf{X}\_t) - \alpha\_{3,t} \nabla D\_t(\mathbf{X}\_t) + \alpha\_{4,t} \mathbf{Z}\_t
> > > $$
> > >
> > > Substituting in for the above discriminator gradient and simplifying results in an update of the form:
> > >
> > >
> > > $$
> > > \mathbf{X}\_{t+1} = \beta\_{1,t} \mathbf{X}\_t - \alpha\_{2,t} \epsilon\_{\theta}(\mathbf{X}\_t) - \beta\_{3,t} \mathbf{X}\_{t-1} + \alpha\_{4,t} \mathbf{Z}\_t + \beta_{5,t},
> > > $$
> > > where $\beta\_{1,t} = \alpha\_{1,t} - \frac{\alpha\_{3,t}\mathfrak{C}^{2}\_k}{\Vert \mathbf{X}\_{t} - \mathbf{X}\_{t-1} \Vert} + \frac{\alpha\_{3,t}\mathfrak{C}^{2}\_k}{\Vert \mathbf{X}\_{t} - \mathbf{d} \Vert}$,  $\beta\_{3,t} = \frac{\alpha\_{3,t}\mathfrak{C}^{2}\_k}{\Vert \mathbf{X}\_{t} - \mathbf{X}\_{t-1} \Vert}$ and  $\beta\_{5,t} = \frac{\alpha\_{3,t}\mathfrak{C}^{2}\_k}{\Vert \mathbf{X}\_{t} - \mathbf{d} \Vert}$ . The above equation defines a second-order update, which resembles the update schemes encountered in momentum-based diffusion models [3] --- we hypothesize that this is one of the sources of *acceleration* in the proposed technique.
> > >
> > > [1] F. Bach, "[Lecture notes on *Statistical Machine Learning and Convex Optimization*](https://www.di.ens.fr/~fbach/fbach_orsay_2018.pdf)," 2018
> > >
> > > [2] B. Recht and S. J. Wright, "Optimization for Modern Data Analysis," Cambridge University Press, 2022
> > >
> > > [3] Wu et al, "Fast diffusion model," arXiv:2306.06991, 2023
> > >
> > >
> > > Thank you for your questions. We hope that the preceding discussion gives some insight into the acceleration achieved by the proposed approach. We will incorporate a discussion on these results while revising the manuscript.

---

### Official Review · Reviewer_9w6G · 2024-10-28

**Soundness:** 3
**Presentation:** 2
**Contribution:** 2
**Rating:** 5
**Confidence:** 1

**Summary:**

**Disclaimer**: I am not familiar with the theorem of optimal GANs. While I briefly reviewed some related works referenced in this paper, I am still not confident in assessing the significance or complete accuracy of the conclusions for the GANs. My rating is primarily based on the experimental results of using this form in diffusions. I am open to discussing this further with the author or other reviewers.

This paper proves the analytical form for optimal generators in IPM GANs. The authors then connect this form to score matching. Then, the authors develop a closed-form discriminator guidance for diffusion models.

**Strengths:**

1. This paper extends previous conclusions on $f$-GAN and on IPM GAN discriminators to IPM GAN generators.
2. The optimal IPM GAN generators naturally connect to diffusion models.
3. The proposed discriminator guidance is plug-and-play. Unlike previous discriminator guidance, it does not require additional training and seems compatible with different DM frameworks.

**Weaknesses:**

As I disclaimed, I am not confident in assessing the significance or complete accuracy of the conclusions for the GANs. Therefore, my questions and concerns will mainly be related to experiments.

1. Experimental results are not fully convincing to me. I did not see gains on CIFAR-10 and ImageNet-64 datasets using the proposed discriminator guidance.
The authors claimed that the proposed guidance could accelerate convergence, leading to fewer time steps in image generation. However, by looking at Fig 15 and 16, it is very obvious that the results of the guidance are much worse than the baseline.
I think a better way to illustrate the effectiveness is to compare the generated samples obtained with the same budget (NFEs). Also, visually checking the results is not enough; FID or other metrics need to be provided for a fair comparison.


2. Experimental settings are not fully clear. If I understand it correctly, the score is estimated by the MC estimators. However, I think I missed the details on the number of samples. Also, the MC estimator requires a batch of generated samples. I did not find details of how this is obtained. I guess it is only possible to draw a batch of samples (instead of 1) at each sampling time. Is this correct? In this way, we can use the samples at the last time step in this batch for the MC estimator.
If this is the case, I would be curious how the number of samples influences the performance. Especially in large-res applications like text-2-image generation, requiring a large batch for each prompt seems too much.

3. I am also a bit confused on the ablation. In Table 2, it seems that the more steps you apply discriminator guidance, the worse result you are getting. So what is the results when $T_D=0$ (w/o any discriminator guidance)? I saw there is one line saying that there is a surge around $T_D=10$. However, seeing this in plots or tables would be more convincing.
Also, why Table 2 only shows different $T_D$ for WANDA. What about LDM+DG?

**Questions:**

1. Line 297-299:
>From Lemma 3, we see that the gradient field of the kernels convolved with the density difference, and the data score $\nabla ln p_d(x)$, serve similar purposes, which is to output an arbitrarily large value at data sample location, and low values elsewhere.

I am quite confused about this: I did not see the data score in Lemma 3?

2. Line 374-376:
>Since the proposed approach suggests the interoperability of the score and the discriminator-kernel gradient in Langevin flow, we also consider discriminator-guided Langevin sampling on the CIFAR-10 and ImageNet-64 datasets, considering EDMs as the baseline

Do you mean just adding the discriminator-kernel gradient and the score together (with some weighting for the guidance strength)?
Also, for guidance, how do you choose the guidance strength?


3. what does the (RGB Color online) mean in the caption in figures?

**Details Of Ethics Concerns:**

This paper has substantial overlap with arXiv:2306.01654. Could the AC, SAC, or PC verify if this indicating any potential plagiarism?

---

> ### Author Response · Authors · 2024-11-22
> **Response to Reviewer's Comments**
>
> Thank you for your valuable feedback! Please find a detailed response to your questions below.
>
> - **Experiments of CIFAR10 and ImageNet64**: The experiments with discriminator guidance (DG$^*$) and EDM sampler are carried out in the image space. As already noted in the caption of Figures 15 and 16, the DG$^*$ approach leads to sub-par image quality on pixel-space generation due to challenges in computing the closed-form discriminator in high-dimensional spaces, which is why we extend DG$^*$ to work with LDMs in Sec. 5. For completeness, we now include experiments on CIFAR-10 with the DPM sampler [1] with and without DG$^*$, applied on the image space. The DPM+DG$^*$ model archives a CLIP-FID of 12.07, compared to the CLIP-FID of 11.51 achieved by the baseline DPM solver. As expected, the model is comparable to, but does not outperform the baselines in the pixel space. We also improve the clarity on the limitation of the experiments with EDM sampler in Lines 411-419 of the revised manuscript.
>
> To further demonstrate the experimental performance of LDM+DG$^*$, we now also report FID scores on the LSUN-Churches-256 dataset, which we summarize in the table below, with additional ablations present in Table 6 of the revised manuscript.
>
> | Method | Clean-FID | KID | CLIP-FID
> |---|---|---|---|
> |LDM| 6.67 |  0.0039 |  4.89 |
> |LDM+DG$^*$| 6.50 |  0.0032 |  4.80 |
>
> [1]  DPM-Solver: A Fast ODE Solver for Diffusion Probabilistic Model Sampling in Around 10 Steps, NeurIPS 2022
>
> - **Score estimation**: In all experiments, the score is computed using the neural network, identical to the corresponding baselines, using the provided pre-trained models. There is no Monte-Carlo sampling required to estimate the score. The closed-form discriminator is computed as a sample estimate using a batch of $M \leq B$ centers, where $B$ is the batch size, and can be selected based on the available compute budget. However, since the discriminator guidance only affects the initial stages of sampling, the choice of $M$ does not affect the generated image quality. To illustrate this effect we now include Figure 21 in the Appendix, which presents images generated for varying numbers of centers \(M\) considered in the closed-form discriminator.
>
> - **WANDA vs LDM-DG**$^*$: $T_D$ is unique to the WANDA algorithm, is the the application of Time-Shifted Sampling (Li et al., 2024) using wavelet-based noise estimation for of the LDM-DG$^*$ algorithm. $T_D$ denotes the time step up to which time-shifting is applied. We have improved the clarity of this distinction in Sec 5.1 of the revised manuscript.
>
> - **The choice of $T_D$**: $T_D$ denotes the transition point from discriminator guidance to score-based sampling in WANDA. When $T_D=0$, there is no discriminator guidance and this corresponds to the baseline LDM algorithm, which is worse than WANDA/LDM+DG$^*$. Intuitively, it is beneficial to include discriminator guidance in the early stages of the samples process, but as we note in Fig. 6 of the Appendix, the iterate convergence (the value of $\|\mathbf{x}_{t-1} - \mathbf{x}_t\|$ as t decreases) is much quicker for DG$^*$. Therefore, a small number of closed-form discriminator guidance steps suffice to improve generation quality.  Please also see our response to Reviewer #Z719, and Table 5 of the Appendix in the revised manuscript for additional details.
>
> - **On Lemma 3**: Lemma 3 does not contain the data score, but draws an analogy between the data score $\nabla \ln p_d$ and the gradient field of the kernels convolved with the density difference, which is what we discuss in Lines 297-299. To illustrate this, we have included a new Figure 19  in the Appendix, comparing the data score and the discriminator kernel gradient field.
>
> - **Combining closed-form discriminator guidance and score**: The score and the closed-form discriminator as combined as shown in Equations on Lines 412-416 (Lines 426-429 of the revision). The guidance strength is given by $w_{dg,t}$. In LDM-DG$^*$, the weight is decayed exponentially, as explained in Lines 417-420, with ablations given in Section 5.1 and Table 5 of the Appendix.
>
> - **RGB Color online**: The note is for accessibility and is meant to indicate that the figure includes color-based disambiguation and/or is best viewed digitally, instead of on-print media.
>
> Please let us know if any additional clarity is needed. We hope that we were able to address your concerns regarding our submission and that you'll consider raising your score in case the questions were resolved.

---

> > ### Comment · Reviewer_9w6G · 2024-11-25
> >
> > Thank you for your classification.
> >
> > 1. **Experiments of CIFAR10 and ImageNet64**. I am still not convinced by the explanation. For a fair comparison, it is more convincing to say, "We achieve the same FID (and also visual performance as FID may be misleading!) with fewer steps, or we achieve much better quality with the same number of steps". To me, it looks like you use fewer steps and obtain worse quality. It is not even clear if the proposed approach is better in the Pareto sense.
> >
> > 2.
> > >$T_D$ denotes the transition point from discriminator guidance to score-based sampling in WANDA. When $T_D=0$
> > , there is no discriminator guidance and this corresponds to the baseline LDM algorithm, which is worse than WANDA/LDM+DG*.
> >
> > Yes but I did not see the baseline performance in the table? Also, my concern is when $T_D$ gets larger, the performance seems to be worse. So a natural concern is how this $T_D$ influence the performance. Does FID decreases first, and then increase?

---

> ### Author Response · Authors · 2024-11-25
>
> Thank you for your response:
>
> - **On the performance on CIFAR-10**: We now also include a comparison on the images generated by the DPM+DG$^*$ sampler, compared against the baseline DPM sampler, in Figure 23 of the revised manuscript. The figure clearly shows that the images generated are visually on par with state-of-the-art performance of DPM.
>
> - **On the visual quality of the generated images**: As we have shown on the FFHQ and CelebA-HQ datasets, the proposed approach (LDM+DG$^*$) does indeed result in improved visual quality. This is evident from Figure 4 of the main manuscript, and Figures 17 and 18 of the Appendix. As stated in the response above, we also show this for CIFAR-10 with the DPM sampler in Figure 23.
>
> - **On the effect of $T_D$**: The baseline corresponds to the line labelled LDM in Table 1. As $T_D$ increases (increasing the number of steps for which we provide guidance, i.e., the first $T_D$ steps starting from iteration $t=T$) the performance indeed improves at first, and then deteriorates, which we also observe from the ablations in Table 5 -- We see that setting $T_D$ to about 10% of the total number of iterations $T$ is a viable choice. This behaviour has already been observed in the literature by Li et al. (2024) in the context of standard time-step shifting. Diffusion models basically contain two stages: (a) The first stage representing the movement of the Gaussian of diffusion models input towards the image distribution, during which no modes are present, and (b) the second stage, wherein patterns and structure emerge from latching onto a specific image to generate from the target distribution. Acceleration mechanisms such as time-step shifting (Li et al., 2024) and the proposed DG$^*$ operate in the first stage, which is why we focus the discriminator guidance to earlier iterations.
>
> We hope that we were able to address your concerns.

---

> > ### Comment · Reviewer_9w6G · 2024-11-25
> >
> > Thank you for your further clarification. I agree that Figure 23 provides a more convincing argument. However, Figures 15 and 16 highlight scalability issues with the proposed approach, as confirmed in your previous reply.
> >
> > Based on this, I have decided to increase my score to 5.
> >
> > At the same time, I want to emphasize that my score is primarily based on the experimental results, where the performance gain of the proposed approach appears relatively modest. As mentioned in my disclaimer, I do not feel confident in assessing the significance of the conclusions for GANs. Accordingly, I have adjusted my confidence level to 1.

---

> ### Author Response · Authors · 2024-11-26
>
> We would like to thank you for your continued discussion. While we understand that your evaluation of our work in not centered about the GAN framework presented, we would like to emphasize that, from an experimental standpoint, we have shown the scalability of the proposed DG$^*$ to generate high-dimensional images by means of the LDM and DPM samplers, and the time-shift scaling acceleration. In these scenarios, inclusion of discriminator guidance demonstrated improvements over each corresponding baseline model, both in FID, and visual quality. Further, we would like to add that:
>
> 1. While we mention brute-force scaling of the closed-form discrimination guidance as challenging in the pixel space, we note that *it is possible to achieve performance gains in the pixel space*, and it requires an in depth analysis which is beyond the scope of this paper. The experiments on CIFAR-10 with the DPM sampler (Figure 23) are a proof of concept for this. For example, to scale, approaches such as improved batching/batch sampling strategies from other domains [4,5] could be employed to pick better centers used in DG$^*$, without scaling batch size $M = B_{DG^*}$ with dimensionality $n$. One could also explore clustering-based approaches to choose more representative centers. Note that $B_{DG^*}$ can be independent of the number of images being generated, $B$. These are all promising directions for future research.
> 2. As you rightly point out, “text-2-image generation” is a crucial direction for diffusion models and latent-space approaches, particularly LDM-based, are state-of-the-art, e.g., [1,2,3]. Given that DG$^*$ has shown performance gains with LDM, (Tables 1,4,6, Figures 4, 17 and 18), we believe that scalability is not an issue with the proposed approach and  closed-form discriminator guidance can also be applied in this setting in the latent space.
>
> [1] (SDXL) Podell  et al., “SDXL: Improving Latent Diffusion Models for High-Resolution Image Synthesis,” ICLR 2024 Spotlight
>
> [2] (Diffusion-ViT)  Bao et al., “All are Worth Words: A ViT Backbone for Diffusion Models,” CVPR 2023
>
> [3] (RAT-Diffusion) Ye et al., “Recurrent Affine Transformation for Text-to-Image Synthesis” Trans. MM, 2023
>
> [4] Dahiya et al., “NGAME: Negative mining-aware mini-batching for extreme classification,” WSDM 2023
>
> [5] Xiong et al., “Approximate Nearest Neighbor Negative Contrastive Learning for Dense Text Retrieval,” ICLR 2021
>
> We hope that this is taken into consideration in rating our work positively, in terms of the experimental merit demonstrated.

---

### Official Review · Reviewer_Z719 · 2024-11-03

**Soundness:** 3
**Presentation:** 2
**Contribution:** 3
**Rating:** 6
**Confidence:** 3

**Summary:**

**Summary**
This paper seeks to extend the concept of discriminator guidance to the Integral probability metric(IPM) GAN model, with the ultimate goal of applying this framework to diffusion models.
This paper derives a more general solution to the GAN-generator optimality using mathematical proof from the **Theorem 1**(optimal f-GAN generator). From this general solution, this paper aims to achieve the goal of **Closed-form Discriminator Guidance**. Following the mathematical approaches, this paper presents a series of comprehensive experiments.

**Contribution**
In addressing generative models, the concepts from GAN have been applied to diffusion model. While prior attempt focused on applying the discriminator of standard GAN to diffusion, this paper expands on that by applying the discriminator from the IPM-GAN framework to diffusion models.

**Strengths:**

**Originality & Significance**
- This paper derive the more general solution to optimal GAN generator.
- Make connection between the solution to optimal IPM-GAN generator and diffusion model.

**Quality**
- This paper demonstrates superior performance in terms of metrics and training time.

**Clarity**
- This paper employs mathematical proof to substantiate its claims.
- In **5.1 ablations** and **Appendix**, the paper rigorously designs and presents robust experiments that substantiate its claims.

**Weaknesses:**

1. It would be better if you could give the explanation for transition from GAN generator optimality to discriminator optimality.
: This paper discusses GAN's alternating optimization and the optimality of the GAN generator. Subsequently, it mentions the transfer of generator optimality to discriminator optimality, but lacks sufficient explanation.

2. In lines 519 (520) - 521, how can this paper assert that 'there is a stark jump initially of about ___10 or so steps___ via the ~'?
: Further explanation or experimental results are required for those figures.

3. Need more ablation for ___DG*___.
In lines 454 (455) - 457 (458), where it states, 'Motivated by the fact ~ not accurate.' the paper asserts that ___DG*___ is not always useful. Subsequently, in section 5.1, the paper presents results for the first $T_D$ = {50, 100, 200} steps using the discriminator.
Why does the paper focus only on the first $T_D$ steps? What about the later stages of training? Could you provide some toy examples or explanations to clarify this?


**Typos**
1. In line 199 : $\epsilon$ -> $\epsilon_t$

**Questions:**

1. In line 479, what is the meaning of $w_{dg, 0}$?
Table 2 may need to be revised for clarity.


2. In the last paragraph of section 5, it appears that the ablation study for $T_D$ is conducted with respect to WANDA.
However, how is this approach applied to LDM-DG? What is the justification for selecting $T_D = 50 $ for LDM-DG$^*$?

---

> ### Author Response · Authors · 2024-11-22
> **Response to Reviewer's Comments**
>
> Thank you for your valuable feedback! Please find a detailed response to your questions below.
>
>  - **On the GAN generator optimality and discriminator optimality:** The optimality of the GAN discriminator has previously been studied by Mroueh et al.,2018; Unterthiner et al., 2018; Liang, 2021; Franceschi et al. 2022, and Asokan et al., 2023. What we show in this submission, is that, given the optimal discriminator $D_{t-1}^*$ (that admits a kernel-based interpolation form) at training iteration $t-1$, the optimal generator at the subsequent iteration $G_t^*$ can be derived as a one that minimizes the value of the convolution between the density difference, and the gradient of the optimal discriminator kernel $((p_d - p_{t-1}) *\nabla\kappa)$. For most popular positive-definite kernels $\kappa$ (cf. Table 3 of the Appendix), this term would be minimized when the generator’s distribution $p_t$ moves towards the data distribution $p_d$. We have incorporated this improved discussion on this optimality of the generator and discriminator in Section 3.1 of the revision.
>
> - **On the jump post discriminator guidance**: Thank you for the suggestion. To further validate our claim, we have included an experiment wherein we plot the time-step jump predicted by the noise-variance-based time-shifted sampler at each step $t$. Since the step can occur at different $t$ for different images, we plot this curve. We performed the experiment over multiple images and observed that on average, the jump is about 2-10\% of the total steps. Illustrative plots of the predicted time vs the actual time $t$ of the iteration, wherein the discriminator guidance improves performance gains over the baseline time-shifting algorithm are provided in Figure 22 of the revision.
>
> - **Focussing on earlier iterations with DG**$^*$: As we show in Fig. 6 of the Appendix, on  comparing the iterate convergence of the closed-form discriminator-guided diffusion (DG$^*$) against the baseline NCSNv1, we observe that the iterate convergence (the value of $\|\mathbf{x}_{t-1} - \mathbf{x}_t\|$ as $t$ decreases) is much quicker for DG$^*$ -- which suggests that it is beneficial for the sampler to introduce the discriminator early on and subsequently switch to the score-based model. Furthermore, as noted by Li et al. (2024), diffusion models basically contain two stages: (a) The first stage representing the movement of the Gaussian of diffusion models input towards the image distribution, during which no modes are present, and (b) the second stage, wherein patterns and structure emerge from latching onto a specific image to generate from the target distribution. Acceleration mechanisms such as time-step shifting (Li et al., 2024) and the proposed DG$^*$ operate in the first stage, which is why we focus the discriminator guidance to earlier iterations. We have included this clarity of the revised manuscript (cf. Section 5)
>
> - **Ablations on DG**$^*$: We have now included additional experiments on DG$^*$, based on the choice of $T_D$ and $w_{dg,t}$ on the FFHQ dataset. The detailed experimentation is present in Table 5 of the Appendix. In summary, we observe that discriminator guidance performs best when run for less than 20\% of the overall iterations (*i.e.,* $T_D=5$ for $T=50$ or $T_D=5,10$ for $T=100$) and with the discriminator weight $w_{dg}\in(0.5,1)$. We observe similar trends when running discriminator guidance with the DPM solver [1] (Please see our response to Reviewer QDuM).
>
> - **Ablations in the last paragraph of Sec. 5**:. You are right in your observation that this experiment is w.r.t. WANDA. We have fixed this bug.
>
> - **Meaning of $w_{dg,0}$**: $w_{dg,0}$ is the weight given to the discriminator guidance term at $t=0$. We have improved the clarity in Section 5.1 of the revised manuscript.
>
> Please let us know if any additional clarity is needed. We hope that we were able to address your concerns regarding our submission and that you'll considering raising your score in case the questions were resolved.
>
> [1]  DPM-Solver: A Fast ODE Solver for Diffusion Probabilistic Model Sampling in Around 10 Steps, NeurIPS 2022

---

> > ### Comment · Reviewer_Z719 · 2024-11-26
> >
> > Thank you for your explanation.  I'll keep my score.

---

### Author Response · Authors · 2024-11-25
**A Summary of our Responses and Revision**

We would like to thank the various reviewers for taking the effort to review our paper! We would like to highlight the work's strengths identified by the reviewers:

1. Reviewer Z719 and QDuM both appreciated the **Originality and Significance** of the derived optimal GAN generator solution.
2. **All Three Reviewers praise the derived natural connections to diffusion models**, and the mathematics proofs provided to substantiate our claims.
3. Reviewer Z719 remarks that the "*the paper rigorously designs and presents robust experiments that substantiate its claims*."
4. Reviewer 9w6G also remarks positively on the **plug-and-play** nature of the proposed novel discriminator guidance, which "*does not require additional training and seems compatible with different DM frameworks.*"
5. Reviewer QDuM remarks that, to the best of their knowledge, this submission is "*the first paper to use IPM-GAN discriminators for diffusion sampling and guidance.*"
6. Reviewer QDuM finds the paper well motivated and clearly presented.


In light of the reviewers’ comments, we carried out various ablations/experiments to further substantiate our claims. We summarize the changes we have incorporated as a result of the feedback received from the reviewers:

1. We have included additional *rigorous* ablations on the choice of the transition time $T_D$ and the guidance weight *w_{dg,t}* for WANDA with the LDM sampler (Tables 5-6 of the revision) which, on CelebA-HQ Dataset, result in state-of-the-art **improvements over baseline samplers with DG**$^*$ **applied for only 20\% of the overall iterations.**
2. We provide ablation on the time step jump obtained by the WANDA samples, and show that **DG**$^*$ **provides a jump of 2-10\% of the total steps.**
3. We also include new experiments on sampling with the LDM sampler on the **LSUN-Churches dataset** to strengthen our results and show that **LDM+DG**$^*$ **once again outperforms the baseline**.
4. We include experiments on the DPM sampler (As suggested by Reviewer QDuM) and show that using **DG**$^*$ **over DPM improves FID on the same number of sampling steps, and archives comparable FID with 25% fewer steps (reduction from 20 to 15)**. (Table 4) on CelebA-HQ and similar gains on pixel-space CIFAR-10.
5. For completeness, we also included a summary of the proof that shows that the generator distribution models the data distribution in the proposed setting (Appendix D.1)

We request that the reviewers consider reverting with additional questions, if any, or consider raising their score in case their questions were resolved.

---

### Author Response · Authors · 2024-12-04
**Summary of Rebuttal Discussions**

We would like to thank the reviewers for taking the effort to actively engage during the rebuttal phase. In addition to the revisions summarized in the global response, we would like to make our closing arguments and highlight the discussions that took place as part of the rebuttal:

1. **Reviewer Z719** appreciated the *Originality and Significance* of the derived optimal GAN generator solution, the *superior performance in terms of metrics and training time*, and the rigorously designed and presented, robust ablations that validate the claims. Post the rebuttal, the reviewer **had no additional concerns** regarding the submission.

2. **Reviewer 9w6G** appreciated the extension of conclusions on $f$-GAN to IPM-GAN discriminators and IPM-GAN generators, the connections between the optimal IPM GAN generators and diffusion models, and the *plug-and-play* nature of the proposed discriminator guidance, which does not require additional training and is compatible with different diffusion model frameworks.

     Post the rebuttal, Reviewer 9w6G **increased their score** and was satisfied with the additional experimental results, ablations, and figures of generated images presented on CIFAR-10 dataset. We have also satisfactorily addressed the reviewer’s  concerns on scalability, by emphasizing our experimental results of the proposed DG$^*$ to generate high-dimensional images by means of the LDM and DPM samplers on CelebA-HQ, FFHQ, and LSUN-Churches.

3. ⁠**Reviewer QDuM** praised the *originality* of using a closed-form expressions for the optimal discriminator, remarking that, *to the best of their knowledge, this is the first paper to use IPM-GAN discriminators for diffusion sampling and guidance*. The reviewer also commended the *quality, clarity of presentation and motivation of the theoretical results* and the significance of the proposed approach.

    Post rebuttal, Reviewer QDuM also **increased their score** and found that we were able to address all concerns regarding experiments. The reviewer also appreciated inclusion of the convergence proof as part of the rebuttal, and found that it strengthens the proposed method significantly. We also satisfactorily addressed the reviewer’s concerns on (a) incorporating these results into the revision; (b) quantifying the acceleration demonstrated by the proposed approach using a momentum type argument; and (c) application of the derived lemma on convergence to strengthen the proposed WANDA setting.

Thanks to the intellectually stimulating and positive interaction during the rebuttal phase, the acceptability of the manuscript has increased significantly. We sincerely hope that both reviewers and Area Chairs take cognizance of this fact and enhance their scores and find our submission worthy of acceptance to ICLR.

---

### Meta-Review · Area_Chair_zaZq · 2024-12-17

**Metareview:**

This paper derives a closed-form expression for the IPM-GAN discriminator, whose gradient is then used to perform Langevin sampling or guide diffusion generation.

While this is an interesting result, the current form of the work is not ready for publication due to the following reasons:
1. Structural changes to the paper are needed to incorporate the extra theory and experiments to address the concerns;
2. The discrepancy between theory and practice on the acceleration effect of discriminator-guided sampling has not been properly addressed.
3. The improvement in experimental results is not convincing enough.
4. The ethical concerns have not been properly addressed.

**Additional Comments On Reviewer Discussion:**

The authors have partially addressed the concerns on theory and practice raised by the reviewers.
However, due to the above reasons, the paper cannot be accepted in the current form.

---

### Decision · Program_Chairs · 2025-01-22

Reject